# New Algorithms for the Learning-Augmented $k$-means Problem

**Junyu Huang**
School of Computer Science and Engineering,
Central South University
Xiangjiang Laboratory
Changsha, China
junyuhuangcsu@foxmail.com

**Qilong Feng** [*]
School of Computer Science and Engineering,
Central South University
Xiangjiang Laboratory
Changsha, China
csufeng@mail.csu.edu.cn

**Ziyun Huang**
Department of Computer Science and
Software Engineering, Penn State
Erie, The Behrend College
Erie, United States
zxh201@psu.edu

**Zhen Zhang**
School of Advanced Interdisciplinary Studies,
Hunan University of Technology and Business
Xiangjiang Laboratory
Changsha, China
csuzz@foxmail.com

**Jinhui Xu** [†]
School of Information Science and Technology,
University of Science and Technology of China
Hefei, Anhui, China
jhxu00@gmail.com

**Jianxin Wang** [*]
School of Computer Science and Engineering,
Central South University
Hunan Provincial Key Lab on Bioinformatics
Xiangjiang Laboratory
Changsha, China
jxwang@mail.csu.edu.cn

## Abstract

In this paper, we study the clustering problems in the learning-augmented setting, where predicted labels for a $d$-dimensional dataset with size $m$ are given by an oracle to serve as auxiliary information to enhance the clustering performance. Following the prior work, the given oracle is parameterized by some error rate $\alpha$, which captures the accuracy of the oracle such that there are at most $\alpha$ fraction of false positives and false negatives in each predicted cluster. In this setting, the goal is to design fast and practical algorithms that can break the computational barriers of inapproximability for clustering problems. The current state-of-the-art learning-augmented $k$-means algorithm relies on sorting strategies to find coordinates approximation, where a $(1 + O(\alpha))$-approximation can be achieved with near-linear running time in the data size. However, the sorting process may limit the scalability of the algorithm for handling large-scale datasets. To address this issue, in this paper, we propose new algorithms that can identify good coordinates approximation using sampling-based strategies, where $(1+O(\alpha))$-approximation can be achieved with linear running time in the data size. To obtain a more practical algorithm for the problem with better clustering quality and running time, we propose a sampling-based heuristic which can directly find center approximations for each cluster. Empirical experiments show that our proposed methods are faster than the state-of-the-art learning-augmented $k$-means algorithms with comparable performances on clustering quality.

---

[*]Corresponding Authors
[†]The research of Jinhui Xu was partially done at the State University of New York at Buffalo.

## 1 INTRODUCTION

Clustering is a fundamental unsupervised learning task that has been extensively studied over the past decades. Among different clustering objectives, one of the most commonly used clustering formulations is the $k$-means clustering. In the $k$-means clustering, we are given a set $P$ of data points in a $d$-dimensional Euclidean space, and the goal is to compute a set $C \subset \mathbb{R}^d$ of centers with size at most $k$ such that the sum of the squared distances between data points in $P$ to their closest centers in $C$ is minimized. The $k$-means problem has received significant attention in the literature and is proven to be NP-hard (Dasgupta, 2008). Furthermore, Cohen-Addad and Karthik (Cohen-Addad & Karthik, 2019) showed that even finding a solution for $k$-means with approximation ratio smaller than 1.07 is NP-hard. The current best approximation ratio for the $k$-means problem is 5.912 (Cohen-Addad et al., 2022), which is based on primal-dual and nested quasi-independent set methods. For fixed dimensionality $d$ or the number of clusters $k$, several $(1 + \epsilon)$-approximation algorithms were known (Jaiswal et al., 2014; Friggstad et al., 2019). However, these algorithms with relatively tight approximation guarantees do not scale well for large-scale datasets. Thus, several practical methods with linear running time in the data size have been proposed, such as the $O(\log k)$-approximation $k$-means++ method (Arthur & Vassilvitskii, 2007) and the $O(1)$-approximation local search methods (Lattanzi & Sohler, 2019; Beretta et al., 2023; Fan et al., 2023; Choo et al., 2020). Although these linear-time algorithms have been widely used in practice, their large approximation ratios could potentially deteriorate the clustering performance in scenarios that require high-quality solutions.

To overcome the barrier of inapproximability and develop more practical approximation algorithms, a series of studies has focused on algorithms augmented with predictions (Mitzenmacher & Vassilvitskii, 2022; Ashtiani et al., 2016; Kraska et al., 2018; Mitzenmacher, 2018). For the clustering problem, Gamlath et al. (Gamlath et al., 2022) proposed clustering recovery with noisy labels, where predicted clustering labels are given as additional information to aid the clustering process. The given predictor is parameterized by some error rate $\alpha \in [0, 1)$, such that the size of the symmetric difference between the predicted cluster and its corresponding optimal cluster is bounded by $\alpha$ times the size of the optimal cluster. Based on this model, Gamlath et al. developed a $(1 + O(\alpha))$-approximation algorithm with polynomial running time, assuming fixed $k$ and $d$. Moreover, to guarantee the approximation, the error rate $\alpha$ is required to satisfy the condition that $\alpha < 1/4$. Ergun et al. (Ergun et al., 2021) introduced another learning-augmented clustering model, aiming at designing fast and practical algorithms for clustering. In their setting, a given predictor also provides auxiliary information for each data point in the form of predicted clustering labels. However, the predictor's reliability is parameterized by an error rate $\alpha \in [0, 1)$, ensuring that there are at most an $\alpha$ fraction of false positives and false negatives in each predicted cluster. Based on this model, an improved $(1 + O(\alpha))$-approximation can be achieved with near-linear running time in the data size.

In this paper, we mainly focus on the learning-augmented clustering problem proposed by Ergun et al. (2021). The motivation for studying the learning-augmented clustering problem is as follows. From a theoretical perspective, learning-augmented $k$-means can overcome the inapproximability barriers, enabling the development of algorithms that can achieve tight clustering quality guarantees with strong scalability. From a practical perspective, as pointed out by Ergun et al. (2021), reliable predictors are available for a wide range of natural datasets. For instance, in datasets with training labels, these labels can act as auxiliary information to enhance the clustering quality on testing datasets. Furthermore, when the clustering instances adhere to specific distributions, the efficient acquisition of a robust predictor is proved to be feasible (Ergun et al., 2021). Even if the instances are not generated from such distributions, empirical evidence (Ergun et al., 2021; Nguyen et al., 2022) suggests that labels produced by the existing clustering methods, such as $k$-means++ (Arthur & Vassilvitskii, 2007) or heuristic approaches (Lloyd, 1982), can serve as good predictors. However, as pointed out in Nguyen et al. (2022), even when the auxiliary labeling partition (or the predictor) is nearly optimal, a single false positive located far from the true clustering centers can significantly influence the clustering structures. Therefore, a key challenge in the learning-augmented clustering problem is to design robust algorithms that can minimize the impact of false positives.

Based on statistical methods, Ergun et al. (Ergun et al., 2021) proposed a randomized algorithm that can achieve a $(1 + 20\alpha)$-approximation in time $O(md \log m)$, where $m$ and $d$ are the data size and dimension, respectively. However, to guarantee the approximation loss, it is required that the label error rate $\alpha$ should be bounded by $\alpha \in [10 \log m/\sqrt{m}, 1/7]$, and each optimal cluster should have size $\Omega(k/\alpha)$. To overcome the label error rate and cluster size constraints, Nguyen et al. (2022) pro-

posed a deterministic searching method, where an improved $(1 + O(\alpha))$-approximation result with better approximation guarantees can be obtained in time $O(md \log m)$ for $\alpha \in [0, 1/2)$. The current learning-augmented clustering algorithms (Ergun et al., 2021; Nguyen et al., 2022) primarily rely on sorting strategies to approximate the optimal clustering centers. However, since sorting processes require near-linear running time in the data size, this can limit the scalability of the existing algorithms for handling large-scale datasets. Moreover, it is well known that comparison-based sorting has a lower bound running time of $O(m \log m)$. As a result, achieving a time complexity better than $O(md \log m)$ using sorting-based strategies is a non-trivial task. It is also worth mentioning that the time complexity of Ergun et al. (2021) and Nguyen et al. (2022) cannot be further improved through dimensionality reduction techniques, such as the JL-method. As pointed out in Theorem 3.4 of Ergun et al. (2021), the JL-method embeds a clustering instance from $\mathbb{R}^d$ into a space of dimension $O(\log m)$ and $O(\log k)$ in $O(md \log m)$ and polynomial time, respectively. Thus, the total running time of using JL-method is still at least $O(md \log m)$. A central challenge in designing faster algorithms for the learning-augmented $k$-means problem is to efficiently approximate the optimal clustering centers in each dimension without using sorting-based strategies.

## 1.1 OUR CONTRIBUTION

In this paper, we aim to further enhance the efficiency of learning-augmented clustering algorithms by proposing new sampling-based methods. A key challenge here is to identify high-quality coordinates without relying on sorting-based strategies. To overcome this challenge, we first propose a sampling-based algorithm called Fast-Sampling. For each dimension of each predicted cluster, Fast-Sampling can identify intervals that contain coordinates sufficiently close to the coordinates of the optimal clustering centers. High-quality coordinates are then constructed within linear running time in the data size through fine-grained divisions of the intervals. With these techniques, a $(1 + O(\alpha))$-approximate solution for the learning-augmented $k$-means problem can be obtained in time $O(md \log(kd))$. To further improve the running time and eliminate the additional $O(\log(kd))$ term, we propose another algorithm called Fast-Estimation. Fast-Estimation accelerates coordinate approximation by designing estimators that provide accurate clustering cost estimates in sublinear time, assuming a bounded aspect ratio (the ratio of the maximum to minimum pairwise distances) in each dimension. Then, the constructed estimators can be used to guide the coordinates selection process, yielding a $(1 + O(\alpha))$-approximation in time $O(md) + \tilde{O}(kd/\alpha)$ [1].

While the proposed sampling-based algorithms can achieve faster running time compared to other learning-augmented $k$-means methods, the improvements come with slight compromises in clustering quality guarantees. To benefit more from the sampling-based strategies, we propose a heuristic algorithm (called Fast-Filtering) to better preserve the clustering quality while maintaining the improved running time. Instead of enumerating all the dimensions of the predicted clusters for coordinates approximation, the Fast-Filtering algorithm directly identifies approximate clustering centers by constructing estimators that can provide accurate clustering cost estimates in sub-linear running time. By filtering out data points that are far from the selected centers, the Fast-Filtering algorithm can obtain high-quality clustering centers while avoiding the loss in clustering quality that arises from integrating coordinate information across all the dimensions.

Table 1 provides a detailed comparison of the results for the learning-augmented $k$-means problem. In the Appendix, we also give a plot (Figure 1 in Appendix) of approximation ratios vs. the error rate $\alpha$. It can be seen from the table that the current best result achieves a $(1 + O(\alpha))$-approximation with $\alpha \in [0, 1/2)$ (Nguyen et al., 2022). Both of our proposed algorithms, Fast-Sampling and Fast-Estimation, achieve linear running time in the data size. Compared to the state-of-the-art results, the Fast-Sampling algorithm can achieve nearly the same theoretical guarantee on clustering quality (i.e., $(1 + O(\alpha))$-approximation) while providing faster running time when the number of clusters $k$ and the dimensionality $d$ are much smaller than data size $m$, which is natural in real-world applications. Furthermore, our Fast-Estimation algorithm provides much faster running time while maintaining clustering quality guarantees comparable to other learning-augmented algorithms.

---

[1]Throughout this paper, we use $\tilde{O}(.)$ notations to suppress $\mathrm{polylog}(m, d)$ factors.

Table 1: Comparison results of learning augmented $k$-means algorithms

| Methods and References | Approximation Ratio | Label Error Range | Time Complexity |
|---|---|---|---|
| Partitioning and Sorting (Ergun et al., 2021) | $1 + 20\alpha$ | $[\frac{10\log m}{\sqrt{m}}, 1/7]$ | $O(md\log m)$ |
| Sorting (Nguyen et al., 2022) | $1 + \frac{\alpha}{1-\alpha} + \frac{4\alpha}{(1-2\alpha)(1-\alpha)}$ | $[0, 1/2)$ | $O(md\log m)$ |
| Fast-Sampling (Ours) | $1 + \frac{\alpha}{1-\alpha} + \frac{4\alpha+\alpha\epsilon}{(1-2\alpha)(1-\alpha)}$ | $[0, 1/2)$ | $O(\epsilon^{-1}md\log(kd))$ |
| Fast-Estimation (Ours) | $1 + \frac{\alpha}{1-\alpha} + \frac{13\alpha-15\alpha^2}{(1-3\alpha-\epsilon)(1-2\alpha-\epsilon)}$ | $(0, 1/3 - \epsilon)$ | $O(md) + \tilde{O}(\epsilon^{-5}kd/\alpha)$ |

## 2 PRELIMINARIES

We use $P \subset \mathbb{R}^d$ and $k$ to denote the given dataset and the number of clusters, respectively. Let $m$ be the data size. For any two points $p, q \in \mathbb{R}^d$, let $\delta(p,q)$ and $\delta^2(p,q)$ denote their distance and squared distance, respectively. Given a point $p \in \mathbb{R}^d$ and a set $C = \{c_1, c_2, ..., c_k\}$ of centers, we also use $\delta(p, C) = \min_{c \in C} \delta(p, c)$ to denote the distance from $p$ to its closest center in $C$. We use $C^* = \{c_1^*, ..., c_k^*\}$ and $\mathcal{P}(C^*) = \{P_1^*, ..., P_k^*\}$ to denote the set of the optimal clustering centers and the optimal clustering partition, respectively. For each $c_i^* \in C^*$, $c_i^*$ can be represented by $d$ coordinates, i.e., $c_i^* = (c_{i1}^*, c_{i2}^*, ..., c_{id}^*)$. We define the clustering cost of $P$ with respect to $C$ as $\delta^2(P, C) = \sum_{x \in P} \delta^2(x, C)$. Given a collection $\mathcal{L}(P) = \{P_1, P_2, ..., P_k\}$ as the predictor, let $Q_i = P_i \cap P_i^*$ be the set of data points in $P_i$ that belong to $P_i^*$. Denote the projections of data points in $P_i$ and $Q_i$ onto the $j$-th dimension as $P_{ij}$ and $Q_{ij}$, respectively. Let $P_{ij}^*$ be the projections of data points in $P_i^*$ onto the $j$-th dimension. Let $m_i$ and $m$ be the size of $P_i$ and $P$, respectively. For a set $V \subset \mathbb{R}^d$ of data points, let $\overline{V}$ be the geometric center for $V$. Denote $P(j)$ as the projections of points in $P$ onto the $j$-th dimension. Let $\Delta_{\max}$ be the maximum aspect ratio of the projected data points, i.e., $\Delta_{\max} = \max_{1 \le j \le d} \frac{\max_{x,y \in P(j)} \delta(x,y)}{\min_{x,y \in P(j), x \ne y} \delta(x,y)}^2$. For a positive integer $t$, let $[t]$ be the set $\{1, 2, ..., t\}$.

**Learning-Augmented $k$-means Problem.** Given a dataset $P \subset \mathbb{R}^d$ of $m$ points, let $C^*$ and $\mathcal{P}(C^*) = \{P_1^*, P_2^*, ..., P_k^*\}$ be an arbitrary optimal solution and its corresponding partition, respectively. In the learning-augmented setting, it is assumed that we have access to a predictor in the form of a labeling partition $\mathcal{L}(P) = \{P_1, P_2, ..., P_k\}$ parameterized by some label error rate $\alpha \in [0, 1)$ such that $|P_i \cap P_i^*| \ge (1-\alpha) \max\{|P_i|, |P_i^*|\}$. The goal of learning-augmented $k$-means clustering is to find a set $C \subset \mathbb{R}^d$ of data points as centers such that $\delta^2(P, C)$ is minimized.

The following lemmas are folklore in $k$-means clustering.

**Lemma 1** (Arthur & Vassilvitskii (2007)) *Given a set $X \subset \mathbb{R}^d$ with size $m$ and an arbitrary data point $c \in \mathbb{R}^d$, it holds that $\delta^2(X, c) = \delta^2(X, \overline{X}) + m \cdot \delta^2(c, \overline{X})$.*

**Lemma 2** (Nguyen et al. (2022)) *Given a set $J \subset \mathbb{R}$, let $J_1 \subseteq J$ be a subset of $J$ with $|J_1| \ge (1 - \zeta)|J|$, where $(0 \le \zeta < 1)$. Then, it holds that $\delta^2(\overline{J}, \overline{J_1}) \le \frac{\zeta}{(1-\zeta)|J|} \delta^2(J, \overline{J})$.*

**Lemma 3** (Nguyen et al. (2022)) *Given a set $X \subset \mathbb{R}^d$ and an $\alpha \in (0, 1]$, let $X' = \arg\min_{X'' \subseteq X, |X''| = \alpha|X|} \delta^2(X'', \overline{X''})$. Then, it holds that $\delta^2(X', \overline{X'}) \le \alpha \cdot \delta^2(X, \overline{X})$.*

## 3 THE FAST-SAMPLING ALGORITHM

The general idea of our Fast-Sampling algorithm is to efficiently approximate the optimal clustering centers in each dimension by identifying high-quality coordinates without using sorting-based strategies. The primary technical challenge lies in handling the false negatives without significantly compromising the approximation guarantees. Although directly sampling a small subset of coordinates from each dimension for the predicted clusters can help identify points near the optimal clustering centers, the uniformly sampled coordinates may not accurately approximate the optimal centers, potentially leading to a constant-factor loss in the approximation guarantees. To address this issue, the Fast-Sampling algorithm first identifies candidate coordinates close to the coordinates of each optimal clustering center within linear running time in the data size. Then, the constructed candidate coordinates are used to define intervals that can precisely capture the positions of the optimal centers, enabling a better approximation through fine-grained divisions of these intervals.

---

[2]Note that it is common to assume that $\Delta_{\max}$ can be bounded a polynomial function of the data size.

The proposed Fast-Sampling algorithm (see Algorithm 1) mainly consists of the following two phases: (1) interval estimation (steps 3-6 of Algorithm 1); (2) candidate coordinates construction (step 7 of Algorithm 1). In the interval estimation phase, for each dimension of the predicted clusters, the interval lengths are estimated through random sampling strategies. The samples are then symmetrically adjusted based on the interval length estimates to construct intervals that can enclose the coordinates of the optimal centers. In the second phase, the derived intervals are further divided into smaller ones, with each corresponding to a new candidate coordinate, enabling a fine-grained approximation of the optimal clustering centers. In the following, we provide a detailed analysis for the proposed algorithm. Due to space limits, all the proofs are delivered to the Appendix.

---

**Algorithm 1** Fast-Sampling

---

**Input:** A $k$-means instance $(P, k, d)$, a set $(P_1, ..., P_k)$ of partitions with error rate $\alpha$, and a parameter $\epsilon \in (0, 1]$.

**Output:** A set $C \subset \mathbb{R}^d$ of centers with $|C| = k$.

1: **for** $i \in [k]$ **do**
2:      **for** $j \in [d]$ **do**
3:          Randomly and independently sample a set $U_{ij}$ from $P_{ij}$ with size $O(\log(kd))$.
4:          **for** $u \in U_{ij}$ **do**
5:              Let $\mathcal{N}_{ij}(u)$ be the set of the nearest $(1 - \alpha)|P_i|$ cooridnates in $P_{ij}$ to $u$.
6:              $l_{ij} = \sqrt{\frac{2\delta^2(\mathcal{N}_{ij}(u), \overline{\mathcal{N}_{ij}(u)})}{(1-\alpha)|P_i|}}$.
7:              $s(u) = \left\{u + \epsilon'\lambda l_{ij} : \lambda \in \left[-\frac{1}{\epsilon'}, \frac{1}{\epsilon'}\right] \cap \mathbb{Z}\right\}$, where $\epsilon' = \sqrt{\frac{\epsilon}{48}}$.
8:          $U'_{ij} = \bigcup_{u \in U_{ij}} s(u)$.
9:          $u_1 = \arg\min_{u \in U'_{ij}} \delta^2(\mathcal{N}_{ij}(u), \overline{\mathcal{N}_{ij}(u)})$.
10:         $I_{ij} = \mathcal{N}_{ij}(u_1)$.
11:      $\hat{c}_i = (\overline{I_{ij}})_{j \in [d]}$.
12: **return** $\{\hat{c}_1, \hat{c}_2, ..., \hat{c}_k\}$.

---

We first consider a single dimension $j \in [d]$ of an arbitrary predicted cluster $P_i$ for some $i \in [k]$. Let $Q'_{ij} \subseteq Q_{ij}$ be the set of the coordinates with size $(1 - \alpha)m_i$ and minimum clustering cost, i.e., $Q'_{ij} = \arg\min_{Q''_{ij} \subseteq Q_{ij}, |Q''_{ij}| = (1-\alpha)m_i} \delta^2(Q''_{ij}, \overline{Q''_{ij}})$. Starting from step 3 of Algorithm 1, a set $U_{ij}$ is constructed by randomly and independently drawing $O(\log(kd))$ samples from $P_{ij}$. The goal here is to find coordinates close to the coordinates of the optimal centers. We will show that, with certain probability, there exists at least one coordinate $u \in U_{ij}$ that can well approximate the geometric center of $Q'_{ij}$. To analyze the success probability, we define $G^\mu_{ij} = \left\{x \in Q'_{ij} : \delta^2(x, \overline{Q'_{ij}}) \leq \mu\delta^2(Q'_{ij}, \overline{Q'_{ij}})/|Q'_{ij}|\right\}$ as the set of coordinates close to $\overline{Q'_{ij}}$, where $\mu$ is a constant with $\mu > 1$. The following lemma argues that $G^\mu_{ij}$ takes a large fraction of $Q'_{ij}$.

**Lemma 4** *For any $Q_{ij} = P^*_{ij} \cap P_{ij}$, it holds that $|G^\mu_{ij}| \geq \frac{\mu - 1}{\mu}|Q'_{ij}|$.*

Let $\mu = 2$. According to Lemma 4, we have $|G^2_{ij}| \geq \frac{1}{2}|Q'_{ij}| = \frac{(1-\alpha)m_i}{2}$. If randomly and independently sampling a set $U_{ij}$ with size $\frac{2}{1-\alpha} \ln(\frac{kd}{\eta})$ from $P_{ij}$, the probability of sampling at least one coordinate $u \in G^2_{ij}$ is $1 - \left(1 - \frac{|G^2_{ij}|}{m_i}\right)^{|U_{ij}|} \geq 1 - e^{|U_{ij}| \ln\left(1 - \frac{|G^2_{ij}|}{m_i}\right)} \geq 1 - e^{-\frac{|U_{ij}||G^2_{ij}|}{m_i}} \geq 1 - \frac{\eta}{kd}$, where the second inequality follows from $\ln(1 - x) \leq -x$ for $x \in (0, 1)$. By taking a union bound success probability over all the dimensions and the predicted clusters, we can argue that with constant probability, there exists at least one coordinate $u \in U_{ij}$ such that $u \in G^2_{ij} \cap U_{ij}$.

**Corollary 1** *With constant probability, for each $i \in [k]$ and $j \in [d]$, there exists at least one coordinate $u \in U_{ij}$ such that $u \in G^2_{ij}$.*

Based on the sampled coordinates, in the remaining steps of the interval estimation phase (steps 4-6 of Algorithm 1), the Fast-Sampling algorihtm estimates the interval lengths to identify potential regions that can enclose the geometric center for $Q'_{ij}$. According to Corollary 1, we can assume that there always exists at least one coordinate $u \in U_{ij} \cap G^2_{ij}$. Then, in step 5 of Algorithm 1, the

algorithm identifies the set $\mathcal{N}_{ij}(u)$ of the nearest $(1-\alpha)m_i$ coordinates in $P_{ij}$ to $u$. The following lemma shows that both lower and upper bounds for $\delta^2(Q'_{ij}, \overline{Q'_{ij}})$ can be established using $\mathcal{N}_{ij}(u)$.

**Lemma 5** *Given an arbitrary coordinate* $u \in G_{ij}^2 \cap U_{ij}$, *it holds that* $\delta^2(Q'_{ij}, \overline{Q'_{ij}}) \le \delta^2(\mathcal{N}_{ij}(u), \overline{\mathcal{N}_{ij}(u)}) \le 3\delta^2(Q'_{ij}, \overline{Q'_{ij}})$.

Lemma 5 shows that, if the sampled point $u \in U_{ij}$ is from $G_{ij}^2$, by identifying the set $\mathcal{N}_{ij}(u)$ of coordinates in $P_{ij}$, we can obtain both lower and upper bounds for $\delta^2(Q'_{ij}, \overline{Q'_{ij}})$. According to the definition of $G_{ij}^2$, we have $\delta^2(u, \overline{Q'_{ij}}) \le 2\delta^2(Q'_{ij}, \overline{Q'_{ij}})/|Q'_{ij}|$. Let $l_{ij} = \sqrt{\frac{2\delta^2(\mathcal{N}_{ij}(u), \overline{\mathcal{N}_{ij}(u)})}{(1-\alpha)m_i}}$ be the interval length. Then, according to Lemma 5, we can get that $\overline{Q'_{ij}} \in [u - l_{ij}, u + l_{ij}]$ and the interval has length at most $2\sqrt{\frac{6\delta^2(Q'_{ij}, \overline{Q'_{ij}})}{(1-\alpha)m_i}}$. In the candidate coordinates construction phase (step 7 of Algorithm 1), the algorithm further divides the intervals into smaller blocks, where the length of each block is parameterized by some $\epsilon' = \sqrt{\frac{\epsilon}{48}}$. Then, it holds trivially that there must exist at least one block $b = [u + \epsilon'\lambda l_{ij}, u + \epsilon'(\lambda+1)l_{ij}]$ for some integer $\lambda \in \mathbb{Z}$, such that $\delta(u + \epsilon'\lambda l_{ij}, \overline{Q'_{ij}}) \le \sqrt{\frac{\epsilon\delta^2(Q'_{ij}, \overline{Q'_{ij}})}{2(1-\alpha)m_i}}$. Consequently, in step 8 of Algorithm 1, we can get that the constructed candidate set $U'_{ij}$ contains at least one coordinate $u' \in U'_{ij}$ such that $\delta(u', \overline{Q'_{ij}}) \le \sqrt{\frac{\epsilon\delta^2(Q'_{ij}, \overline{Q'_{ij}})}{2(1-\alpha)m_i}}$.

**Corollary 2** *With constant probability, for each* $i \in [k]$ *and* $j \in [d]$, *there exists at least one coordinate* $u' \in U'_{ij}$ *such that* $\delta(u', \overline{Q'_{ij}}) \le \sqrt{\frac{\epsilon\delta^2(Q'_{ij}, \overline{Q'_{ij}})}{2(1-\alpha)m_i}}$.

Starting from step 9 in Algorithm 1, the Fast-Sampling algorithm enumerates all the constructed candidate coordinates and their $(1-\alpha)m_i$ nearest neighbors to identify the set of the coordinates with the minimum clustering cost. Then, the geometric center for the set of the coordinates with minimum clustering cost is selected to serve as the coordinate for the clustering center. Let $I_{ij}$ be the set of the coordinates found in step 10 of Algorithm 1. The following lemma shows that the distance between $\overline{Q_{ij}}$ and $\overline{I_{ij}}$ can be bounded using $\overline{I_{ij} \cap Q_{ij}}$ as a bridge.

**Lemma 6** *The following bound holds:* $\delta^2(\overline{I_{ij}}, \overline{Q_{ij}}) \le \frac{(4\alpha + \alpha\epsilon)\delta^2(Q_{ij}, \overline{Q_{ij}})}{|Q_{ij}|(1-2\alpha)}$.

To this end, we can combine Lemma 6 with Lemma 2 to bound the distance between $\overline{I_{ij}}$ and $\overline{P_{ij}^*}$.

**Lemma 7** *The following bound holds:* $\delta^2(\overline{I_{ij}}, \overline{P_{ij}^*}) \le \left( \frac{\alpha}{1-\alpha} + \frac{\alpha(4+\epsilon)}{(1-2\alpha)(1-\alpha)} \right) \frac{\delta^2(P_{ij}^*, \overline{P_{ij}^*})}{|P_{ij}^*|}$.

Putting all these together, we can get the following result for learning-augmented $k$-means problem.

**Theorem 1** *There exists a learning-augmented $k$-means algorithm that can output a $(1 + O(\alpha))$-approximate solution in time* $O(\epsilon^{-1}md\log(kd))$ *with constant probability, where* $\alpha \in [0, 1/2)$.

## 4  THE FAST-ESTIMATION ALGORITHM

Although the proposed Fast-Sampling algorithm achieves linear running time in the data size while maintaining the approximation guarantees, it introduces an additional $O(\log(kd))$ factor loss when taking a union bound success probability. The additional loss in the running time might influence the practical performance of the algorithm when handling large-scale datasets. To address this issue, in this section, we propose a faster sampling-based algorithm called Fast-Estimation. The proposed Fast-Estimation algorithm can efficiently approximate the coordinates of each predicted cluster within linear running time, with only a small trade-off in clustering quality guarantees.

The formal description for the Fast-Estimation algorithm is given in Algorithm 2. The general idea behind the algorithm is to first generate candidate coordinates that can closely approximate the coordinates of the optimal clustering centers. Then, in each dimension of each predicted cluster, an estimator is constructed using uniform sampling strategy. The estimator is designed to provide

accurate clustering cost estimates for subsets of coordinates with sizes $(1 - \alpha)m_i$. In particular, for each dimension of each predicted cluster, the estimator is built by randomly selecting a set $S_{ij}$ from $P_{ij}$. Each sampled coordinate is then assigned a uniform weight, enabling approximate clustering costs to be calculated using the weighted samples rather than the entire predicted cluster. With the constructed estimators, finding the set of coordinates with minimum clustering cost can be done in sub-linear time, removing the multiplicative $O(\log(kd))$ factor from the running time of the Fast-Sampling algorithm. Due to the space limits, all the proofs are delivered to the Appendix.

---

**Algorithm 2** Fast-Estimation

---

**Input:** A $k$-means instance $(P, k, d)$, a set $(P_1, P_2, ..., P_k)$ of partitions with error rate $\alpha$, and a parameter $0 < \epsilon < 0.5$.
**Output:** A set $C \subset \mathbb{R}^d$ of centers with $|C| = k$.
1: **for** $i \in [k]$ **do**
2:     **for** $j \in [d]$ **do**
3:         Randomly and independently sample a set $U_{ij}$ from $P_{ij}$ with size $O(\log(kd))$, then initialize $U'_{ij} = \emptyset$ and $\epsilon_1 = \frac{\epsilon}{126}$.
4:         **for** $q = 0$ to $O(\log(m\Delta_{\max}^2))$ **do**
5:             $l_{ij} = \sqrt{\frac{2^{q-1}}{(1-\alpha)m_i}}$.
6:             **for** $u \in U_{ij}$ **do**
7:                 $s(u) = \left\{ u + \epsilon_2 \lambda l_{ij} : \lambda \in \left[ -\frac{1}{\epsilon_2}, \frac{1}{\epsilon_2} \right] \cap \mathbb{Z} \right\}$, where $\epsilon_2 = \sqrt{\frac{\epsilon_1}{32}}$.
8:                 $U'_{ij} = U'_{ij} \cup s(u)$.
9:         Randomly and independently sample a set $S_{ij}$ from $P_{ij}$ with size $O\left( \frac{\log(m^3 d \log^3(m\Delta_{\max}^2)/\epsilon_1^2) \log(m\Delta_{\max}^2)}{\alpha \epsilon_1^4} \right)$, assign each point in $S_{ij}$ a weight $\frac{m_i}{|S_{ij}|}$.
10:         Construct an estimator $\omega$ such that $\forall u \in U'_{ij}$, $\omega(u) = \sum_{p \in S_{ij} \setminus \mathcal{F}(u)} \frac{m_i}{|S_{ij}|} \delta^2(p, u)$, where $\mathcal{F}(u)$ is the set of the furthest $(1 + 3\epsilon_1)\alpha |S_{ij}|$ points from $S_{ij}$ to $u$.
11:         $c_{ij} = \arg\min_{u \in U'_{ij}} \omega(u)$.
12:         Let $I_{ij}$ be the set of the nearest $(1 - 2\alpha - \alpha\epsilon)m_i$ coordinates from $P_{ij}$ to $c_{ij}$.
13:     $\hat{c}_i = (\overline{I_{ij}})_{j \in [d]}$.
14: **return** $\{\hat{c}_1, \hat{c}_2, ..., \hat{c}_k\}$.

---

In the following, we give the formal analysis for Algorithm 2. In each step 3 of Algorithm 2, for each dimension of the predicted cluster, the algorithm selects a random sample $U_{ij}$ to approximate the coordinates of the optimal clustering centers. According to Lemma 4, with constant probability, there exists at least one sampled coordinate $u \in U_{ij}$ such that $\delta(u, \overline{Q'_{ij}}) \leq \sqrt{2\delta^2(Q'_{ij}, \overline{Q'_{ij}})/|Q'_{ij}|}$. Then, starting from step 4 of Algorithm 2, the algorithm enumerates all the possible lengths of intervals for constructing the set of candidate coordinates. Without loss of generality, we can assume that the minimum pairwise distance between the coordinates in $P_{ij}$ is 1, and the maximum pairwise distance between the coordinates in $P_{ij}$ is $\Delta_{\max}$. Consequently, in step 5 of Algorithm 2, there exists at least one guess for the interval length such that $\sqrt{\frac{2\delta^2(Q'_{ij}, \overline{Q'_{ij}})}{(1-\alpha)m_i}} \leq l_{ij} \leq \sqrt{\frac{4\delta^2(Q'_{ij}, \overline{Q'_{ij}})}{(1-\alpha)m_i}}$. Then, in steps 7-8 of Algorithm 2, according to Lemma 5, there also exists at least one coordinate $u' \in U'_{ij}$ such that $u'$ is close enough to the geometric center for $Q'_{ij}$, i.e., $\delta(u', \overline{Q'_{ij}}) \leq \sqrt{\epsilon_1 \delta^2(Q'_{ij}, \overline{Q'_{ij}})/|Q'_{ij}|}$.

For each $u \in U'_{ij}$, denote $\mathcal{N}_{ij}(u)$ as the set of the nearest $(1 - \alpha)m_i$ coordinates from $P_{ij}$ to $u$. Let $\mathcal{O}(u) = P_{ij} \setminus \mathcal{N}_{ij}(u)$ be the set of the furthest $\alpha m_i$ coordinates from $P_{ij}$ to $u$. Before the construction of the estimator $\omega$ (steps 9-10 of Algorithm 2), we start by dividing $\mathcal{N}_{ij}(u)$ into $\gamma = \frac{(1+\epsilon_1)\log(m\Delta_{\max}^2)}{\epsilon_1}$ blocks. Specifically, for each $u \in U'_{ij}$, $\mathcal{N}_{ij}(u)$ is decomposed into $\gamma$ blocks (denoted as $\mathcal{B}_u^1, \mathcal{B}_u^2, ..., \mathcal{B}_u^\gamma$) based on the distances from the coordinates in $\mathcal{N}_{ij}(u)$ to $u$, where $\mathcal{B}_u^l = \left\{ x \in \mathcal{N}_{ij}(u) : (1 + \epsilon_1)^l \leq \delta^2(x, u) < (1 + \epsilon_1)^{l+1} \right\}$. Then, we further divide the blocks into two groups based on the sizes of the blocks, where $\mathcal{L}(u) = \left\{ \mathcal{B}_u^l : |\mathcal{B}_u^l| \geq \frac{\epsilon_1^2 \alpha m_i}{(1+\epsilon_1)\log(m_i\Delta_{\max}^2)}, l \in [\gamma] \right\}$

and $\mathcal{S}(u) = \left\{ \mathcal{B}_u^1, ..., \mathcal{B}_u^\gamma \right\} \backslash \mathcal{L}(u)$ are the groups of large and small blocks, respectively. Our goal is to well approximate each large block in $\mathcal{L}(u)$ while allowing to ignore the coordinates in small blocks. The following lemma shows that the estimator, constructed by randomly selecting a set $S_{ij}$ from $P_{ij}$ with size $\frac{c \log(m^3 d \log^3(m\Delta_{\max}^2)/\epsilon_1^2) \log(m\Delta_{\max}^2)}{\alpha \epsilon_1^4}$ ($c$ is a large enough constant to be specified) can well approximate each large block and the set $\mathcal{O}(u)$ of coordinates with high probability.

**Lemma 8** *With probability at least* $1 - \frac{\epsilon_1}{m^3 d \log^2(m\Delta_{\max}^2)}$*, we have* $(1 - \epsilon_1)E[|\mathcal{B}_u^l \cap S_{ij}|] \leq |\mathcal{B}_u^l \cap S_{ij}| \leq (1+\epsilon_1)E[|\mathcal{B}_u^l \cap S_{ij}|]$ *holds for each* $\mathcal{B}_u^l \in \mathcal{L}(u)$*, and* $(1-\epsilon_1)E[|\mathcal{O}(u) \cap S_{ij}|] \leq |\mathcal{O}(u) \cap S_{ij}| \leq (1 + \epsilon_1)E[|\mathcal{O}(u) \cap S_{ij}|]$.

For the small blocks in $\mathcal{S}(u)$, denote $\mathcal{J}(u) = \bigcup_{\mathcal{B}_u^l \in \mathcal{S}(u)} \mathcal{B}_u^l$ as the set of the coordinates in small blocks. Given that the number of coordinates in small blocks constitutes only a small fraction of the entire predicted cluster, we will show that the number of coordinates selected from these small blocks through random sampling can be approximately upper bounded by $O(\epsilon \alpha |S_{ij}|)$.

**Lemma 9** *With probability at least* $1 - \frac{\epsilon_1}{m^3 d \log^2(m\Delta_{\max}^2)}$*, it holds that* $|\mathcal{J}(u) \cap S_{ij}| \leq 2\epsilon_1 \alpha |S_{ij}|$.

Next, we prove that the estimator can give accurate estimations for the clustering cost induced by the set of the nearest $(1-\alpha)m_i$ coordinates from some sampled coordinate $u \in U_{ij}'$. For any coordinate $u \in U_{ij}'$, let $\mathcal{F}^\dagger(u)$ be the set of the furthest $(2+20\epsilon_1)\alpha m_i$ coordinates from $P_{ij}$ to $u$. The following lemma establishes the lower and upper bounds for the clustering cost given by the estimator $\omega$.

**Lemma 10** *Given an arbitrary coordinate* $u \in U_{ij}'$*, with high probability, we have* $\delta^2(P_{ij} \backslash \mathcal{F}^\dagger(u))/(1 + 7\epsilon_1) \leq \omega(u) \leq (1 + \epsilon_1)^2 \delta^2(\mathcal{N}_{ij}(u), u)$.

According to Lemma 9 and Lemma 10, for an arbitrary sample $u \in U_{ij}'$, with probability at least $1 - \frac{\epsilon_1}{m^2 d \log^2(m\Delta_{\max}^2)}$, the constructed estimator $\omega$ can give approximate clustering cost estimations such that $\frac{\delta^2(P_{ij} \backslash \mathcal{F}^\dagger(u))}{1+7\epsilon_1} \leq \omega(u) \leq (1 + \epsilon_1)^2 \delta^2(\mathcal{N}_{ij}(u), u)$. Observe that there are at most $O(\epsilon_1^{-1/2} \log(kd) \log(m\Delta_{\max}))$ constructed coordinates in each dimension of each predicted cluster. Then, by taking a union bound success probability over all the dimensions of the predicted clusters, we can get that with constant probability, $\frac{\delta^2(P_{ij} \backslash \mathcal{F}^\dagger(u), u)}{1+7\epsilon_1} \leq \omega(u) \leq (1 + \epsilon_1)^2 \delta^2(\mathcal{N}_{ij}(u), u)$ holds for each $u \in U_{ij}'$ where $i \in [k]$ and $j \in [d]$. Based on the properties of the estimator, we will show that in each dimension of each predicted cluster, the Fast-Estimation algorithm can find good coordinate approximation for the optimal clustering centers.

**Lemma 11** *The following bound holds:* $\delta^2(\overline{I_{ij}}, \overline{Q_{ij}}) \leq O(\alpha)\delta^2(Q_{ij}, \overline{Q_{ij}})/|Q_{ij}|$.

Finally, by using Lemma 7, Theorem 2 can be proved.

**Theorem 2** *There exists a learning-augmented $k$-means algorithm that can output a $(1 + O(\alpha))$-approximate solution in time $O(md) + \tilde{O}(\epsilon^{-5}kd/\alpha)$ with constant probability for $\alpha \in (0, 1/3 - \epsilon)$.*

## 5 THE FAST-FILTERING ALROTIHM

For the Fast-Sampling and Fast-Estimation algorithms, clustering centers are generated by finding coordinates approximation in each dimension. However, the sampling process may introduce cumulative errors, potentially leading to a degradation in the overall clustering quality. In this section, based on our Fast-Sampling and Fast-Estimation algorithms, we provide a more practical heuristic algorithm to better preserve the clustering quality while maintaining the improved running time.

The proposed algorithm is presented in Algorithm 3, where the main idea is to directly find center approximations for each predicted cluster. In step 2 of Algorithm 3, a set of samples is drawn randomly and independently from each predicted cluster to serve as candidate centers. Then in steps 3-4 of Algorithm 3, estimators are constructed using similar ideas from the Fast-Estimation algorithm. Based on the estimators, the candidate center with the minimum clustering cost is selected in step 5 of Algorithm 3 to identify intervals containing the nearest $(1 - \alpha)m_i$ points. Finally, in step 7 of Algorithm 3, the geometric centers of the identified intervals are selected as the final clustering centers. In Appendix A.4, we give a theoretical analysis for the Fast-Filtering algorithm. We show

that, with adjusted number of nearest neighbors (steps 4 and 6 of Algorithm 3) and sample sizes $R_1$ and $R_2$, the Fast-Filtering algorithm can also give a $(1 + O(\sqrt{\alpha}))$-approximate solution.

---

**Algorithm 3** Fast-Filtering

---

**Input:** A $k$-means instance $(P, k, d)$, a set $(P_1, P_2, ..., P_k)$ of partitions with error rate $\alpha$, parameters $R_1 > 0$, $R_2 > 0$ and $0 < \epsilon < 1$.
**Output:** A set $C \subset \mathbb{R}^d$ of centers with $|C| \leq k$.
1: **for** $i \in [k]$ **do**
2:     Randomly and independently sample a set $U_i$ from $P_i$ with size $R_1$.
3:     Randomly and independently sample a set $S_i$ from $P_i$ with size $R_2$ , and assign each point in $S_i$ a weight $\frac{m_i}{|S_i|}$.
4:     Construct an estimator $\omega$ such that $\forall u \in U_i, \omega(u) = \sum_{p \in S_i \setminus \mathcal{F}(u)} \frac{m_i}{|S_i|} \delta^2(p, u)$, where $\mathcal{F}(u)$ is the set of the furthest $(1 + \epsilon)\alpha|S_i|$ points from $S_i$ to $u$.
5:     $c_i = \arg \min_{u \in U_i} \omega(u)$.
6:     Let $I_i$ be the set of the nearest $(1 - \alpha)m_i$ points from $P_i$ to $c_i$.
7:     $\hat{c}_i = \overline{I_i}$.
8: **return** $\{\hat{c}_1, \hat{c}_2, ..., \hat{c}_k\}$.

---

## 6 EXPERIMENTS

In this section, we give empirical evaluations on the performances of our proposed algorithms. All algorithms are implemented and executed in Python. The experiments were done on a machine with i7-12700KF processor and 256GB RAM. Following the prior work (Nguyen et al., 2022; Ergun et al., 2021), we run each algorithm 10 times and report the average results with deviations.

**Datasets.** Following the work in Nguyen et al. (2022) and Ergun et al. (2021), we test the algorithms on datasets CIFAR10 ($m = 10,000$, $d = 3,072$), PHY ($m = 10,000$, $d = 50$) and MNIST ($m = 1,797$, $d = 64$) with varying error rate $\alpha$ and the number of clusters $k$. We also test the performances of the algorithms on other large datasets from UCI Machine Learning Repository [3] including SUSY ($m = 5,000,000$, $d = 18$) and HIGGS ($m = 11,000,000$, $d = 27$), and one large-scale dataset SIFT ($m = 100,000,000$, $d = 128$) from Matsui et al. (2017).

**Algorithms.** In our experiments, we mainly compare our proposed Fast-Sampling, Fast Estimation and Fast-Filtering algorithms (the version in Appendix A.4 with theoretical guarantees) with other learning-augmented algorithms, including the algorithm in Ergun et al. (2021) (denoted as Ergun) and the algorithm in Nguyen et al. (2022) (denoted as Det). For the Fast-Sampling algorithm, the sample size is set to 4, and we fix $\epsilon = 1$. For the Fast-Filtering and Fast-Estimation algorithms, we fix $R_1 = 10$, $R_2 = m/20$ and $\epsilon = 0.3$, where $m$ is the size of the given clustering instance. To further demonstrate the advantage of learning-augmented clustering model, we also give comparisons between our algorithms and the $k$-means++ method (Arthur & Vassilvitskii, 2007) without prediction information.

**Predictor Description.** Following the prior work (Nguyen et al., 2022), the predictor is generated as follows. For each dataset, we first run the $k$-means++ (Arthur & Vassilvitskii, 2007) method as an initialization, and then run the Lloyd's algorithm (Lloyd, 1982) until convergence, where the labels returned are regarded as the optimal labeling partitions (denoted as $\{P_1, ..., P_k\}$). To test the performance of the algorithms under different error rates of the predictor, following the previous work of Nguyen et al. (2022), we randomly change the labels of the $\alpha m_i$ points closest to $c_i$ for each cluster $P_i$ to generate the corrupted labeling partitions $\{P'_1, ..., P'_k\}$ as the predictors. For every dataset, we generate the set of corrupted labels for $\alpha$ ranging from 0.1 to 0.5.

**Detailed Algorithm Implementations.** As pointed out in Nguyen et al. (2022), in most situations, we will not have access to the error rate $\alpha$ and must try different guesses of $\alpha$ to return the clustering with the best cost. Therefore, for each algorithm (including those in Ergun et al. (2021); Nguyen et al. (2022) as well as ours), we iterate over 15 possible values of $\alpha$ uniformly distributed in the interval $[0.01, 0.5]$ as the inputs for the algorithms. The guessed $\alpha$ value that yields the best cluster-

---

[3]https://archive.ics.uci.edu/

ing cost is selected as the final result. For each algorithm, the runtime for each execution includes the cumulative time spent across 15 iterations of guessing error rates and solving the corresponding $k$-means instances based on predictors. Additionally, we also compare the ARI amd NMI values for different algorithms to show the quality of clustering with respect to the ground truth labeling.

Table 2: Comparisons on dataset SIFT for varying $\alpha$ and fixed $k = 20$

| Dataset SIFT (100,000,000 × 128) | | | | | | |
|---|---|---|---|---|---|---|
| Method | Ref | $\alpha$ | Cost | NMI | ARI | Time(s) |
| k-means++ | | | 1.6884E+13±1.45E+11 | 0.3285±0.0138 | 0.1530±0.0137 | **1000.89±10.84** |
| Ergun | | | 9.9799E+12±1.03E+05 | 0.9243±0.0000 | 0.9181±0.0001 | 16748.88±5776.25 |
| Det | | | 9.7791E+12±0.00E+00 | 0.9490±0.0000 | 0.9491±0.0000 | 13152.95±2160.94 |
| Fast-Sampling | 1.0542E+13(844.18s) | 0.1 | 9.7666E+12±0.00E+00 | **0.9519±0.0000** | **0.9531±0.0000** | 13057.36±1717.68 |
| Fast-Filtering | | | **9.7150E+12±2.90E+08** | 0.9316±0.0090 | 0.9333±0.0107 | **1006.31±43.79** |
| Fast-Estimation | | | 9.8007E+12±3.74E+08 | 0.9465±0.0003 | 0.9466±0.0002 | 8874.66±2871.26 |
| k-means++ | | | 1.6585E+13±7.91E+10 | 0.3634±0.0182 | 0.1940±0.0221 | **1077.12±71.57** |
| Ergun | | | 1.0210E+13±2.89E+08 | 0.9043±0.0000 | 0.8901±0.0000 | 17410.75±6132.76 |
| Det | | | 9.9919E+12±0.00E+00 | 0.9019±0.0000 | 0.8867±0.0000 | 13681.80±2073.36 |
| Fast-Sampling | 9.7055E+12(1011.24s) | 0.2 | 9.9576E+12±0.00E+00 | 0.9037±0.0000 | 0.8895±0.0000 | 13270.53±1989.65 |
| Fast-Filtering | | | **9.7914E+12±9.59E+08** | 0.8690±0.0116 | 0.8515±0.0146 | **1088.53±92.09** |
| Fast-Estimation | | | 1.0004E+13±6.51E+08 | **0.9093±0.0002** | **0.8979±0.0002** | 9567.52±2691.74 |
| k-means++ | | | 1.6561E+13±9.48E+10 | 0.3531±0.0278 | 0.1814±0.0206 | **927.07±31.75** |
| Ergun | | | 1.0526E+13±4.08E+07 | 0.8663±0.0000 | 0.8361±0.0000 | 17586.20±6488.30 |
| Det | | | 1.0291E+13±0.00E+00 | 0.8625±0.0000 | 0.8299±0.0000 | 13214.91±1914.86 |
| Fast-Sampling | 9.2478E+12(1330.99s) | 0.3 | 1.0238E+13±0.00E+00 | **0.8743±0.0000** | **0.8496±0.0000** | 13032.26±1657.72 |
| Fast-Filtering | | | **9.9098E+12±7.74E+09** | 0.8180±0.0048 | 0.7833±0.0064 | **1095.48±66.17** |
| Fast-Estimation | | | 1.0300E+13±3.31E+08 | 0.8663±0.0002 | 0.8371±0.0002 | 8618.38±2378.02 |
| k-means++ | | | 1.6814E+13±4.91E+11 | 0.3582±0.0111 | 0.1752±0.0126 | **991.80±148.57** |
| Ergun | | | 1.0924E+13±4.27E+08 | 0.8273±0.0000 | 0.7801±0.0001 | 16291.70±5926.28 |
| Det | | | 1.0683E+13±0.00E+00 | 0.8248±0.0000 | 0.7749±0.0000 | 12999.81±2144.98 |
| Fast-Sampling | 8.9739E+12(1342.73s) | 0.4 | 1.0613E+13±0.00E+00 | **0.8353±0.0000** | **0.7930±0.0000** | 13658.40±1766.14 |
| Fast-Filtering | | | **1.0125E+13±3.14E+09** | 0.7879±0.0048 | 0.7393±0.0032 | **1091.53±94.64** |
| Fast-Estimation | | | 1.0687E+13±8.25E+08 | 0.8260±0.0001 | 0.7781±0.0003 | 8725.94±2691.41 |
| k-means++ | | | 1.7542E+13±2.81E+11 | 0.3313±0.0073 | 0.1580±0.0065 | **972.59±60.40** |
| Ergun | | | 1.1414E+13±4.92E+08 | 0.7885±0.0000 | 0.7140±0.0000 | 17256.11±6160.91 |
| Det | | | 1.1156E+13±0.00E+00 | 0.7863±0.0000 | 0.7105±0.0000 | 13121.68±1901.27 |
| Fast-Sampling | 8.7576E+12(1412.61s) | 0.5 | 1.1089E+13±0.00E+00 | **0.7963±0.0000** | **0.7290±0.0000** | 13042.91±1762.42 |
| Fast-Filtering | | | **1.0504E+13±5.81E+09** | 0.7086±0.0103 | 0.6133±0.0097 | **1051.20±34.37** |
| Fast-Estimation | | | 1.1169E+13±1.68E+09 | 0.7886±0.0005 | 0.7153±0.0012 | 8532.96±2152.19 |

**Results.** Table 2 compares our proposed algorithms with other learning-augmented $k$-means methods on the SIFT dataset for varying error rates and fixed clusters. "Ref" reports clustering costs of optimal labeling partitions and the running time for generating them using the Lloyd's algorithm. Due to space limit, the results for varying clusters and other datasets are given in Appendix A.6.

The results show that our Fast-Sampling algorithm achieves clustering costs comparable to state-of-the-art methods, while Fast-Filtering consistently outperforms other learning-augmented algorithms, with an average 1.5% reduction in clustering cost across all datasets. In terms of running time, Fast-Filtering is significantly faster than other algorithms, especially for large and high-dimensional datasets, achieving at least 3x speedup over current methods. On the SIFT dataset, it is the only method faster than Lloyd's algorithm, where the running time is at least 10 times faster than other methods. For the NMI and ARI values, our algorithms consistently achieve NMI and ARI values above 0.80 across most datasets, with particularly better results on MNIST and SIFT due to their spatial coherence. Meanwhile, the Det algorithm performs better on high-dimensional datasets (SUSY, HIGGS, and PHY), while Ergun's algorithm excels on CIFAR10 with its complex image features.

## 7 CONCLUSION

In this work, we present new sampling-based algorithms with linear running time in the data size for the learning-augmented $k$-means problem. We show experimentally that our algorithm achieves better performances on different datasets compared with other state-of-the-art algorithms. An interesting future direction is how to design algorithms with better approximation ratios while maintaining a linear running time in learning-augmented settings.

## ACKNOWLEDGEMENTS

This work was supported by National Natural Science Foundation of China (62432016, 62172446). This work was also carried out in part using computing resources at the High Performance Computing Center of Central South University.

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

# A APPENDIX

## A.1 PLOT OF APPROXIMATION RATIO V.S. ERROR RATE $\alpha$

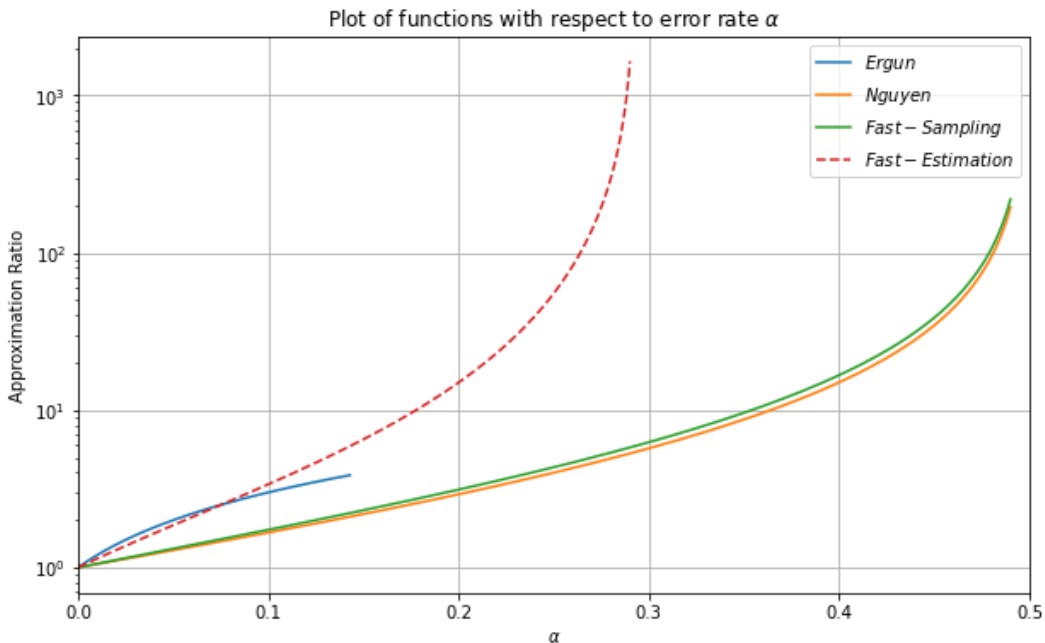

Figure 1: Plot of approximation ratios v.s. the error rate $\alpha$.

## A.2 MISSING PROOFS IN SECTION 3

**Lemma 4.** *For any $Q_{ij} = P_{ij}^* \cap P_{ij}$, it holds that $|G_{ij}^\mu| \geq \frac{\mu-1}{\mu}|Q_{ij}'|$.*

**Proof** Observe that

$$
\begin{aligned}
\delta^2(Q_{ij}', \overline{Q_{ij}'}) &\geq \delta^2(Q_{ij}' \backslash G_{ij}^\mu, \overline{Q_{ij}'}) \\
&\geq |Q_{ij}'| \left(1 - \frac{|G_{ij}^\mu|}{|Q_{ij}'|}\right) \frac{\mu \delta^2(Q_{ij}', \overline{Q_{ij}'})}{|Q_{ij}'|},
\end{aligned}
$$

which implies that $|G_{ij}^\mu| \geq \frac{\mu-1}{\mu}|Q_{ij}'|$. $\qquad\square$

**Lemma 5.** *Given an arbitrary coordinate $u \in G_{ij}^2 \cap U_{ij}$, it holds that $\delta^2(Q_{ij}', \overline{Q_{ij}'}) \leq \delta^2(\mathcal{N}_{ij}(u), \overline{\mathcal{N}_{ij}(u)}) \leq 3\delta^2(Q_{ij}', \overline{Q_{ij}'})$.*

**Proof** According to the definition of $Q_{ij}'$ that $Q_{ij}'$ is the subset of $P_{ij}$ with size $(1 - \alpha)m_i$ and minimum clustering cost, it holds trivially that $\delta^2(\mathcal{N}_{ij}(u), \overline{\mathcal{N}_{ij}(u)}) \geq \delta^2(Q_{ij}', \overline{Q_{ij}'})$ since $\mathcal{N}_{ij}(u) \subseteq P_{ij}$ and $|\mathcal{N}_{ij}(u)| = (1 - \alpha)m_i$. On the other hand, we can establish an upper bound for $\delta^2(\mathcal{N}_{ij}(u), \overline{\mathcal{N}_{ij}(u)})$ as

$$
\begin{aligned}
\delta^2(\mathcal{N}_{ij}(u), \overline{\mathcal{N}_{ij}(u)}) &\leq \delta^2(\mathcal{N}_{ij}(u), u) \\
&\leq \delta^2(Q_{ij}', u) \\
&= \delta^2(Q_{ij}', \overline{Q_{ij}'}) + |Q_{ij}'|\delta^2(u, \overline{Q_{ij}'}) \\
&\leq 3\delta^2(Q_{ij}', \overline{Q_{ij}'}),
\end{aligned}
$$

where the first step follows from the optimality of the geometric center, the second step follows from the definition of $\mathcal{N}_{ij}(u)$ that $\mathcal{N}_{ij}(u)$ contains the nearest $(1-\alpha)m_i$ coordinates from $P_{ij}$ to $u$, the third step follows from Lemma 1, and the last step follows from the definition of $G_{ij}^2$. $\qquad\square$

**Lemma 6.** *The following bound holds:* $\delta^2(\overline{I_{ij}}, \overline{Q_{ij}}) \leq \frac{(4\alpha+\alpha\epsilon)\delta^2(Q_{ij},\overline{Q_{ij}})}{|Q_{ij}|(1-2\alpha)}.$

**Proof** We first bound the distance between $\overline{I_{ij} \cap Q_{ij}}$ and $\overline{I_{ij}}$. Denote $I'_{ij}$ as the set of the nearest $(1-\alpha)m_i$ points from $P_{ij}$ to $u'$ such that $u' \in U'_{ij}$ and $\delta(u', \overline{Q'_{ij}}) \leq \sqrt{\frac{\epsilon\delta^2(Q'_{ij},\overline{Q'_{ij}})}{2(1-\alpha)m_i}}$. Then, it holds that

$$
\begin{aligned}
\delta^2(I'_{ij}, \overline{I'_{ij}}) &\leq \delta^2(I'_{ij}, u') \\
&\leq \delta^2(Q'_{ij}, u') \\
&= \delta^2(Q'_{ij}, \overline{Q'_{ij}}) + |Q'_{ij}|\delta^2(u', \overline{Q'_{ij}}) \\
&\leq \left(1 + \frac{\epsilon}{2}\right)\delta^2(Q'_{ij}, \overline{Q'_{ij}}),
\end{aligned}
$$

where the first step follows from the optimality of geometric center, the second step follows from the definition for $I'_{ij}$ that $I'_{ij}$ contains the nearest $(1-\alpha)m_i$ coordinates from $P_{ij}$ to $u'$, the third step follows from Lemma 1, and the last step follows from $\delta(u', \overline{Q'_{ij}}) \leq \sqrt{\frac{\epsilon\delta^2(Q'_{ij},\overline{Q'_{ij}})}{2(1-\alpha)m_i}}$. Since $I_{ij}$ is the set of coordinates with minimum clustering cost found in step 10 of Algorithm 1, it holds that

$$
\begin{aligned}
\delta^2(I_{ij}, \overline{I_{ij}}) &\leq \delta^2(I'_{ij}, \overline{I'_{ij}}) \\
&\leq \left(1 + \frac{\epsilon}{2}\right)\delta^2(Q'_{ij}, \overline{Q'_{ij}}).
\end{aligned}
$$

Let $\zeta = |Q_{ij} \cap I_{ij}|/|I_{ij}| = 1 - (1 - |Q_{ij} \cap I_{ij}|/|I_{ij}|)$. According to Lemma 2, by assigning $J = I_{ij}$ and $J_1 = Q_{ij} \cap I_{ij}$, we can get that

$$
\begin{aligned}
\delta^2(\overline{I_{ij} \cap Q_{ij}}, \overline{I_{ij}}) &\leq \frac{1-\zeta}{\zeta} \cdot \frac{\delta^2(I_{ij}, \overline{I_{ij}})}{|I_{ij}|} \\
&= \frac{1 - |Q_{ij} \cap I_{ij}|/|I_{ij}|}{|Q_{ij} \cap I_{ij}|/|I_{ij}|} \cdot \frac{\delta^2(I_{ij}, \overline{I_{ij}})}{|I_{ij}|} \\
&= \frac{|I_{ij}| - |Q_{ij} \cap I_{ij}|}{|Q_{ij} \cap I_{ij}|} \cdot \frac{\delta^2(I_{ij}, \overline{I_{ij}})}{|I_{ij}|},
\end{aligned}
$$

where the first step follows from Lemma 2. According to the definition of $Q_{ij}$, since $|P_{ij}\backslash Q_{ij}| \leq \alpha m_i$, it holds that $|Q_{ij} \cap I_{ij}| \geq |I_{ij}| - \alpha m_i$. Then, we have

$$
\begin{aligned}
\delta^2(\overline{I_{ij} \cap Q_{ij}}, \overline{I_{ij}}) &\leq \frac{\alpha}{1-2\alpha} \frac{\delta^2(I_{ij}, \overline{I_{ij}})}{|I_{ij}|} \\
&\leq \frac{\alpha}{1-2\alpha} \cdot \frac{(1+\frac{\epsilon}{2})\delta^2(Q'_{ij}, \overline{Q'_{ij}})}{|I_{ij}|} \\
&\leq \frac{\alpha + 0.5\alpha\epsilon}{1-2\alpha} \cdot \frac{|Q'_{ij}|}{|Q_{ij}| \cdot |I_{ij}|} \cdot \delta^2(Q_{ij}, \overline{Q_{ij}}) \\
&\leq \frac{\alpha + 0.5\alpha\epsilon}{1-2\alpha} \cdot \frac{\delta^2(Q_{ij}, \overline{Q_{ij}})}{|Q_{ij}|},
\end{aligned}
$$

where the first inequality follows from Lemma 2, the second inequality follows from $\delta^2(I_{ij}, \overline{I_{ij}}) \leq (1 + \frac{\epsilon}{2})\delta^2(Q'_{ij}, \overline{Q'_{ij}})$, the third inequality follows from Lemma 3, and the last inequality follows from $|Q'_{ij}| = |I_{ij}| = (1-\alpha)m_i$.

Next, we give an upper bound for the distance between $\overline{I_{ij} \cap Q_{ij}}$ and $\overline{Q_{ij}}$. Let $\zeta' = (1 - |Q_{ij}\backslash I_{ij}|/|Q_{ij}|)$. According to Lemma 2, by assigning $J = Q_{ij}$ and $J_1 = Q_{ij} \cap I_{ij}$, we can get that

$$
\begin{aligned}
\delta^2(\overline{I_{ij} \cap Q_{ij}}, \overline{Q_{ij}}) &\leq \frac{(|Q_{ij}\backslash I_{ij}|/|Q_{ij}|)\delta^2(Q_{ij}, \overline{Q_{ij}})}{(1 - |Q_{ij}\backslash I_{ij}|/|Q_{ij}|)|Q_{ij}|} \\
&= \frac{|Q_{ij}\backslash I_{ij}|\delta^2(Q_{ij}, \overline{Q_{ij}})}{(|Q_{ij}| - |Q_{ij}\backslash I_{ij}|)|Q_{ij}|} \\
&\leq \frac{\alpha\delta^2(Q_{ij}, \overline{Q_{ij}})}{(1 - 2\alpha)|Q_{ij}|},
\end{aligned}
$$

where the last inequality follows from $|Q_{ij}\backslash I_{ij}| \leq \alpha m_i$. Putting all these together and using the triangle inequality, we have $\delta^2(\overline{I_{ij}}, \overline{Q_{ij}}) \leq \frac{(\alpha(4+\epsilon))\delta^2(Q_{ij}, \overline{Q_{ij}})}{|Q_{ij}|(1-2\alpha)}$. $\qquad\square$

**Lemma 7.** *The following bound holds:* $\delta^2(\overline{I_{ij}}, \overline{P_{ij}^*}) \leq \left(\frac{\alpha}{1-\alpha} + \frac{\alpha(4+\epsilon)}{(1-2\alpha)(1-\alpha)}\right) \frac{\delta^2(P_{ij}^*, \overline{P_{ij}^*})}{|P_{ij}^*|}.$

**Proof** Since $Q_{ij} \subseteq P_{ij}^*$, we have

$$
|P_{ij}^*|\overline{P_{ij}^*} = |P_{ij}^*\backslash Q_{ij}|\overline{P_{ij}^*\backslash Q_{ij}} + |Q_{ij}|\overline{Q_{ij}}.
$$

Then, we can get that $\delta^2(\overline{P_{ij}^*}, \overline{P_{ij}^*\backslash Q_{ij}}) = \left(\frac{|P_{ij}^*| - |P_{ij}^*\backslash Q_{ij}|}{|P_{ij}^*\backslash Q_{ij}|}\right)^2 \delta^2(\overline{P_{ij}^*}, \overline{Q_{ij}})$. Define $\gamma = \frac{|P_{ij}^*\backslash Q_{ij}|}{|P_{ij}^*|}$. We have $\delta^2(\overline{P_{ij}^*}, \overline{P_{ij}^*\backslash Q_{ij}}) = \left(\frac{1-\gamma}{\gamma}\right)^2 \delta^2(\overline{P_{ij}^*}, \overline{Q_{ij}})$. By decomposing the clustering cost of $P_{ij}^*$ with respect to $\overline{P_{ij}^*}$ into $\delta^2(P_{ij}^*\backslash Q_{ij}, \overline{P_{ij}^*})$ and $\delta^2(P_{ij}^* \cap Q_{ij}, \overline{P_{ij}^*})$, it holds that

$$
\begin{aligned}
\delta^2(P_{ij}^*, \overline{P_{ij}^*}) &= \delta^2(P_{ij}^*\backslash Q_{ij}, \overline{P_{ij}^*}) + \delta^2(Q_{ij}, \overline{P_{ij}^*}) \\
&= \delta^2(P_{ij}^*\backslash Q_{ij}, \overline{P_{ij}^*\backslash Q_{ij}}) + \gamma|P_{ij}^*|\delta^2(\overline{P_{ij}^*}, \overline{P_{ij}^*\backslash Q_{ij}}) + \delta^2(Q_{ij}, \overline{P_{ij}^*}) \\
&= \delta^2(P_{ij}^*\backslash Q_{ij}, \overline{P_{ij}^*\backslash Q_{ij}}) + \gamma|P_{ij}^*|\left(\frac{1-\gamma}{\gamma}\right)^2 \delta^2(\overline{P_{ij}^*}, \overline{Q_{ij}}) \\
&\quad + \delta^2(Q_{ij}, \overline{Q_{ij}}) + (1-\gamma)|P_{ij}^*|\delta^2(\overline{Q_{ij}}, \overline{P_{ij}^*}) \\
&= \delta^2(P_{ij}^*\backslash Q_{ij}, \overline{P_{ij}^*\backslash Q_{ij}}) + \delta^2(Q_{ij}, \overline{Q_{ij}}) + \frac{1-\gamma}{\gamma}|P_{ij}^*|\delta^2(\overline{Q_{ij}}, \overline{P_{ij}^*}) \\
&\geq \delta^2(Q_{ij}, \overline{Q_{ij}}) + \frac{1-\alpha}{\alpha}|P_{ij}^*|\delta^2(\overline{Q_{ij}}, \overline{P_{ij}^*}),
\end{aligned}
$$

where the first step follows from Lemma 1, and the last step follows from the definition of the predicted clusters that $\gamma \leq \alpha$. By Lemma 6, we have

$$
\delta^2(P_{ij}^*, \overline{P_{ij}^*}) \geq \frac{1-2\alpha}{\alpha(4+\epsilon)}(1-\alpha)|P_{ij}^*|\delta^2(\overline{I_{ij}}, \overline{Q_{ij}}) + \frac{1-\alpha}{\alpha}|P_{ij}^*|\delta^2(\overline{Q_{ij}}, \overline{P_{ij}^*}).
$$

Then, by using Cauchy-Schwarz Inequality, we have

$$
(\delta(\overline{P_{ij}^*}, \overline{Q_{ij}}) + \delta(\overline{Q_{ij}}, \overline{I_{ij}}))^2 \leq \frac{\delta^2(P_{ij}^*, \overline{P_{ij}^*})}{|P_{ij}^*|}\left(\frac{\alpha}{1-\alpha} + \frac{\alpha(4+\epsilon)}{(1-2\alpha)(1-\alpha)}\right).
$$

Finally, we can conclude the proof using the fact that $\delta^2(\overline{P_{ij}^*}, \overline{I_{ij}}) \leq (\delta(\overline{P_{ij}^*}, \overline{Q_{ij}}) + \delta(\overline{I_{ij}}, \overline{Q_{ij}}))^2$. $\square$

**Theorem 1.** *There exists a learning-augmented $k$-means algorithm that can output a $(1 + O(\alpha))$-approximate solution in time $O(\epsilon^{-1}md\log(kd))$ with constant probability, where $\alpha \in [0, 1/2)$.*

**Proof** Denote $C$ as the set of centers returned by Algorithm 1, where $C = \{\hat{c}_1, \hat{c}_2, ..., \hat{c}_k\}$ and $\hat{c}_i$ is consisting of $d$ coordinates $[c_{i1}, c_{i2}, ..., c_{id}]$. Then, we have

$$
\begin{aligned}
\delta^2(P, C) &\leq \sum_{i=1}^{k} \sum_{j=1}^{d} \delta^2(P_{ij}^*, c_{ij}) \\
&= \sum_{i=1}^{k} \sum_{j=1}^{d} \delta^2(P_{ij}^*, \overline{P_{ij}^*}) + |P_{ij}^*| \delta^2(\overline{P_{ij}^*}, c_{ij}) \\
&\leq \sum_{i=1}^{k} \sum_{j=1}^{d} \left(1 + \frac{\alpha}{1 - \alpha} + \frac{\alpha(4 + \epsilon)}{(1 - 2\alpha)(1 - \alpha)}\right) \delta^2(P_{ij}, \overline{P_{ij}^*}) \\
&= \left(1 + \frac{\alpha}{1 - \alpha} + \frac{\alpha(4 + \epsilon)}{(1 - 2\alpha)(1 - \alpha)}\right) \delta^2(P, C^*).
\end{aligned}
$$

Then, we give the runtime analysis for the proposed Fast-Sampling algorithm. In each step 3 of the Fast-Sampling algorithm, sampling coordinates from the predicted cluster takes $O(1)$ time. In each step 5 of the Fast-Sampling algorithm, finding the nearest $(1 - \alpha)m_i$ coordinates in $P_{ij}$ to a given sample can be executed in linear time $O(m_i)$ using linear selection methods (Blum et al., 1973). In each step 7 of Algorithm 1, note that the interval length can be bounded by $O(1)\sqrt{\frac{\delta^2(Q_{ij}, \overline{Q_{ij}})}{(1 - \alpha)m_i}}$. Hence, by dividing the interval into smaller blocks with length $\epsilon' l_{ij}$ where $\epsilon' = \sqrt{\frac{\epsilon}{48}}$, we can get that the number of candidate coordinates constructed in each step 7 of Algorithm 1 can be bounded by $O(\epsilon^{-1})$. According to the sample size of $U_{ij}$, the total number of coordinates constructed in step 8 of Algorithm 1 can be bounded by $O(\epsilon^{-1} \log(kd))$. Then, finding the subset of coordinates with minimum clustering cost in steps 9-10 of Algorithm 1 can be executed in time $O(\epsilon^{-1} m_i \log(kd))$ using linear selection methods (Blum et al., 1973). Since the Fast-Sampling algorithm needs to enumerate all the dimensions of each predicted cluster, the overall running time can be bounded by $\sum_{i=1}^{k} \sum_{j=1}^{d} O(\epsilon^{-1} m_i \log(kd)) = O(\epsilon^{-1} md \log(kd))$. □

## A.3 MISSING PROOFS IN SECTION 4

**Lemma 8.** *With probability at least* $1 - \frac{\epsilon_1}{m^3 d \log^2(m\Delta_{\max}^2)}$, *we have* $(1 - \epsilon_1)E[|\mathcal{B}_u^l \cap S_{ij}|] \leq |\mathcal{B}_u^l \cap S_{ij}| \leq (1 + \epsilon_1)E[|\mathcal{B}_u^l \cap S_{ij}|]$ *holds for each* $\mathcal{B}_u^l \in \mathcal{L}(u)$, *and* $(1 - \epsilon_1)E[|\mathcal{O}(u) \cap S_{ij}|] \leq |\mathcal{O}(u) \cap S_{ij}| \leq (1 + \epsilon_1)E[|\mathcal{O}(u) \cap S_{ij}|]$.

**Proof** Observe that $|S_{ij}| = \frac{c \log(m^3 d \log^3(m\Delta_{\max}^2)/\epsilon_1^2) \log(m\Delta_{\max}^2)}{\alpha \epsilon_1^4}$ where $c$ is some large enough constant. Hence, for an arbitrary large block $\mathcal{B}_u^l \in \mathcal{L}(u)$, we have

$$
\begin{aligned}
E[|S_{ij} \cap \mathcal{B}_u^l|] &= \frac{|S_{ij}| \cdot |\mathcal{B}_u^l|}{|P_{ij}|} \\
&\geq \frac{\epsilon_1^2 \alpha}{(1 + \epsilon_1) \log(m_i \Delta_{\max}^2)} \cdot \frac{c \log(m^3 d \log^3(m\Delta_{\max}^2)/\epsilon_1^2) \log(m\Delta_{\max}^2)}{\alpha \epsilon_1^4} \\
&\geq \frac{c \log(m^3 d \log^3(m\Delta_{\max}^2)/\epsilon_1^2)}{(1 + \epsilon_1)\epsilon_1^2}.
\end{aligned}
$$

Thus, by applying the Chernoff Bound, we can get that

$$
\begin{aligned}
P_r\left((1 - \epsilon_1)E[|\mathcal{B}_u^l \cap S_{ij}|] \leq |\mathcal{B}_u^l \cap S_{ij}| \leq (1 + \epsilon_1)E[|\mathcal{B}_u^l \cap S_{ij}|]\right) &\geq 1 - 2e^{-\frac{\epsilon_1^2 E[|\mathcal{B}_u^l \cap S_{ij}|]}{3}} \\
&\geq \Omega\left(1 - \frac{\epsilon_1^2}{m^3 d \log^3(m\Delta_{\max}^2)}\right),
\end{aligned}
$$

where the last inequality follows from the fact that $c$ is a large enough constant.

Similarly, for the set $\mathcal{O}(u)$ of coordinates, we can get that with probability at least $\Omega\left(1 - \frac{\epsilon_1^2}{m^3 d \log^3(m\Delta_{\max}^2)}\right)$, it holds that $(1 - \epsilon_1)E[|\mathcal{O}(u) \cap S_{ij}|] \leq |\mathcal{O}(u) \cap S_{ij}| \leq (1 + \epsilon_1)E[|\mathcal{O}(u) \cap S_{ij}|]$. By taking a union bound success probability, Lemma 8 can be proved. □

**Lemma 9.** *With probability at least* $1 - \frac{\epsilon_1}{m^3 d \log^2(m\Delta_{\max}^2)}$, *it holds that* $|\mathcal{J}(u) \cap S_{ij}| \leq 2\epsilon_1 \alpha |S_{ij}|$.

**Proof** According to the definition of small blocks, it holds that $|\mathcal{J}(u)| \leq \epsilon_1 \alpha m_i$. Observe that $E[|S_{ij} \cap \mathcal{J}(u)|] = \frac{|S_{ij}| \cdot |\mathcal{J}(u)|}{m_i}$. Let $\lambda' = \frac{2\epsilon_1 \alpha m_i}{|\mathcal{J}(u)|} - 1$. By applying the Chernoff Bound, we can get that

$$
\begin{aligned}
P_r(|\mathcal{J}(u) \cap S_{ij}| \leq (1+\lambda')E[|\mathcal{J}(u) \cap S_{ij}|]) &\geq 1 - e^{-\frac{E[|S_{ij} \cap \mathcal{J}(u)|](\lambda')^2}{3}} \\
&= 1 - e^{-\frac{|\mathcal{J}(u)| \cdot |S_{ij}|}{3m_i}\left(\frac{2\epsilon_1 \alpha m_i}{|\mathcal{J}(u)|} - 1\right)^2} \\
&\geq 1 - e^{-\frac{|\mathcal{J}(u)| \cdot |S_{ij}|}{3m_i}\left(\frac{\epsilon_1 \alpha m_i}{|\mathcal{J}(u)|}\right)^2} \\
&\geq 1 - e^{-\frac{\epsilon_1^2 \alpha^2 m_i |S_{ij}|}{3|\mathcal{J}(u)|}} \\
&\geq \Omega\left(1 - \frac{\epsilon_1}{m^3 d \log^2(m\Delta_{\max}^2)}\right),
\end{aligned}
$$

where the first step follows from Chernoff Bound, the second step follows from the definition of $\lambda'$, the third step follows from the fact that $|\mathcal{J}(u)| \leq \epsilon_1 \alpha m_i$, and the last step follows from $|S_{ij}| \geq \frac{c \log(m^3 d \log^3(m\Delta_{\max}^2)/\epsilon_1^2) \log(m\Delta_{\max}^2)}{\epsilon_1^4 \alpha}$. Since $(1+\lambda')E[|\mathcal{J}(u) \cap S_{ij}|] = (1+\lambda')\frac{|\mathcal{J}(u)| \cdot |S_{ij}|}{m_i} \leq 2\epsilon_1 \alpha |S_{ij}|$, we have $|\mathcal{J}(u) \cap S_{ij}| \leq 2\epsilon_1 \alpha |S_{ij}|$. $\square$

**Lemma 10.** *Given an arbitrary coordinate* $u \in U'_{ij}$, *with high probability, we have* $\delta^2(P_{ij} \backslash \mathcal{F}^\dagger(u))/(1+7\epsilon_1) \leq \omega(u) \leq (1+\epsilon_1)^2 \delta^2(\mathcal{N}_{ij}(u), u)$.

**Proof** According to Lemma 8 and Lemma 9, by taking a union bound over the success probability, we have $|\mathcal{J}(u) \cap S_{ij}| \leq 2\epsilon_1 \alpha |S_{ij}|$ and $|\mathcal{O}(u) \cap S_{ij}| \leq (1+\epsilon_1)\alpha |S_{ij}|$ hold with probability at least $\left(1 - \frac{\epsilon_1}{m^2 d \log^2(m\Delta_{\max}^2)}\right)$. Define $\mathcal{F}'(u) = (\mathcal{J}(u) \cup \mathcal{O}(u)) \cap \mathcal{S}_{ij}$. Then, it holds that $|\mathcal{F}'(u)| \leq (1+3\epsilon_1)\alpha |\mathcal{S}_{ij}|$. For each large block $\mathcal{B}_u^l \in \mathcal{L}(u)$, by using Lemma 8, we have $|\mathcal{B}_u^l \cap S_{ij}| \leq (1+\epsilon_1)E[|\mathcal{B}_u^l \cap S_{ij}|] = \frac{(1+\epsilon_1)|S_{ij}|}{m_i}|\mathcal{B}_u^l|$. Thus, the clustering cost of $\mathcal{B}_u^l \cap S_{ij}$ can be upper bounded by

$$
\begin{aligned}
\delta^2(\mathcal{B}_u^l \cap S_{ij}, u) &= \sum_{x \in \mathcal{B}_u^l \cap S_{ij}} \delta^2(x, u) \\
&\leq (1+\epsilon_1)^{l+1}|\mathcal{B}_u^l \cap S_{ij}| \\
&\leq (1+\epsilon_1)^l \cdot \frac{(1+\epsilon_1)^2 |S_{ij}|}{m_i}|\mathcal{B}_u^l| \\
&\leq \frac{(1+\epsilon_1)^2 |S_{ij}|}{m_i}\delta^2(\mathcal{B}_u^l, u),
\end{aligned}
$$

where the second step follows from the definition of $\mathcal{B}_u^l \in \mathcal{L}(u)$ that $\delta^2(x, u) \leq (1+\epsilon_1)^{l+1}$ holds for each $x \in \mathcal{B}_u^l$, and the fourth step follows from the definition of $\mathcal{B}_u^l \in \mathcal{L}(u)$ that $\delta^2(x, u) \geq (1+\epsilon_1)^l$ holds for each $x \in \mathcal{B}_u^l$. Consequently, by taking a summation over all the coordinates sampled in large blocks, we can get that $\sum_{\mathcal{B}_u^l \in \mathcal{L}(u)} \delta^2(\mathcal{B}_u^l \cap S_{ij}, u) \leq \frac{(1+\epsilon_1)^2}{m_i}|S_{ij}|\delta^2(\mathcal{N}_{ij}(u), u)$. Denote $\mathcal{F}(u)$ as the set of the furthest $(1+3\epsilon_1)\alpha |S_{ij}|$ points from $S_{ij}$ to $u$. According to the definition of the constructed estimator $\omega$ in step 10 of Algorithm 2, we have

$$
\begin{aligned}
\omega(u) &= \frac{m_i}{|S_{ij}|}\delta^2(S_{ij} \backslash \mathcal{F}(u), u) \\
&\leq \frac{m_i}{|S_{ij}|}\delta^2(S_{ij} \backslash \mathcal{F}'(u), u) \\
&= \frac{m_i}{|S_{ij}|}\sum_{\mathcal{B}_u^l \in \mathcal{L}(u)} \delta^2(\mathcal{B}_u^l \cap S_{ij}, u) \\
&\leq \frac{m_i}{|S_{ij}|}\frac{(1+\epsilon_1)^2 |S_{ij}|}{m_i}\delta^2(\mathcal{N}_{ij}(u), u) \\
&= (1+\epsilon_1)^2 \delta^2(\mathcal{N}_{ij}(u), u),
\end{aligned}
$$

where the second step follows from $|\mathcal{F}'(u)| \leq (1 + 3\epsilon_1)\alpha|S_{ij}| = |\mathcal{F}(u)|$. Hence, an upper bound for the clustering cost estimation can be obtained as $\omega(u) \leq (1 + \epsilon_1)^2\delta^2(\mathcal{N}_{ij}(u), u)$.

Then, we show that the estimator can also give a lower bound for the clustering cost induced by the nearest $(1 - \alpha)m_i$ coordinates from some sampled coordinate $u \in U'_{ij}$. For each large block $\mathcal{B}^l_u \in \mathcal{L}(u)$, define $\mathcal{Z}^l_u = \mathcal{F}(u) \cap \mathcal{B}^l_u$. Let $\mathcal{H}^l_u$ be an arbitrary subset of the coordinates in $\mathcal{B}^l_u$ with size $(1 + 3\epsilon_1)|\mathcal{Z}^l_u|\frac{m_i}{|S_{ij}|}$. Denote $\mathcal{F}''(u) = (\mathcal{O}(u) \cup \mathcal{J}(u)) \cup (\bigcup_{\mathcal{B}^l_u \in \mathcal{L}(u)} \mathcal{H}^l_u)$. It holds that

$$
\begin{aligned}
|\mathcal{F}''(u)| &\leq \alpha m_i + 2\epsilon_1\alpha m_i + (1 + 3\epsilon_1)(m_i/|S_{ij}|)|\mathcal{Z}^l_u| \\
&\leq \alpha m_i + 2\epsilon_1\alpha m_i + (1 + 3\epsilon_1)(m_i/|S_{ij}|) \cdot ((1 + 3\epsilon_1)\alpha|S_{ij}|) \\
&\leq (2 + 20\epsilon_1)\alpha m_i,
\end{aligned}
$$

where the second step follows from $\sum_{\mathcal{B}^l_u \in \mathcal{L}(u)} |\mathcal{Z}^l_u| \leq |\mathcal{F}(u)| \leq (1 + 3\epsilon_1)\alpha|S_{ij}|$. Define $\mathcal{F}^\dagger(u)$ as the set of the furthest $(2 + 20\epsilon_1)\alpha m_i$ points from $P_{ij}$ to $u$. Then, we can get that

$$
\begin{aligned}
\delta^2(P_{ij}\backslash\mathcal{F}^\dagger(u), u) &\leq \delta^2(P_{ij}\backslash\mathcal{F}''(u), u) \\
&= \sum_{\mathcal{B}^l_u \in \mathcal{L}(u)} \delta^2(\mathcal{B}^l_u\backslash\mathcal{H}^l_u, u) \\
&\leq \sum_{\mathcal{B}^l_u \in \mathcal{L}(u)} (1 + \epsilon_1)^{l+1}(|\mathcal{B}^l_u| - |\mathcal{H}^l_u|) \\
&\leq \sum_{\mathcal{B}^l_u \in \mathcal{L}(u)} (1 + \epsilon_1) \left( \frac{|\mathcal{B}^l_u \cap S_{ij}|m_i}{(1 - \epsilon_1)|S_{ij}|} - \frac{(1 + 3\epsilon_1)|\mathcal{Z}^l_u|m_i}{|S_{ij}|} \right) \cdot (1 + \epsilon_1)^l \\
&\leq \frac{(1 + \epsilon_1)(1 + 3\epsilon_1)m_i}{|S_{ij}|} \sum_{\mathcal{B}^l_u \in \mathcal{L}(u)} (|\mathcal{B}^l_u \cap S_{ij}| - |\mathcal{Z}^l_u|) \cdot (1 + \epsilon_1)^l \\
&\leq (1 + 7\epsilon_1)\omega(u),
\end{aligned}
$$

where the first step follows from the definition of $\mathcal{F}^\dagger(u)$ that $|\mathcal{F}''(u)| \leq |\mathcal{F}^\dagger(u)|$, the third step follows from the definition of $\mathcal{L}(u)$ that $\delta^2(x, u) \leq (1 + \epsilon_1)^{l+1}$ holds for each $x \in \mathcal{B}^l_u$, the fourth step follows from Lemma 8, and the second to the last step follows from $\frac{1}{1-\epsilon_1} \leq 1 + 3\epsilon_1$ for $0 < \epsilon_1 < 0.5$. □

**Lemma 11.** *The following bound holds:* $\delta^2(\overline{I_{ij}}, \overline{Q_{ij}}) \leq O(\alpha)\delta^2(Q_{ij}, \overline{Q_{ij}})/|Q_{ij}|$.

**Proof** According to Lemma 5, with constant probability, there exists at least one coordinate $u_1 \in U'_{ij}$ such that $\delta^2(u_1, \overline{Q'_{ij}}) \leq \frac{\epsilon_1\delta^2(Q'_{ij}, \overline{Q'_{ij}})}{|Q'_{ij}|}$. Hence, we can get that $\delta^2(\mathcal{N}_{ij}(u_1), u_1) \leq \delta^2(Q'_{ij}, u_1) \leq (1 + \epsilon_1)\delta^2(Q'_{ij}, \overline{Q'_{ij}})$, where the last step follows from Lemma 1. Let $c_{ij}$ be the coordinate chosen by the estimator in step 11 of Algorithm 2. According to the property of the estimator, we have

$$
\begin{aligned}
\frac{\delta^2(P_{ij}\backslash\mathcal{F}^\dagger(c_{ij}), c_{ij})}{1 + 7\epsilon_1} &\leq \omega(c_{ij}) \\
&\leq \omega(u_1) \\
&\leq (1 + \epsilon_1)^2\delta^2(\mathcal{N}_{ij}(u_1), u_1) \\
&\leq (1 + \epsilon_1)^3\delta^2(Q'_{ij}, \overline{Q'_{ij}}).
\end{aligned}
$$

Denote $I_{ij}$ as the set of coordinates found in step 12 of Algorithm 2. Then, we can give the bound between $\overline{I_{ij}}$ and $\overline{Q_{ij}}$. Since $\epsilon_1 = \frac{\epsilon}{126}$, according to Lemma 1 and the properties of the estimator, we have

$$
\begin{aligned}
\delta^2(I_{ij}, \overline{I_{ij}}) &\leq \delta^2(I_{ij}, c_{ij}) \\
&\leq (1 + 7\epsilon_1)\omega(c_{ij}) \\
&\leq (1 + \epsilon_1)^3(1 + 7\epsilon_1)\delta^2(Q'_{ij}, \overline{Q'_{ij}}) \\
&\leq (1 + \epsilon/2)\delta^2(Q'_{ij}, \overline{Q'_{ij}}),
\end{aligned}
$$

where the last step follows from $\epsilon_1 = \frac{\epsilon}{126}$. Next, we bound the distance between $\overline{I_{ij}}$ and $\overline{Q_{ij}}$. According to Lemma 6, it holds that

$$
\begin{aligned}
\delta^2(I_{ij}, \overline{I_{ij} \cap Q_{ij}}) &\leq \frac{(2\alpha + \alpha\epsilon)(1+\epsilon)}{(1-3\alpha-\epsilon)|I_{ij}|} \delta^2(Q'_{ij}, \overline{Q'_{ij}}) \\
&\leq \frac{(2\alpha + \alpha\epsilon)(1+\epsilon)}{(1-3\alpha-\epsilon)} \frac{|Q'_{ij}|}{|I_{ij}||Q_{ij}|} \delta^2(Q_{ij}, \overline{Q_{ij}}) \\
&\leq \frac{(2\alpha + \alpha\epsilon)(1+\epsilon)(1-\alpha)}{(1-3\alpha-\epsilon)(1-2\alpha-\epsilon)} \frac{\delta^2(Q_{ij}, \overline{Q_{ij}})}{|Q_{ij}|}.
\end{aligned}
$$

Similarly, according to Lemma 6, we also have $\delta^2(\overline{Q_{ij}}, \overline{Q_{ij} \cap I_{ij}}) \leq \frac{2\alpha+\alpha\epsilon}{1-3\alpha-\epsilon} \frac{\delta^2(Q_{ij}, \overline{Q_{ij}})}{|Q_{ij}|}$. Putting all these together and using the triangle inequality, it holds that $\delta^2(\overline{I_{ij}}, \overline{Q_{ij}}) \leq \frac{13\alpha-15\alpha^2}{(1-3\alpha-\epsilon)(1-2\alpha-\epsilon)} \frac{\delta^2(Q_{ij}, \overline{Q_{ij}})}{|Q_{ij}|}$, where the last inequality follows from $\epsilon < 0.5$. □

**Theorem 2.** *There exists a learning-augmented $k$-means algorithm that can output a $(1 + O(\alpha))$-approximate solution in time $O(md) + \tilde{O}(\epsilon^{-5}kd/\alpha)$ with constant probability for $\alpha \in (0, 1/3 - \epsilon)$.*

**Proof** According to Lemma 7, we have $\delta^2(\overline{P^*_{ij}}, \overline{I_{ij}}) \leq \left( \frac{\alpha}{1-\alpha} + \frac{13\alpha-15\alpha^2}{(1-3\alpha-\epsilon)(1-2\alpha-\epsilon)} \right) \frac{\delta^2(P_{ij}, \overline{P^*_{ij}})}{|P^*_{ij}|}$. Denote $C$ as the set of centers returned by Algorithm 2, where $C = \{\hat{c}_1, \hat{c}_2, ..., \hat{c}_k\}$ and $\hat{c}_i$ is consisting of $d$ coordinates $[c_{i1}, c_{i2}, ..., c_{id}]$. Then, we have

$$
\begin{aligned}
\delta^2(P_{ij}, C) &\leq \sum_{i=1}^{k} \sum_{j=1}^{d} \delta^2(P^*_{ij}, c_{ij}) \\
&= \sum_{i=1}^{k} \sum_{j=1}^{d} \delta^2(P^*_{ij}, \overline{P^*_{ij}}) + |P^*_{ij}| \delta^2(\overline{P^*_{ij}}, c_{ij}) \\
&\leq \sum_{i=1}^{k} \sum_{j=1}^{d} \left( 1 + \frac{\alpha}{1-\alpha} + \frac{13\alpha-15\alpha^2}{(1-3\alpha-\epsilon)(1-2\alpha-\epsilon)} \right) \delta^2(P_{ij}, \overline{P^*_{ij}}) \\
&\leq \left( 1 + \frac{\alpha}{1-\alpha} + \frac{13\alpha-15\alpha^2}{(1-3\alpha-\epsilon)(1-2\alpha-\epsilon)} \right) \delta^2(P, C^*).
\end{aligned}
$$

Then, we give the runtime analysis for the proposed Fast-Estimation algorithm. Similar to the analysis for Algorithm 2, in each step 3 of the Fast-Estimation algorithm, sampling coordinates from the given predicted clusters takes $O(1)$ time. Then, in step 7 of Algorithm 2, we can get that the constructed candidate set of coordinates has size $O(\epsilon^{-1} \log(m\Delta_{\max}) \log(kd))$. Note that in each step 10 of Algorithm 2, the constructed estimator has size $\tilde{O}(\frac{1}{\alpha\epsilon^4})$ assuming bounded aspect ratio $\Delta_{\max}$ that $\Delta_{\max} \leq poly(m)$. Hence, in step 10 of Algorithm 2, the running time for estimating the clustering cost can be bounded by $\tilde{O}(\frac{1}{\alpha\epsilon^5})$. Finally, in each step 12 of Algorithm 2, finding the nearest $(1 - 2\alpha - \alpha\epsilon)$ coordinates in $P_{ij}$ to the selected coordinate can be executed in time $O(m_i)$ using linear selection methods (Blum et al., 1973). Then, the overall running time can be bounded by $\sum_{i=1}^{k} \sum_{j=1}^{d} O(m_i) + \tilde{O}(\frac{1}{\alpha\epsilon^5}) = O(md) + \tilde{O}(\epsilon^{-5}kd/\alpha)$.

### A.4 THEORETICAL ANALYSIS FOR THE FAST-FILTERING ALGORITHM

In this section, we present a theoretical analysis for the proposed Fast-Filtering algorithm (Algorithm 3). The intuitive idea behind is as follows. In each step 2 of Algorithm 3, we show that by carefully adjusting the sample size $R_1$, with constant probability, it is possible to sample candidate data points from each predicted cluster such that the sampled data points are close enough to the optimal clustering centers. Then, in steps 3-4 of Algorithm 3, a random and independent sample is drawn from each predicted cluster to construct an estimator that can accurately approximate the clustering cost induced by the candidate centers. Finally, by using the constructed estimator to select the best candidate center for each predicted cluster (step 5 of Algorithm 3), we prove that the clustering cost of each optimal cluster can be well approximated by using the nearest neighbor searching process (step 6 of Algorithm 3). The modified Fast-Filtering algorithm is presented in Algorithm 4.

---

**Algorithm 4** Fast-Filtering (modified)

---

**Input:** A $k$-means instance $(P, k, d)$, a set $(P_1, P_2, ..., P_k)$ of partitions with error rate $\alpha$, parameter $0 < \epsilon < 1/3$.
**Output:** A set $C \subset \mathbb{R}^d$ of centers with $|C| \leq k$.
1: **for** $i \in [k]$ **do**
2:     Randomly and independently sample a set $U_i$ from $P_i$ with size $R_1 = O(\frac{\log k}{1-2\alpha})$.
3:     Randomly and independently sample a set $S_i$ from $P_i$ with size $R_2 = O\left(\frac{\log(m^3 d \log^3(m\Delta^2)/\epsilon^2) \log(m\Delta^2)}{\alpha\epsilon^4}\right)$, and assign each point in $S_i$ a weight $\frac{m_i}{|S_i|}$.
4:     Construct an estimator $\omega$ such that $\forall u \in U_i$, $\omega(u) = \sum_{p \in S_i \setminus \mathcal{F}(u)} \frac{m_i}{|S_i|} \delta^2(p, u)$, where $\mathcal{F}(u)$ is the set of the furthest $(1 + 3\epsilon_1)\alpha|S_i|$ points from $S_i$ to $u$, where $\epsilon_1 = \epsilon/126$.
5:     $c_i = \arg\min_{u \in U_i} \omega(u)$.
6:     Let $I_i$ be the set of the nearest $(1 - 2\alpha - 20\alpha\epsilon_1)m_i$ points from $P_i$ to $c_i$.
7:     $\hat{c}_i = \overline{I_i}$.
8: **return** $\{\hat{c}_1, \hat{c}_2, ..., \hat{c}_k\}$.

---

**Theorem 3** *Let $R_1 = O(\frac{\log k}{1-2\alpha})$ and $R_2 = O\left(\frac{\log(m^3 d \log^3(m\Delta^2)/\epsilon^2) \log(m\Delta^2)}{\alpha\epsilon^4}\right)$, where $\epsilon$ is a parameter with $0 < \epsilon < 1/3$ and $\Delta$ [4] is the aspect ratio of the whole dataset (i.e., $\Delta = \frac{\max_{x,y \in P} \delta(x,y)}{\min_{x,y \in P, x \neq y} \delta(x,y)}$). With constant probability, Algorithm 4 can return a $(1 + O(\sqrt{\alpha}))$-approximate solution in time $O(md) + \tilde{O}(\frac{kd}{\epsilon^4(1-2\alpha)\alpha})$ for $\alpha \in (0, 1/3 - \epsilon)$.*

Recall that $Q_i = P_i \cap P_i^*$ is the set of data points in $P_i$ that belong to the optimal cluster $P_i^*$. Define $G_\mu(P_i^*) = \{x \in P_i^* : \delta^2(x, c_i^*) \leq \mu\delta^2(P_i^*, c_i^*)/|P_i^*|\}$ as the set of data points in $P_i^*$ that are close to $c_i^*$ (the distance is parameterized by some constant $\mu > 1$). According to Lemma 4 and the definition of $Q_i$, we have $|G_2(P_i^*)| \geq \frac{|P_i^*|}{2}$ and $|P_i^* \setminus G_2(P_i^*)| < \frac{|P_i^*|}{2}$. Then, it holds that $|P_i \cap G_2(P_i^*)| \geq |P_i^* \cap P_i| - |P_i^* \setminus G_2(P_i^*)| \geq (1-\alpha)|P_i^*| - \frac{|P_i^*|}{2} = (\frac{1}{2} - \alpha)|P_i^*|$, where the last inequality follows from the definition of the learning-augmented $k$-means model. Additionally, according to the definition of $Q_i$, we have $|P_i^*| \geq |Q_i| \geq (1-\alpha)|P_i|$, which indicates that $|P_i| \leq \frac{|P_i^*|}{1-\alpha}$. Let $\zeta_i = \frac{|P_i \cap G_2(P_i^*)|}{|P_i|}$ be the probability that a data point from $G_2(P_i^*)$ can be sampled from $P_i$ through uniform sampling strategy. Then, it holds trivially that $\zeta_i \geq (1-\alpha)(\frac{1}{2} - \alpha)$. Hence, according to Corollary 1, by randomly and independently taking a sample $U_i$ with size $\Theta(\frac{1}{1-2\alpha} \log(\frac{k}{\eta}))$ from $P_i$, with probability at least $1 - \frac{\eta}{k}$, there exists at least one data point $u \in U_i$ such that $u \in G_2(P_i^*)$. By taking a union bound over the success probability across all the predicted clusters, we can argue that with constant probability, there exists at least one data point $u \in U_i$ such that $u \in G_2(P_i^*) \cap U_i$ for each predicted cluster $i \in [k]$.

**Corollary 3** *Let $R_1 = \Theta(\frac{\log k}{1-2\alpha})$. For each $i \in [k]$, with constant probability, there exists at least one data point $u \in U_i$ such that $u \in G_2(P_i^*)$.*

Let $\epsilon_1 = \frac{\epsilon}{126}$. Similar to the analysis for the Fast-Estimation algorithm, in steps 3-4 of Algorithm 4, we can also construct a clustering cost estimator $\omega$ by randomly and independently taking a small sample $R_2$ from $P_i$ with size $O\left(\frac{\log(m^3 d \log^3(m\Delta^2)/\epsilon_1^2) \log(m\Delta^2)}{\alpha\epsilon_1^4}\right)$. Given an arbitrary data point $u \in U_i$, let $\mathcal{Z}^\dagger(u)$ be the set of the furthest $(2 + 20\epsilon_1)\alpha m_i$ data points from $P_i$ to $u$. Denote $H_i(u)$ as the set of the nearest $(1 - \alpha)m_i$ points in $P_i$ to $u$. According to Lemma 10, the constructed estimator $\omega$ in step 4 of Algorithm 4 can well approximate the clustering cost using data points in $U_i$ as centers.

**Corollary 4** *Let $R_2 = O\left(\frac{\log(m^3 d \log^3(m\Delta^2)/\epsilon_1^2) \log(m\Delta^2)}{\alpha\epsilon_1^4}\right)$. Given an arbitrary data point $u \in U_i$, with high probability, it holds that $\delta^2(P_i \setminus \mathcal{Z}^\dagger(u))/(1 + 7\epsilon_1) \leq \omega(u) \leq (1 + \epsilon_1)^2 \delta^2(H_i(u), u)$.*

According to Corollary 3, with constant probability, a data point $u_i \in U_i \cap G_2(P_i^*)$ can be sampled from $P_i$ for each predicted cluster $i \in [k]$. Hence, we have $\delta^2(H_i(u_i), u_i) \leq \delta^2(Q_i, u_i) \leq$

---

[4]It is naturally to assume that the aspect ratio of a given dataset can be bounded by a polynomial function of the data size, i.e., $\Delta = poly(n)$.

$\delta^2(P_i^*, u_i) = \delta^2(P_i^*, c_i^*) + |P_i^*|\delta^2(u_i, c_i^*) \leq 3\delta^2(P_i^*, c_i^*)$, where the first inequality follows from the definition of $H_i(u)$ that $H_i(u)$ contains the nearest $(1-\alpha)m_i$ points from $P_i$ to $u_i$, the second inequality follows from $Q_i \subseteq P_i^*$, the second to the last step follows from Lemma 1, and the last step follows from $u_i \in G_2(P_i^*)$. Then, by using Corollary 4, we can argue that with constant probability, the center $c_i$ chosen by the estimator $\omega$ in step 5 of Algorithm 4 satisfies that $\delta^2(P_i \backslash \mathcal{Z}^\dagger(c_i), c_i) \leq (1+\epsilon)\delta^2(H_i(u_i), u_i) \leq 4\delta^2(P_i^*, c_i^*)$, where the last inequality follows from $\epsilon < 1/3$.

For any predicted cluster $P_i$, recall that $\mathcal{Z}^\dagger(c_i)$ is the set of the furthest $(2+20\epsilon_1)\alpha m_i$ data points in $P_i$ to $c_i$. Let $I_i = P_i \backslash \mathcal{Z}^\dagger(c_i)$. Define $A_i = Q_i \cap \mathcal{Z}^\dagger(c_i)$, where data points in $A_i$ can be regarded as the set of the "false negatives" discarded by $c_i$. Let $B_i = P_i \backslash (Q_i \cup \mathcal{Z}^\dagger(c_i))$, where data points in $B_i$ can be regarded as the set of the "false positives" included by $c_i$. The following lemma bounds the distance between $\overline{I_i}$ and $c_i^*$.

**Lemma 12** *The following bound holds:* $\delta^2(\overline{I_i}, c_i^*) \leq \frac{9\delta^2(P_i^*, c_i^*)}{(1-(3+\epsilon)\alpha)m_i}$.

**Proof** Observe that $|I_i| = (1-(2+20\epsilon_1)\alpha)m_i$ and $|P_i \backslash P_i^*| \leq \alpha m_i$. Then, it holds that $|I_i \cap P_i^*| \geq (1-(3+20\epsilon_1)\alpha)m_i$. Observe that the clustering cost of $\delta^2(I_i, \overline{I_i})$ can be upper bounded by $\delta^2(I_i, \overline{I_i}) \leq \delta^2(I_i, c_i) \leq 4\delta^2(P_i^*, c_i^*)$. Hence, by using the relaxed triangle inequality and taking a summation over the distances between data points in $I_i$ and $c_i^*$, we can get that

$$\delta^2(\overline{I_i}, c_i^*) \leq \frac{\sum_{p \in I_i \cap P_i^*}(1+1/2)\delta^2(\overline{I_i}, p) + (1+2)\delta^2(p, c_i^*)}{|I_i \cap P_i^*|}$$
$$\leq \frac{(1+1/2)\delta^2(I_i, \overline{I_i}) + 3\delta^2(P_i^*, c_i^*)}{(1-(3+20\epsilon_1)\alpha)m_i}$$
$$\leq \frac{9\delta^2(P_i^*, c_i^*)}{(1-(3+\epsilon)\alpha)m_i},$$

where the last inequality follows from $20\epsilon_1 \leq \epsilon$. $\qquad \square$

Observe that the clustering cost of $P_i^*$ with respect to $\overline{I_i}$ can be decomposed into $\delta^2(P_i^*, \overline{I_i}) = \delta^2(P_i^* \cap P_i, \overline{I_i}) + \delta^2(P_i^* \backslash P_i, \overline{I_i})$. In the following, we first give an upper bound for the clustering cost of $\delta^2(P_i^* \cap P_i, \overline{I_i})$.

**Lemma 13** *The following bound holds:* $\delta^2(Q_i, \overline{I_i}) \leq \delta^2(Q_i, c_i^*) + \frac{O(\sqrt{\alpha})}{1-(3+\epsilon)\alpha}\delta^2(P_i^*, c_i^*)$.

**Proof** According to the definition of $Q_i$, we have $Q_i = P_i \cap P_i^* = (I_i \backslash B_i) \cup A_i$. Hence, it holds that $\delta^2(Q_i, \overline{I_i}) - \delta^2(Q_i, c_i^*) = \delta^2(I_i, \overline{I_i}) + \delta^2(A_i, \overline{I_i}) - \delta^2(B_i, \overline{I_i}) - (\delta^2(I_i, c_i^*) + \delta^2(A_i, c_i^*) - \delta^2(B_i, c_i^*))$.

We first give an upper bound for $\delta^2(B_i, c_i^*) - \delta^2(B_i, \overline{I_i})$. Observe that

$$\delta^2(B_i, c_i^*) - \delta^2(B_i, \overline{I_i}) = \sum_{b \in B_i} \delta^2(b, c_i^*) - \delta^2(B_i, \overline{I_i})$$
$$\leq \sum_{b \in B_i} (1+\sqrt{\alpha})\delta^2(b, \overline{I_i}) + (1+1/\sqrt{\alpha})\delta^2(c_i^*, \overline{I_i}) - \delta^2(B_i, \overline{I_i})$$
$$\leq \sqrt{\alpha}\delta^2(B_i, \overline{I_i}) + (1+1/\sqrt{\alpha})|B_i|\delta^2(c_i^*, \overline{I_i}),$$

where the second inequality follows from the relaxed triangle inequality.

Similarly, we can also establish an upper bound for $\delta^2(A_i, \overline{I_i}) - \delta^2(A_i, c_i^*)$. We have

$$\delta^2(A_i, \overline{I_i}) - \delta^2(A_i, c_i^*) = \sum_{a \in A_i} \delta^2(a, \overline{I_i}) - \delta^2(A_i, c_i^*)$$
$$\leq \sum_{a \in A_i} (1+\sqrt{\alpha})\delta^2(a, c_i^*) + (1+1/\sqrt{\alpha})\delta^2(c_i^*, \overline{I_i}) - \delta^2(A_i, c_i^*)$$
$$\leq \sqrt{\alpha}\delta^2(A_i, c_i^*) + (1+1/\sqrt{\alpha})|A_i|\delta^2(c_i^*, \overline{I_i}),$$

where the second inequality follows from the relaxed triangle inequality.

According to the optimality of the geometric center, it holds trivially that $\delta^2(I_i, \overline{I}_i) \leq \delta^2(I_i, c_i^*)$. Hence, we can get that

$$
\begin{aligned}
\delta^2(Q_i, \overline{I}_i) - \delta^2(Q_i, c_i^*) &\leq \sqrt{\alpha}(\delta^2(A_i, c_i^*) + \delta^2(B_i, \overline{I}_i)) + (1 + 1/\sqrt{\alpha})(|A_i| + |B_i|)\delta^2(c_i^*, \overline{I}_i) \\
&\leq \sqrt{\alpha}(\delta^2(P_i^*, c_i^*) + \delta^2(I_i, \overline{I}_i)) + 4\alpha(1 + 1/\sqrt{\alpha})m_i\delta^2(c_i^*, \overline{I}_i) \\
&\leq O(\sqrt{\alpha})\delta^2(P_i^*, c_i^*) + \frac{O(\sqrt{\alpha})}{1 - (3 + \epsilon)\alpha}\delta^2(P_i^*, c_i^*) \\
&\leq \frac{O(\sqrt{\alpha})}{1 - (3 + \epsilon)\alpha}\delta^2(P_i^*, c_i^*),
\end{aligned}
$$

where the second inequality follows from $B_i \subseteq I_i$, $|A_i| \leq |\mathcal{Z}^\dagger(c_i)| \leq 3\alpha m_i$ and $|B_i| \leq \alpha m_i$, and the last inequality follows from Corollary 4 and Lemma 12. $\square$

Finally, we can give an upper bound for the clustering cost of $P_i^*$ with respect to $\overline{I}_i$.

**Lemma 14** *The following bound holds:* $\delta^2(P_i^*, \overline{I}_i) \leq \left(1 + \frac{O(\sqrt{\alpha})}{(1-\alpha)(1-(3+\epsilon)\alpha)}\right)\delta^2(P_i^*, c_i^*).$

**Proof** According to Lemma 13, we can get that

$$
\begin{aligned}
\delta^2(P_i^*, \overline{I}_i) &= \delta^2(P_i \cap P_i^*, \overline{I}_i) + \delta^2(P_i^* \backslash P_i, \overline{I}_i) \\
&\leq \frac{O(\sqrt{\alpha})}{1 - (3 + \epsilon)\alpha}\delta^2(P_i^*, c_i^*) + \delta^2(P_i^* \cap P_i, c_i^*) + \sum_{p \in P_i^* \backslash P_i} \delta^2(p, \overline{I}_i) \\
&\leq \frac{O(\sqrt{\alpha})}{1 - (3 + \epsilon)\alpha}\delta^2(P_i^*, c_i^*) + \delta^2(P_i^* \cap P_i, c_i^*) \\
&\quad + \sum_{p \in P_i^* \backslash P_i} (1 + \sqrt{\alpha})\delta^2(p, c_i^*) + (1 + 1/\sqrt{\alpha})\delta^2(c_i^*, \overline{I}_i) \\
&\leq \left(1 + \frac{O(\sqrt{\alpha})}{1 - (3 + \epsilon)\alpha}\right)\delta^2(P_i^*, c_i^*) + (1 + 1/\sqrt{\alpha})|P_i^* \backslash P_i|\delta^2(c_i^*, \overline{I}_i) \\
&\leq \left(1 + \frac{O(\sqrt{\alpha})}{1 - (3 + \epsilon)\alpha}\right)\delta^2(P_i^*, c_i^*) \\
&\quad + (1 + 1/\sqrt{\alpha}) \cdot \frac{\alpha m_i}{1 - \alpha} \cdot \frac{9\delta^2(P_i^*, c_i^*)}{(1 - (3 + \epsilon)\alpha)m_i} \\
&\leq \left(1 + \frac{O(\sqrt{\alpha})}{(1 - \alpha)(1 - (3 + \epsilon)\alpha)}\right)\delta^2(P_i^*, c_i^*),
\end{aligned}
$$

where the first inequality follows from Lemma 13, the second inequality follows from the relaxed triangle inequality, and the second to the last inequality follows from Lemma 12. $\square$

Using similar ideas for the Fast-Sampling and Fast-Estimation algorithms, by taking a summation over all the clustering cost of the optimal clusters, Theorem 3 can be proved. As for the running time, for each predicted cluster $i \in [k]$, sampling $U_i$ and $S_i$ from $P_i$ takes $O(1)$ time. Then, in each step 5 of Algorithm 4, estimating the clustering cost of each candidate center in $U_i$ takes time $O(R_1 \times R_2)$, which is $\tilde{O}(\frac{\epsilon^{-4}}{(1-2\alpha)\alpha})$. Finally, in each step 6 of Algorithm 4, finding the nearest neighbors for a given data point can be executed in time $O(m_i d)$ using linear selection technique (Blum et al., 1973). Thus, the overall running time for Algorithm 4 is $O(md) + \tilde{O}(\frac{kd}{\epsilon^4(1-2\alpha)\alpha})$. To conclude, the Fast-Filtering algorithm can return a $(1 + O(\sqrt{\alpha}))$-approximate solution with constant probability in time $O(md) + \tilde{O}(\frac{kd}{\epsilon^4(1-2\alpha)\alpha})$.

## A.5 EXTENSION TO THE $k$-MEDIAN OBJECTIVE

In this subsection, we show how to extend our proposed sampling-based methods to the learning-augmented $k$-median problem. The key challenge here arises from the difference in optimization

objectives. In particular, for an arbitrary set $S \subset \mathbb{R}^d$ of coordinates, the geometric center of $S$ can no longer serve as the optimal clustering center for $S$ under $k$-median objective, making it difficult to identify high-quality candidate coordinates or centers. As a result, existing learning-augmented $k$-median algorithms often struggle to achieve high-quality approximation guarantees.

To overcome this challenge, our goal is to use sampling-based strategies to construct a set $U_i$ of centers that are close to the optimal clustering centers for each predicted cluster $P_i$. Then, by grid discretization, we can generate candidate centers that can well approximate the optimal clustering centers. Finally, by enumerating the constructed candidate centers, we prove that the clustering cost of each optimal cluster can be well approximated using the best center chosen from enumerations.

Table 3: Comparison results of learning augmented $k$-means algorithms

| Methods and References | Approximation Ratio | Label Error Range | Time Complexity |
| --- | --- | --- | --- |
| Paritioning and Sorting (Ergun et al., 2021) | $1 + \tilde{O}((k\alpha)^{1/4})$ | Small Constant | $O(md\log^3 m + poly(k, \log m))$ |
| Sorting (Nguyen et al., 2022) | $1 + \frac{\alpha(7+10\alpha-10\alpha^2)}{(1-\alpha)(1-2\alpha)}$ | $[0, 1/2)$ | $O(\frac{md\log^3 m \log^2(k/\delta)}{1-2\alpha})$ |
| Fast-Sampling (Ours) | $1 + \frac{\alpha(6+4\epsilon-4\alpha-3\epsilon\alpha)}{(1-\alpha)(1-2\alpha)}$ | $(0, 1/2)$ | $O(\frac{md\log(kd)\log(m\Delta)}{1-2\alpha} \cdot (\frac{\sqrt{d}}{\epsilon\alpha})^{O(d)})$ |

Table 3 provides a detailed comparison of the results for the learning-augmented $k$-median problem. We also give a plot (Figure 2) of approximation ratios vs. the error rate $\alpha$. It can be seen from the table that the current best result achieves a $(1 + O(\alpha))$-approximation with $\alpha \in [0, 1/2)$ (Nguyen et al., 2022). Compared to the state-of-the-art results, the Fast-Sampling algorithm can achieve better clustering quality guarantees with slightly worse running time for fixed dimension $d$.

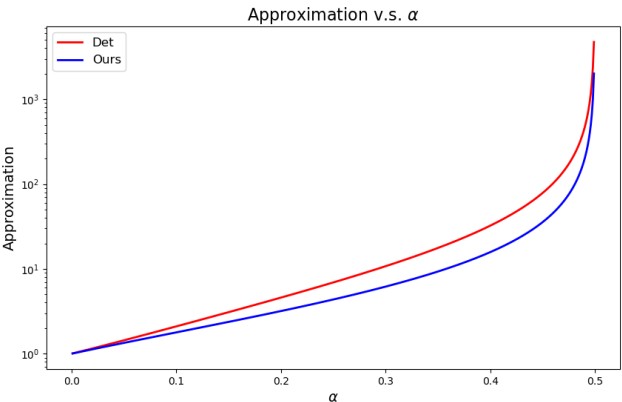

Figure 2: Plot of approximation ratios v.s. the error rate $\alpha$ for the $k$-median objective.

The formal description for the learning-augmented $k$-median algorithm is presented in Algorithm 5. The general idea behind the algorithm is to first generate candidate centers that can closely approximate the optimal clustering centers for each predicted cluster. Then, by picking the best center with minimum $k$-median cost, we prove that the proposed algorithm can give better approximation guarantees for the learning-augmented $k$-median problem. In the following, we give a formal analysis for the proposed algorithm.

Without loss of generality, we can assume that the minimum pairwise distance between data points in $P$ is 1 while the maximum pairwise distance is $\Delta$. Note that this can be done using standard scaling techniques. According to Lemma 4, in each step 2 of Algorithm 5, with probability at least $1 - \frac{1}{k}$, at least one center $u \in U_i$ can be found such that $\delta(u, c_i^*) \leq 2\delta(P_i^*, c_i^*)/|P_i^*| \leq \frac{2\delta(P_i^*, c_i^*)}{(1-\alpha)m_i}$, where the last step follows from the fact that $|P_i^*| \geq |Q_i| \geq (1 - \alpha)m_i$. Then, in step 3 of Algorithm 5, since the algorithm enumerates all the possible values between 1 and $\log(m\Delta)$, there exists at least one guess for the clustering radius (step 4 of Algorithm 5) such that $\delta(P_i^*, c_i^*)/(1-\alpha)m_i \leq l_i \leq 2\delta(P_i^*, c_i^*)/(1-\alpha)m_i$. Hence, in step 6 of Algorithm 5, the grid centered at $u$ with side length $2l_i$ ($G(u)$) should contain the optimal clustering center $c_i^*$. Then, in step 7 of Algorithm 5, by decomposing the grid $G(u)$ into smaller subgrids with side length $(1-\alpha)\alpha\epsilon_1 l_i/\sqrt{d}$ for some

---

**Algorithm 5** Fast-Sampling ($k$-median)

---

**Input:** A $k$-median instance $(P, k, d)$, a set $(P_1, ..., P_k)$ of partitions with error rate $\alpha$, and a parameter $\epsilon \in (0, 1]$.
**Output:** A set $C \subset \mathbb{R}^d$ of centers with $|C| = k$.
 1: **for** $i \in [k]$ **do**
 2:     Randomly and independently sample a set $U_i$ from $P_i$ with size $O\left(\frac{\log(kd)}{1-2\alpha}\right)$, then initialize $U_i' = \emptyset$.
 3:     **for** $q = 0$ to $O(\log(m\Delta))$ **do**
 4:         $l_i = 2^{q-1}/(1-\alpha)m_i$.
 5:         **for** $u \in U_i$ **do**
 6:             Let $G(u)$ be the grid centered at $u$ with side length $2l_i$.
 7:             Decompose $G(u)$ into smaller subgrids with side length $(1-\alpha)\alpha\epsilon_1 l_i/\sqrt{d}$, and let $s(u)$ be the set that contains the centers for the decomposed subgrids, where $\epsilon_1 < \epsilon/4$.
 8:             $U_i' = U_i' \cup s(u)$.
 9:     Set $u_i = \arg\min_{u \in U_i'} \delta(\mathcal{N}_i(u), u)$, where $\mathcal{N}_i(u)$ is the set of the nearest $(1-\alpha)m_i$ points in $P_i$ to $u \in U_i'$.
10:     $\hat{c}_i = u_i$.
11: **return** $\{\hat{c}_1, \hat{c}_2, ..., \hat{c}_k\}$.

---

$\epsilon_1 < \frac{\epsilon}{4}$, the optimal clustering center $c_i^*$ must also belong to some of the subgrids. Since the subgrid has side length $(1-\alpha)\alpha\epsilon_1 l_i/\sqrt{d}$, there also exists at least one $u' \in U_i'$ such that $u'$ is close enough to $c_i^*$, i.e., $\delta(u', c_i^*) \leq (1-\alpha)\alpha\epsilon_1 l_i \leq \alpha\epsilon\delta(P_i^*, c_i^*)/m_i$. Let $u_i$ be the point chosen in step 9 of Algorithm 5. For any data point $u \in U_i'$, let $\mathcal{N}_i(u)$ be the set of the nearest $(1-\alpha)m_i$ points in $P_i$ to $u$. Consequently, we have

$$
\begin{aligned}
\delta(\mathcal{N}_i(u_i), u_i) &\leq \delta(\mathcal{N}_i(u'), u') \\
&\leq \delta(\mathcal{N}_i(u_i), u') \\
&\leq \delta(\mathcal{N}_i(u_i), c_i^*) + |\mathcal{N}_i(u_i)|\delta(u', c_i^*) \\
&\leq \delta(\mathcal{N}_i(u_i), c_i^*) + \alpha\epsilon\delta(P_i^*, c_i^*),
\end{aligned}
$$

where the first inequality follows from the fact that $\delta(\mathcal{N}_i(u_i), u_i)$ induces the minimum clustering cost, the second inequality follows from the definition of $\mathcal{N}_i(u')$ that $\mathcal{N}_i(u')$ contains the nearest $(1-\alpha)m_i$ points in $P_i$ to $u'$, the third inequality follows from the triangle inequality, and the last inequality follows from $\delta(u', c_i^*) \leq \alpha\epsilon\delta(P_i^*, c_i^*)/m_i$ and $|\mathcal{N}_i(u_i)| \leq m_i$.

**Corollary 5** *For some $i \in [k]$, with probability at least $1 - 1/k$, it holds that $\delta(\mathcal{N}_i(u_i), u_i) \leq \delta(\mathcal{N}_i(u_i), c_i^*) + \alpha\epsilon\delta(P_i^*, c_i^*)$.*

For any predicted cluster $P_i$, let $\mathcal{Z}^\dagger(u_i)$ be the set of the furthest $\alpha m_i$ coordinates in $P_i$ to $u_i$, i.e., $\mathcal{Z}^\dagger(u_i) = P_i \backslash \mathcal{N}_i(u_i)$. Let $A_i = Q_i \cap \mathcal{Z}^\dagger(u_i)$ and $B_i = P_i \backslash (Q_i \cup \mathcal{Z}^\dagger(u_i))$ be the set of the "false negatives" and "false positives", respectively. In the following lemma, we first give an upper bound for the distance between $c_i^*$ and $u_i$.

**Lemma 15** *The following bound holds: $\delta(u_i, c_i^*) \leq \frac{(2+\alpha\epsilon)\delta(P_i^*, c_{ij}^*)}{(1-2\alpha)m_i}$.*

**Proof** Since $|\mathcal{N}_i(u_i)| = (1-\alpha)m_i$ and $|P_i \backslash P_i^*| \leq \alpha m_i$, it holds trivially that $|\mathcal{N}_i(u_i) \cap P_i^*| \geq (1-2\alpha)m_i$. According to the definition of $u_i$ that $\mathcal{N}_i(u_i)$ induces the minimum clustering cost, we can get that $\delta(\mathcal{N}_i(u_i), u_i) \leq \delta(\mathcal{N}_i(u'), u') \leq \delta(Q_i, u') \leq \delta(Q_i, c_i^*) + |Q_i|\delta(u', c_i^*) \leq (1+\alpha\epsilon)\delta(P_i^*, c_i^*)$, where the last inequality follows from $\delta(u', c_i^*) \leq \alpha\epsilon\delta(P_i^*, c_i^*)/m_i$. Then, by using the triangle inequality, we have

$$
\begin{aligned}
\delta(u_i, c_i^*) &\leq \frac{\sum_{p \in \mathcal{N}_i(u_i) \cap P_i^*} \delta(p, u_i) + \delta(p, c_i^*)}{|\mathcal{N}_i(u_i) \cap P_i^*|} \\
&\leq \frac{\delta(\mathcal{N}_i(u_i), u_i) + \delta(P_i^*, c_i^*)}{(1-2\alpha)m_i} \\
&\leq \frac{(2+\alpha\epsilon)\delta(P_i^*, c_i^*)}{(1-2\alpha)m_i},
\end{aligned}
$$

where the first inequality follows from the triangle inequality. □

The following lemma gives an upper bound for the clustering cost of $Q_i$ with respect to $u_i$.

**Lemma 16** *The following bound holds:* $\delta(Q_i, u_i) \leq \delta(Q_i, c_i^*) + \frac{\alpha(4+3\epsilon)}{1-2\alpha}\delta(P_i^*, c_i^*)$

**Proof** Since $Q_i = \mathcal{N}_i(u_i)\backslash B_i \cup A_i$, we have $\delta(Q_i, u_i) - \delta(Q_i, c_i^*) = \delta(\mathcal{N}_i(u_i), u_i) - \delta(B_i, u_i) + \delta(A_i, u_i) - (\delta(\mathcal{N}_i(u_i), c_i^*) - \delta(B_i, c_i^*) + \delta(A_i, c_i^*))$. We consider to bound these terms separately.

For the clustering cost of $\delta(A_i, u_i) - \delta(A_i, c_i^*)$, we can get that $\delta(A_i, u_i) - \delta(A_i, c_i^*) \leq \delta(A_i, c_i^*) + |A_i|\delta(c_i^*, u_i) - \delta(A_i, c_i^*) \leq |A_i|\delta(c_i^*, u_i)$. Similarly, the clustering cost of $\delta(B_i, c_i^*) - \delta(B_i, u_i)$ can be bounded by $\delta(B_i, c_i^*) - \delta(B_i, u_i) \leq \delta(B_i, u_i) + |B_i|\delta(u_i, c_i^*) - \delta(B_i, u_i) \leq |B_i|\delta(u_i, c_i^*)$.

Next, we bound the size of $|A_i|$ and $|B_i|$. Observe that $Q_i = \mathcal{N}_i(u_i)\backslash B_i \cup A_i$. Consequently, we have $|Q_i| = |\mathcal{N}_i(u_i)| + |A_i| - |B_i| = (1-\alpha)m_i + |A_i| - |B_i| \geq (1-\alpha)m_i$, where the last inequality follows from $|Q_i| \geq (1 - \alpha)m_i$. Hence, we can get that $|B_i| \leq |A_i| \leq \alpha m_i$.

Then, it holds that

$$
\begin{aligned}
\delta(Q_i, u_i) - \delta(Q_i, c_i^*) &\leq \delta(\mathcal{N}_i(u_i), u_i) - \delta(\mathcal{N}_i(u_i), c_i^*) + (|A_i| + |B_i|)\delta(u_i, c_i^*) \\
&\leq \delta(\mathcal{N}_i(u_i), c_i^*) + \alpha\epsilon\delta(P_i^*, c_i^*) - \delta(\mathcal{N}_i(u_i), c_i^*) \\
&\quad + (|A_i| + |B_i|)\delta(u_i, c_i^*) \\
&\leq \alpha\epsilon\delta(P_i^*, c_i^*) + 2\alpha m_i \frac{(2+\alpha\epsilon)\delta(P_i^*, c_i^*)}{(1-2\alpha)m_i} \\
&\leq \frac{\alpha(4+3\epsilon)}{1-2\alpha}\delta(P_i^*, c_i^*).
\end{aligned}
$$

Finally, we can bound the clustering cost of $P_i^*$ with respect to $u_i$ for each $i \in [k]$.

**Lemma 17** *For each* $i \in [k]$, *with probability at least* $1 - 1/k$, $\delta(P_i^*, u_i) \leq \left(1 + \frac{6\alpha + 4\alpha\epsilon - 4\alpha^2 - 3\epsilon\alpha^2}{(1-\alpha)(1-2\alpha)}\right)\delta(P_i^*, c_i^*)$.

**Proof** For an arbitrary predicted cluster $P_i^*$, the clustering cost of $P_i^*$ with respect to $u_i$ can be decomposed into $\delta(P_i^*, u_i) = \delta(P_i^* \cap P_i, u_i) + \delta(P_i^*\backslash P_i, u_i)$. Hence, we can get that

$$
\begin{aligned}
\delta(P_i^*, u_i) &= \delta(P_i^* \cap P_i, u_i) + \delta(P_i^*\backslash P_i, u_i) \\
&\leq \delta(P_i \cap P_i^*, c_i^*) + \frac{\alpha(4+3\epsilon)}{1-2\alpha}\delta(P_i^*, c_i^*) + \delta(P_i^*\backslash P_i, u_i) \\
&\leq \frac{\alpha(4+3\epsilon)}{1-2\alpha}\delta(P_i^*, c_i^*) + \delta(P_i \cap P_i^*, c_i^*) + \delta(P_i^*\backslash P_i, c_i^*) + |P_i^*\backslash P_i|\delta(c_i^*, u_i) \\
&\leq \delta(P_i^*, c_i^*) + \frac{\alpha m_i}{1-\alpha} \cdot \frac{(2+\alpha\epsilon)\delta(P_i^*, c_i^*)}{(1-2\alpha)m_i} + \frac{\alpha(4+3\epsilon)}{1-2\alpha}\delta(P_i^*, c_i^*) \\
&\leq \left(1 + \frac{6\alpha + 4\alpha\epsilon - 4\alpha^2 - 3\epsilon\alpha^2}{(1-\alpha)(1-2\alpha)}\right)\delta(P_i^*, c_i^*).
\end{aligned}
$$

Putting all these together, we can obtain a $(1 + O(\alpha))$-approximation for the learning-augmented $k$-median problem. As for the running time, for each predicted cluster, the sampling process takes $O(1)$ time in step 2 of Algorithm 5. Guessing the optimal clustering cost in step 3 of Algorithm 5 induces an multiplicative $O(\log(m\Delta))$ factor loss. Then, constructing the candidate coordinates takes time $O(\frac{\log(kd)}{1-2\alpha} \cdot (\frac{\sqrt{d}}{\epsilon\alpha})^{O(d)})$. Finally, finding the nearest coordinates for each candidate center takes time $O(m_i d)$ in step 9 of Algorithm 5 using linear selection method (Blum et al., 1973). Consequently, the overall running time of Algorithm 5 is $O(\frac{md\log(kd)\log(m\Delta)}{1-2\alpha} \cdot (\frac{\sqrt{d}}{\epsilon\alpha})^{O(d)})$.

## A.6 COMPLEMENTARY EXPERIMENTS

**Other Results with Varying Error Rate and Fixed Clusters.** Tables 4-8 show the experimental results on datasets Mnist, PHY, CIFAR10, SUSY, and HIGGS for varying error rate with fixed clusters,

where "Ref" reports the clustering cost of the optimal labeling partitions and the running time for constructing the labeling partitions using $k$-means++ and Lloyd's algorithms. On average, by calculating the results over all the datasets used in the experiments, the Fast-Filtering algorithm achieves a $1.5\%$ reduction in clustering cost compared to the current state-of-the-art learning-augmented $k$-means algorithm. As for the running time, it can be seen from the tables that our Fast-Filtering algorithm is significantly faster than other learning-augmented algorithms across all the datasets, with this trend becoming more pronounced as the data sizes and dimensionality grow. On average, Fast-Filtering is at least 3 times faster than the current state-of-the-art learning-augmented algorithm. For the Fast-Estimation, on average, the Fast-Estimation algorithm is at least 2 times faster than previous learning-augmented algorithms on large-scale datasets with sizes over 5 million.

**Results with Varying Clusters and Fixed Error Rate.** Tables 9-14 present the comparison results for varying number of clusters with fixed error rate. For clustering cost, our Fast-Sampling algorithm is comparable with other learning-augmented algorithms, while our Fast-Filtering algorithm consistently achieves better clustering quality for most cases. On average, by calculating the results over all the datasets used in the experiments, the Fast-Filtering algorithm achieves at least $1.6\%$ reduction in clustering cost compared to the current state-of-the-art learning-augmented $k$-means algorithm. Regarding the running time, it can be seen from the table that the running time for all learning-augmented algorithms remains relatively stable as the number of clusters increases. However, our Fast-Filtering algorithm is significantly faster than other learning-augmented algorithms across all the datasets, with this trend becoming more pronounced as the data sizes and dimensionality grow. On average, our Fast-Filtering algorithm is at least 3 times faster than the current state-of-the-art learning-augmented algorithm. On the largest dataset SIFT, the running time of our Fast-Filtering algorithm is at least 10 times faster than other learning-augmented methods, which further demonstrates the effectiveness of our proposed sampling-based strategies for handling large-scale datasets.

**Comparisons with the Lloyd's and $k$-means++ Methods.** When comparing with the Lloyd's method, it can be seen from the tables that our Fast-Filtering algorithm is at least $50.9\%$ faster than the Lloyd's method across different datasets. However, other learning-augmented algorithms may not achieve much better running time than Lloyd's method (such as on dataset SIFT). When comparing with the $k$-means++ method, on average, our Fast-Filtering algorithm achieves at least a 20% reduction in clustering cost compared to $k$-means++. Furthermore, it provides over 50% improvements in ARI and NMI values across most datasets. These results show the advantage of incorporating learning-augmented and sampling strategies for improving the clustering quality.

**Discussions on the Limitations for the Algorithms.** In the following, we give a brief discussion on the bottlenecks for different algorithms. For sorting-based algorithms, such as those in Ergun et al. (2021); Nguyen et al. (2022), the primary bottlenecks are their running time due to the sorting processes, where an additional $O(\log m)$ term is included in the running time. The experiments demonstrate that the sorting-based algorithms do not scale well when handling large-scale datasets. On the other hand, one of the limitations for our sampling-based algorithms is that they might achieve worse clustering quality guarantees compared with sorting-based methods. However, these guarantees are worst-case scenarios. Experiments show that our algorithms perform competitively on large-scale datasets like SUSY, HIGGS and SIFT.

### A.6.1 Experimental Results

Table 4: Comparisons on dataset MNIST for varying $\alpha$ and fixed $k = 20$

| Method | Ref | $\alpha$ | Cost | NMI | ARI | Time(s) |
|---|---|---|---|---|---|---|
| Dataset MNIST (1,797 × 64) | | | | | | |
| k-means++ | | | 1.5445E+06±2.49E+04 | 0.7087±0.0139 | 0.5502±0.0255 | **0.03±0.04** |
| Ergun | | | 9.9174E+05±1.19E+03 | 0.9680±0.0036 | 0.9665±0.0053 | 0.34±0.68 |
| Det | | | 9.7566E+05±0.00E+00 | 0.9819±0.0000 | 0.9821±0.0000 | 0.13±0.17 |
| Fast-Sampling | 9.6032E+05(1.37s) | 0.1 | **9.6471E+05±5.93E+02** | **0.9849±0.0025** | **0.9844±0.0027** | **0.11±0.09** |
| Fast-Filtering | | | 9.6640E+05±3.80E+02 | 0.9814±0.0029 | 0.9793±0.0035 | 0.48±1.36 |
| Fast-Estimation | | | 9.7553E+05±3.79E+02 | 0.9806±0.0023 | 0.9801±0.0025 | 1.82±0.37 |
| k-means++ | | | 1.5475E+06±3.70E+04 | 0.6861±0.0170 | 0.4825±0.0230 | **0.02±0.01** |
| Ergun | | | 1.0201E+06±2.35E+03 | 0.9482±0.0055 | 0.9394±0.0077 | 0.12±0.01 |
| Det | | | 1.0036E+06±0.00E+00 | 0.9531±0.0000 | 0.9460±0.0000 | 0.09±0.02 |
| Fast-Sampling | 9.6177+05(1.49s) | 0.2 | 9.9901E+05±1.32E+03 | 0.9551±0.0049 | 0.9484±0.0069 | 0.09±0.02 |
| Fast-Filtering | | | **9.7548E+05±9.86E+02** | **0.9632±0.0116** | **0.9553±0.0157** | **0.03±0.01** |
| Fast-Estimation | | | 1.0093E+06±1.76E+03 | 0.9556±0.0022 | 0.9487±0.0027 | 1.84±0.14 |
| k-means++ | | | 1.5551E+06±3.41E+04 | 0.7072±0.0191 | 0.5426±0.0494 | **0.01±0.00** |
| Ergun | | | 1.0656E+06±3.63E+03 | 0.9316±0.0052 | 0.9254±0.0058 | 0.12±0.00 |
| Det | | | 1.0411E+06±0.00E+00 | 0.9405±0.0000 | **0.9336±0.0000** | 0.08±0.01 |
| Fast-Sampling | 9.5226E+05(1.76s) | 0.3 | 1.0713E+06±3.64E+03 | 0.9174±0.0040 | 0.8942±0.0065 | 0.08±0.00 |
| Fast-Filtering | | | **9.9369E+05±4.74E+03** | **0.9422±0.0096** | 0.9335±0.0132 | **0.03±0.00** |
| Fast-Estimation | | | 1.0595E+06±2.56E+03 | 0.9331±0.0065 | 0.9254±0.0094 | 1.78±0.13 |
| k-means++ | | | 1.5541E+06±3.35E+04 | 0.6893±0.0231 | 0.4897±0.0451 | **0.01±0.00** |
| Ergun | | | 1.1859E+06±5.28E+03 | 0.8804±0.0059 | 0.8359±0.0086 | 0.12±0.01 |
| Det | | | 1.1454E+06±0.00E+00 | 0.8812±0.0000 | 0.8395±0.0000 | 0.08±0.00 |
| Fast-Sampling | 9.6728E+05(1.32s) | 0.4 | 1.1379E+06±3.84E+03 | **0.8975±0.0069** | 0.8497±0.0100 | 0.09±0.01 |
| Fast-Filtering | | | **1.0633E+06±1.05E+04** | 0.8975±0.0082 | **0.8577±0.0150** | **0.03±0.00** |
| Fast-Estimation | | | 1.1736E+06±3.85E+03 | 0.8851±0.0091 | 0.8433±0.0137 | 1.84±0.09 |
| k-means++ | | | 1.5464E+06±4.35E+04 | 0.7042±0.0234 | 0.5263±0.0468 | **0.02±0.02** |
| Ergun | | | 1.2969E+06±7.41E+03 | 0.8483±0.0084 | 0.7822±0.0120 | 0.12±0.01 |
| Det | | | 1.2417E+06±0.00E+00 | 0.8601±0.0000 | 0.8040±0.0000 | 0.08±0.00 |
| Fast-Sampling | 9.4992E+05(1.29s) | 0.5 | 1.2578E+06±5.77E+03 | **0.8622±0.0088** | **0.8076±0.0134** | 0.09±0.01 |
| Fast-Filtering | | | **1.1622E+06±1.41E+04** | 0.8581±0.0122 | 0.8000±0.0195 | **0.03±0.00** |
| Fast-Estimation | | | 1.2844E+06±4.67E+03 | 0.8470±0.0049 | 0.7805±0.0080 | 1.81±0.08 |

Table 5: Comparisons on dataset CIFAR10 for varying $\alpha$ and fixed $k = 20$

| Method | Ref | $\alpha$ | Cost | NMI | ARI | Time(s) |
|---|---|---|---|---|---|---|
| Dataset CIFAR10 (10,000 × 3,072) | | | | | | |
| k-means++ | | | 1.2782E+11±6.26E+09 | 0.3995±0.0171 | 0.1683±0.0207 | **1.90±0.08** |
| Ergun | | | 7.8812E+10±2.06E+06 | 0.9096±0.0007 | 0.8879±0.0012 | 19.15±1.00 |
| Det | | | 7.8619E+10±0.00E+00 | 0.9009±0.0000 | 0.8761±0.0000 | 20.45±0.51 |
| Fast-Sampling | 7.8337E+10(37.41s) | 0.1 | **7.8335E+10±4.65E+07** | 0.8970±0.0131 | 0.8671±0.0215 | 19.91±0.49 |
| Fast-Filtering | | | 7.8611E+10±1.59E+07 | 0.9106±0.0042 | 0.8941±0.0060 | **5.61±1.53** |
| Fast-Estimation | | | 7.8633E+10±6.58E+05 | **0.9186±0.0006** | **0.9074±0.0006** | 185.51±2.30 |
| k-means++ | | | 1.2248E+11±4.44E+09 | 0.4043±0.0165 | 0.1601±0.0129 | **1.89±0.04** |
| Ergun | | | 7.9135E+10±3.07E+06 | **0.8700±0.0010** | **0.8292±0.0013** | 18.67±0.32 |
| Det | | | **7.8899E+10±0.00E+00** | 0.8515±0.0000 | 0.7999±0.0000 | 20.20±0.09 |
| Fast-Sampling | 7.8306E+10(33.19s) | 0.2 | 7.9257E+10±1.62E+08 | 0.8087±0.0218 | 0.7354±0.0348 | 19.73±0.08 |
| Fast-Filtering | | | 7.9112E+10±4.77E+07 | 0.8314±0.0100 | 0.7682±0.0172 | **5.19±0.10** |
| Fast-Estimation | | | 7.9180E+10±3.93E+06 | 0.8568±0.0008 | 0.8086±0.0013 | 187.49±0.93 |
| k-means++ | | | 1.2844E+11±7.17E+09 | 0.3975±0.0114 | 0.1643±0.0248 | **1.97±0.07** |
| Ergun | | | 8.0341E+10±6.09E+06 | **0.7999±0.0006** | **0.7190±0.0010** | 18.77±0.20 |
| Det | | | **8.0095E+10±0.00E+00** | 0.7727±0.0000 | 0.6711±0.0000 | 20.33±0.18 |
| Fast-Sampling | 7.8202E+10(30.42s) | 0.3 | 8.0278E+10±2.15E+08 | 0.7435±0.0165 | 0.6173±0.0301 | 19.86±0.12 |
| Fast-Filtering | | | 8.0760E+10±3.01E+08 | 0.7406±0.0124 | 0.6072±0.0231 | **5.06±0.08** |
| Fast-Estimation | | | 8.1302E+10±1.61E+07 | 0.7826±0.0011 | 0.6857±0.0013 | 188.72±0.92 |
| k-means++ | | | 1.2489E+11±3.46E+09 | 0.4062±0.0104 | 0.1628±0.0275 | **1.99±0.06** |
| Ergun | | | **8.1533E+10±1.07E+07** | **0.7528±0.0010** | **0.6316±0.0013** | 18.94±0.18 |
| Det | | | 8.1549E+10±0.00E+00 | 0.7347±0.0000 | 0.6011±0.0000 | 20.16±0.08 |
| Fast-Sampling | 7.8061E+10(26.96s) | 0.4 | 8.1915E+10±4.78E+08 | 0.6751±0.0357 | 0.5105±0.0600 | 19.80±0.17 |
| Fast-Filtering | | | 8.3463E+10±4.21E+08 | 0.6767±0.0132 | 0.4966±0.0174 | **5.03±0.08** |
| Fast-Estimation | | | 8.3900E+10±3.76E+07 | 0.7331±0.0011 | 0.5772±0.0017 | 188.79±1.89 |
| k-means++ | | | 1.2305E+11±7.21E+09 | 0.4116±0.0221 | 0.1646±0.0264 | **1.95±0.04** |
| Ergun | | | **8.3372E+10±1.82E+07** | **0.7160±0.0010** | **0.5644±0.0016** | 18.76±0.23 |
| Det | | | 8.4653E+10±0.00E+00 | 0.6877±0.0000 | 0.5090±0.0000 | 20.24±0.10 |
| Fast-Sampling | 7.8211E+10(22.77s) | 0.5 | 8.4686E+10±7.21E+08 | 0.6131±0.0207 | 0.4061±0.0306 | 19.85±0.15 |
| Fast-Filtering | | | 8.8030E+10±5.56E+08 | 0.6348±0.0144 | 0.4052±0.0170 | **4.95±0.04** |
| Fast-Estimation | | | 8.7858E+10±5.03E+07 | 0.6875±0.0012 | 0.4834±0.0014 | 190.86±2.17 |

Table 6: Comparisons on dataset PHY for varying $\alpha$ and fixed $k = 20$

| Method | Ref | $\alpha$ | Cost | NMI | ARI | Time(s) |
|---|---|---|---|---|---|---|
| | | | Dataset PHY (10,000 $\times$ 50) | | | |
| k-means++ | | | 4.2591E+11±2.21E+10 | 0.8505±0.0131 | 0.6795±0.0348 | **0.19±0.08** |
| Ergun | | | 2.9413E+11±7.21E+07 | 0.9921±0.0018 | 0.9919±0.0024 | 3.17±0.61 |
| Det | 2.9135E+11(15.33s) | 0.1 | 2.9368E+11±0.00E+00 | **0.9937±0.0000** | **0.9942±0.0000** | 2.39±0.17 |
| Fast-Sampling | | | 3.0865E+11±9.02E+08 | 0.9658±0.0065 | 0.9514±0.0118 | 2.37±0.13 |
| Fast-Filtering | | | **2.9171E+11±5.22E+07** | 0.9869±0.0012 | 0.9842±0.0017 | **1.04±1.52** |
| Fast-Estimation | | | 2.9413E+11±7.43E+07 | 0.9888±0.0025 | 0.9870±0.0034 | 5.98±0.39 |
| k-means++ | | | 4.3154E+11±2.92E+10 | 0.8522±0.0134 | 0.6796±0.0382 | **0.18±0.07** |
| Ergun | | | 3.1388E+11±2.41E+08 | 0.9815±0.0022 | 0.9767±0.0034 | 2.72±0.10 |
| Det | 3.0196E+11(19.86s) | 0.2 | 3.0880E+11±0.00E+00 | **0.9867±0.0000** | **0.9853±0.0000** | 2.35±0.07 |
| Fast-Sampling | | | 3.1571E+11±8.30E+09 | 0.9512±0.0092 | 0.9231±0.0175 | 2.38±0.11 |
| Fast-Filtering | | | **3.0304E+11±9.26E+07** | 0.9791±0.0041 | 0.9727±0.0065 | **0.56±0.04** |
| Fast-Estimation | | | 3.0805E+11±3.20E+08 | 0.9793±0.0024 | 0.9741±0.0034 | 6.00±0.13 |
| k-means++ | | | 4.1248E+11±1.74E+10 | 0.8552±0.0107 | 0.6915±0.0301 | **0.14±0.05** |
| Ergun | | | 3.2578E+11±9.33E+08 | **0.9749±0.0031** | **0.9681±0.0051** | 2.58±0.10 |
| Det | 3.0531E+11(22.04s) | 0.3 | 3.2548E+11±6.10E-05 | 0.9738±0.0000 | 0.9654±0.0000 | 2.28±0.04 |
| Fast-Sampling | | | 3.5979E+11±1.14E+10 | 0.9291±0.0165 | 0.8746±0.0375 | 2.28±0.06 |
| Fast-Filtering | | | **3.1726E+11±1.78E+09** | 0.9506±0.0023 | 0.9248±0.0049 | **0.51±0.04** |
| Fast-Estimation | | | 3.2336E+11±1.21E+09 | 0.9508±0.0061 | 0.9228±0.0122 | 5.80±0.13 |
| k-means++ | | | 4.3777E+11±2.97E+10 | 0.8514±0.0209 | 0.6740±0.0611 | **0.15±0.11** |
| Ergun | | | 3.3504E+11±1.13E+09 | **0.9793±0.0033** | **0.9704±0.0064** | 2.63±0.10 |
| Det | 3.0385E+11(20.17s) | 0.4 | 3.3356E+11±6.10E-05 | 0.9709±0.0000 | 0.9619±0.0000 | 2.34±0.06 |
| Fast-Sampling | | | 4.2907E+11±4.32E+10 | 0.8951±0.0181 | 0.7914±0.0505 | 2.35±0.07 |
| Fast-Filtering | | | **3.3306E+11±1.58E+10** | 0.9279±0.0077 | 0.8836±0.0157 | 0.53±0.04 |
| Fast-Estimation | | | 3.3655E+11±3.64E+09 | 0.9336±0.0101 | 0.8883±0.0225 | 6.11±0.22 |
| k-means++ | | | 4.3389E+11±3.42E+10 | 0.8434±0.0112 | 0.6601±0.0345 | **0.12±0.04** |
| Ergun | | | 3.2413E+11±6.91E+08 | 0.9796±0.0031 | **0.9758±0.0044** | 2.68±0.09 |
| Det | 2.9137E+11(16.62s) | 0.5 | 3.2330E+11±6.10E-05 | **0.9791±0.0000** | 0.9756±0.0000 | 2.40±0.07 |
| Fast-Sampling | | | 6.4952E+11±7.75E+10 | 0.8711±0.0138 | 0.7214±0.0346 | 2.36±0.08 |
| Fast-Filtering | | | **3.2278E+11±1.09E+11** | 0.8684±0.0108 | 0.7249±0.0309 | **0.56±0.08** |
| Fast-Estimation | | | 3.4066E+11±9.39E+09 | 0.9157±0.0093 | 0.8503±0.0203 | 6.07±0.14 |

Table 7: Comparisons on dataset SUSY for varying $\alpha$ and fixed $k = 20$

| Method | Ref | $\alpha$ | Cost | NMI | ARI | Time(s) |
|---|---|---|---|---|---|---|
| | | | Dataset SUSY (5,000,000 $\times$ 18) | | | |
| k-means++ | | | 3.4792E+07±6.17E+05 | 0.4743±0.0152 | 0.2597±0.0163 | **8.61±0.21** |
| Ergun | | | 2.5432E+07±5.55E+03 | 0.8778±0.0002 | 0.8672±0.0003 | 130.97±2.89 |
| Det | 2.4406E+07(424.67s) | 0.1 | 2.5221E+07±0.00E+00 | 0.9138±0.0000 | 0.9178±0.0000 | 154.27±2.24 |
| Fast-Sampling | | | 2.5247E+07±8.81E+03 | **0.9166±0.0013** | **0.9209±0.0017** | 119.50±2.01 |
| Fast-Filtering | | | **2.4921E+07±6.71E+04** | 0.8434±0.0073 | 0.8203±0.0104 | **36.77±1.15** |
| Fast-Estimation | | | 2.5246E+07±2.43E+03 | 0.9031±0.0008 | 0.9040±0.0010 | 51.30±0.76 |
| k-means++ | | | 3.3779E+07±1.77E+05 | 0.4833±0.0130 | 0.2604±0.0245 | **9.31±0.54** |
| Ergun | | | 2.7453E+07±7.38E+03 | 0.8474±0.0001 | 0.8224±0.0001 | 134.35±1.44 |
| Det | 2.4551E+07(318.64s) | 0.2 | 2.7162E+07±0.00E+00 | **0.8866±0.0000** | **0.8801±0.0000** | 155.83±0.99 |
| Fast-Sampling | | | 2.7253E+07±1.12E+05 | 0.8627±0.0373 | 0.8447±0.0541 | 121.29±1.64 |
| Fast-Filtering | | | **2.6769E+07±2.96E+05** | 0.8207±0.0224 | 0.7920±0.0341 | **39.96±1.85** |
| Fast-Estimation | | | 2.7201E+07±3.12E+03 | 0.8770±0.0003 | 0.8674±0.0004 | 53.58±1.20 |
| k-means++ | | | 3.4445E+07±4.52E+05 | 0.4908±0.0125 | 0.2675±0.0131 | **9.25±0.46** |
| Ergun | | | 2.9191E+07±3.78E+03 | 0.8395±0.0002 | 0.8000±0.0004 | 134.53±2.65 |
| Det | 2.4465E+07(257.38s) | 0.3 | 2.9043E+07±0.00E+00 | **0.8778±0.0000** | **0.8593±0.0000** | 157.23±1.92 |
| Fast-Sampling | | | 2.9245E+07±9.94E+04 | 0.8489±0.0256 | 0.8281±0.0343 | 122.78±2.85 |
| Fast-Filtering | | | **2.8439E+07±2.30E+05** | 0.8232±0.0153 | 0.7839±0.0249 | **41.32±1.29** |
| Fast-Estimation | | | 2.9051E+07±7.21E+03 | 0.8669±0.0004 | 0.8443±0.0005 | 54.03±0.94 |
| k-means++ | | | 3.4685E+07±5.34E+05 | 0.4764±0.0177 | 0.2616±0.0283 | **9.37±0.41** |
| Ergun | | | 3.2623E+07±8.38E+03 | 0.8323±0.0002 | 0.8178±0.0004 | 133.73±1.38 |
| Det | 2.4851E+07(303.73s) | 0.4 | 3.2695E+07±0.00E+00 | **0.8590±0.0000** | **0.8595±0.0000** | 154.98±1.35 |
| Fast-Sampling | | | 3.2412E+07±1.32E+05 | 0.8145±0.0513 | 0.7796±0.0758 | 124.09±2.68 |
| Fast-Filtering | | | **3.2218E+07±3.81E+05** | 0.8094±0.0172 | 0.7873±0.0226 | **41.59±0.98** |
| Fast-Estimation | | | 3.2704E+07±1.05E+04 | 0.8525±0.0008 | 0.8508±0.0007 | 54.45±0.77 |
| k-means++ | | | 3.4139E+07±6.91E+05 | 0.4963±0.0087 | 0.2795±0.0168 | **9.47±0.58** |
| Ergun | | | 3.4316E+07±8.13E+03 | 0.8350±0.0004 | 0.8033±0.0006 | 133.74±1.96 |
| Det | 2.4360E+07(252.52s) | 0.5 | 3.4546E+07±0.00E+00 | 0.8387±0.0000 | 0.8093±0.0000 | 154.20±1.72 |
| Fast-Sampling | | | 3.3944E+07±3.04E+05 | 0.7282±0.0613 | 0.6470±0.0946 | 123.75±2.04 |
| Fast-Filtering | | | **3.3883E+07±4.14E+05** | 0.7705±0.0293 | 0.7183±0.0482 | **40.01±2.86** |
| Fast-Estimation | | | 3.4622E+07±1.39E+04 | **0.8467±0.0116** | **0.8225±0.0147** | 54.91±1.70 |

Table 8: Comparisons on dataset HIGGS for varying $\alpha$ and fixed $k = 20$

| Dataset HIGGS (11,000,000 × 27) | | | | | | |
|---|---|---|---|---|---|---|
| Method | Ref | $\alpha$ | Cost | NMI | ARI | Time(s) |
| k-means++ | | | 2.0309E+08±1.91E+06 | 0.3749±0.0182 | 0.1813±0.0112 | **21.36±0.25** |
| Ergun | | | 1.4083E+08±1.59E+03 | 0.9093±0.0006 | 0.9094±0.0009 | 432.42±2.72 |
| Det | | | 1.4019E+08±0.00E+00 | 0.9613±0.0000 | **0.9674±0.0000** | 481.06±2.76 |
| Fast-Sampling | 1.3924E+08(861.32s) | 0.1 | 1.4198E+08±8.33E+03 | **0.9618±0.0007** | 0.9639±0.0007 | 428.25±3.12 |
| Fast-Filtering | | | **1.4002E+08±3.72E+04** | 0.8989±0.0021 | 0.8937±0.0027 | **136.62±7.03** |
| Fast-Estimation | | | 1.4014E+08±4.02E+03 | 0.9471±0.0010 | 0.9524±0.0008 | 171.12±2.17 |
| k-means++ | | | 2.0562E+08±7.85E+05 | 0.3627±0.0206 | 0.1756±0.0186 | **20.68±0.36** |
| Ergun | | | 1.4383E+08±3.49E+03 | 0.9060±0.0010 | 0.9056±0.0013 | 425.37±3.68 |
| Det | | | 1.4236E+08±0.00E+00 | **0.9146±0.0000** | **0.9149±0.0000** | 481.58±3.41 |
| Fast-Sampling | 1.3911E+08(724.24s) | 0.2 | 1.4304E+08±3.68E+04 | 0.8780±0.0019 | 0.8721±0.0021 | 432.06±1.99 |
| Fast-Filtering | | | **1.4172E+08±1.34E+05** | 0.8597±0.0114 | 0.8376±0.0165 | **146.53±5.21** |
| Fast-Estimation | | | 1.4252E+08±1.36E+04 | 0.9001±0.0013 | 0.8950±0.0018 | 171.63±3.24 |
| k-means++ | | | 2.0574E+08±4.32E+06 | 0.3676±0.0148 | 0.1793±0.0095 | **20.51±0.82** |
| Ergun | | | 1.4860E+08±3.97E+03 | **0.8947±0.0015** | **0.8942±0.0023** | 418.54±7.96 |
| Det | | | 1.4613E+08±0.00E+00 | 0.8739±0.0000 | 0.8697±0.0000 | 469.16±6.86 |
| Fast-Sampling | 1.3929E+08(729.15s) | 0.3 | 1.4700E+08±2.02E+05 | 0.8198±0.0059 | 0.7848±0.0109 | 413.38±12.49 |
| Fast-Filtering | | | **1.4508E+08±1.37E+05** | 0.8475±0.0070 | 0.8256±0.0090 | **144.12±4.13** |
| Fast-Estimation | | | 1.4595E+08±4.48E+04 | 0.8557±0.0068 | 0.8412±0.0086 | 167.43±7.17 |
| k-means++ | | | 2.0099E+08±3.56E+06 | 0.3818±0.0094 | 0.1917±0.0150 | **20.00±0.32** |
| Ergun | | | 1.5418E+08±1.05E+04 | **0.8777±0.0024** | **0.8706±0.0035** | 413.61±1.45 |
| Det | | | 1.4999E+08±0.00E+00 | 0.8011±0.0000 | 0.7693±0.0000 | 459.55±1.29 |
| Fast-Sampling | 1.3957E+08(707.38s) | 0.4 | 1.5068E+08±2.42E+05 | 0.7479±0.0230 | 0.6781±0.0345 | 397.79±4.56 |
| Fast-Filtering | | | **1.4814E+08±3.55E+05** | 0.8054±0.0082 | 0.7716±0.0092 | **139.27±9.70** |
| Fast-Estimation | | | 1.4818E+08±7.22E+04 | 0.7965±0.0108 | 0.7702±0.0127 | 160.61±2.36 |
| k-means++ | | | 2.0108E+08±2.95E+06 | 0.3927±0.0080 | 0.1976±0.0034 | **20.54±0.48** |
| Ergun | | | 1.6151E+08±1.65E+05 | 0.6733±0.0072 | 0.5333±0.0119 | 431.85±5.45 |
| Det | | | 1.5447E+08±0.00E+00 | 0.7332±0.0000 | 0.6156±0.0000 | 478.20±8.28 |
| Fast-Sampling | 1.4127E+08(771.82s) | 0.5 | 1.5672E+08±2.18E+05 | 0.5783±0.0209 | 0.3969±0.0388 | 428.04±6.82 |
| Fast-Filtering | | | **1.5395E+08±1.17E+06** | 0.7593±0.0112 | **0.6771±0.0156** | **129.40±4.43** |
| Fast-Estimation | | | 1.5431E+08±6.90E+04 | **0.7498±0.0356** | 0.6550±0.0615 | 164.92±4.15 |

Table 9: Comparisons on dataset MNIST for varying $k$ and fixed $\alpha = 0.2$

| Dataset MNIST (1,797 × 64) | | | | | | |
|---|---|---|---|---|---|---|
| Method | Ref | $k$ | Cost | NMI | ARI | Time(s) |
| k-means++ | | | 1.9717E+06±2.63E+04 | 0.6392±0.0415 | 0.5213±0.0737 | **0.01±0.01** |
| Ergun | | | 1.2304E+06±1.88E+03 | 0.9465±0.0039 | 0.9490±0.0039 | 0.33±0.74 |
| Det | | | 1.2186E+06±2.33E-10 | 0.9456±0.0000 | 0.9449±0.0000 | 0.14±0.17 |
| Fast-Sampling | 1.1695+06(2.01s) | 10 | 1.2201E+06±1.50E+03 | 0.9330±0.0059 | 0.9256±0.0078 | **0.12±0.11** |
| Fast-Filtering | | | **1.1894E+06±2.58E+03** | **0.9489±0.0123** | **0.9520±0.0130** | 0.49±1.41 |
| Fast-Estimation | | | 1.2209E+06±1.85E+03 | 0.9425±0.0053 | 0.9426±0.0057 | 1.32±0.41 |
| k-means++ | | | 1.5669E+06±4.62E+04 | 0.6857±0.0135 | 0.4761±0.0197 | **0.01±0.00** |
| Ergun | | | 1.0119E+06±2.02E+03 | 0.9519±0.0043 | 0.9391±0.0061 | 0.11±0.00 |
| Det | | | 9.9410E+05±0.00E+00 | 0.9591±0.0000 | 0.9479±0.0000 | 0.07±0.00 |
| Fast-Sampling | 9.5547+05(1.99s) | 20 | 9.9914E+05±1.84E+03 | 0.9584±0.0058 | 0.9493±0.0108 | 0.08±0.00 |
| Fast-Filtering | | | **9.6684E+05±1.14E+03** | 0.9619±0.0069 | 0.9498±0.0099 | **0.02±0.00** |
| Fast-Estimation | | | 1.0011E+06±9.51E+02 | **0.9619±0.0032** | **0.9505±0.0047** | 1.60±0.00 |
| k-means++ | | | 1.3399E+06±2.30E+04 | 0.7127±0.0129 | 0.4844±0.0313 | **0.02±0.01** |
| Ergun | | | 9.0804E+05±1.79E+03 | 0.9457±0.0042 | 0.9283±0.0067 | 0.15±0.02 |
| Det | | | 8.8892E+05±1.16E-10 | 0.9592±0.0000 | 0.9482±0.0000 | 0.09±0.01 |
| Fast-Sampling | 8.5000E+05(1.67s) | 30 | 8.9789E+05±9.56E+02 | 0.9488±0.0049 | 0.9266±0.0098 | 0.10±0.02 |
| Fast-Filtering | | | **8.6468E+05±1.23E+03** | **0.9622±0.0060** | **0.9524±0.0091** | **0.03±0.00** |
| Fast-Estimation | | | 8.9725E+05±9.94E+02 | 0.9503±0.0019 | 0.9345±0.0026 | 2.45±0.24 |
| k-means++ | | | 1.2120E+06±2.20E+04 | 0.7373±0.0071 | 0.4599±0.0197 | **0.03±0.01** |
| Ergun | | | 8.3789E+05±5.56E+02 | 0.9508±0.0041 | 0.9207±0.0062 | 0.17±0.02 |
| Det | | | 8.1200E+05±1.16E-10 | **0.9640±0.0000** | **0.9451±0.0000** | 0.11±0.01 |
| Fast-Sampling | 7.7018E+05(1.44s) | 40 | 8.2700E+05±1.01E+03 | 0.9573±0.0037 | 0.9328±0.0068 | 0.12±0.02 |
| Fast-Filtering | | | **7.8453E+05±1.41E+03** | 0.9595±0.0083 | 0.9360±0.0145 | **0.04±0.01** |
| Fast-Estimation | | | 8.2322E+05±9.22E+02 | 0.9622±0.0039 | 0.9427±0.0068 | 3.08±0.12 |
| k-means++ | | | 1.1248E+06±5.29E+03 | 0.7384±0.0069 | 0.4296±0.0179 | **0.03±0.01** |
| Ergun | | | 7.9776E+05±2.46E+03 | 0.9434±0.0042 | 0.8944±0.0078 | 0.19±0.01 |
| Det | | | 7.6429E+05±0.00E+00 | 0.9601±0.0000 | **0.9274±0.0000** | 0.12±0.01 |
| Fast-Sampling | 7.2273E+05(1.53s) | 50 | 7.7443E+05±1.12E+03 | 0.9553±0.0033 | 0.9224±0.0057 | 0.13±0.00 |
| Fast-Filtering | | | **7.4142E+05±2.77E+03** | **0.9611±0.0108** | 0.9257±0.0237 | **0.05±0.01** |
| Fast-Estimation | | | 7.7657E+05±1.15E+03 | 0.9567±0.0030 | 0.9200±0.0074 | 3.62±0.06 |

Table 10: Comparisons on dataset CIFAR10 for varying $k$ and fixed $\alpha = 0.2$

| Dataset CIFAR10 (10,000 × 3,072) | | | | | | |
|---|---|---|---|---|---|---|
| Method | Ref | $k$ | Cost | NMI | ARI | Time(s) |
| k-means++ | | | 1.4120E+11±9.75E+09 | 0.3873±0.0302 | 0.2036±0.0404 | **1.02±0.09** |
| Ergun | | | 8.5415E+10±4.64E+06 | **0.8480±0.0046** | **0.8335±0.0056** | 19.59±1.09 |
| Det | 8.4673E+10(10.94s) | 10 | **8.5355E+10±1.53E-05** | 0.8335±0.0000 | 0.8160±0.0000 | 21.54±0.57 |
| Fast-Sampling | | | 8.5777E+10±1.49E+08 | 0.7855±0.0303 | 0.7499±0.0452 | 20.76±0.48 |
| Fast-Filtering | | | 8.5735E+10±1.00E+08 | 0.8186±0.0116 | 0.7920±0.0177 | **6.15±1.41** |
| Fast-Estimation | | | 8.5537E+10±4.04E+06 | 0.8469±0.0011 | 0.8327±0.0013 | 125.27±2.99 |
| k-means++ | | | 1.2415E+11±7.80E+09 | 0.4005±0.0294 | 0.1541±0.0311 | **1.85±0.05** |
| Ergun | | | 7.8972E+10±4.51E+06 | **0.8700±0.0009** | **0.8281±0.0011** | 18.55±0.27 |
| Det | 7.8808E+10(36.81s) | 20 | **7.8750E+10±0.00E+00** | 0.8516±0.0000 | 0.7993±0.0000 | 20.19±0.26 |
| Fast-Sampling | | | 7.9135E+10±9.31E+07 | 0.8033±0.0157 | 0.7266±0.0247 | 19.66±0.13 |
| Fast-Filtering | | | 7.8920E+10±8.79E+07 | 0.8361±0.0084 | 0.7750±0.0157 | **4.92±0.08** |
| Fast-Estimation | | | 7.9032E+10±3.84E+06 | 0.8602±0.0010 | 0.8140±0.0016 | 182.33±1.94 |
| k-means++ | | | 1.1647E+11±3.62E+09 | 0.4304±0.0109 | 0.1612±0.0131 | **3.44±0.12** |
| Ergun | | | 7.6223E+10±3.55E+06 | **0.8618±0.0009** | **0.7931±0.0016** | 18.82±0.22 |
| Det | 7.5010E+10(29.54s) | 30 | **7.5958E+10±1.53E-05** | 0.8472±0.0000 | 0.7746±0.0000 | 19.82±0.11 |
| Fast-Sampling | | | 7.6109E+10±7.85E+07 | 0.8147±0.0167 | 0.7209±0.0283 | 19.49±0.22 |
| Fast-Filtering | | | 7.5991E+10±5.29E+07 | 0.8292±0.0061 | 0.7471±0.0111 | **4.74±0.11** |
| Fast-Estimation | | | 7.6431E+10±2.60E+06 | 0.8510±0.0008 | 0.7906±0.0013 | 238.71±4.44 |
| k-means++ | | | 1.1570E+11±4.48E+09 | 0.4279±0.0152 | 0.1268±0.0233 | **4.67±0.20** |
| Ergun | | | 7.3929E+10±2.10E+06 | **0.8851±0.0008** | **0.8288±0.0013** | 19.25±0.33 |
| Det | 7.2836E+10(37.33s) | 40 | **7.3547E+10±0.00E+00** | 0.8667±0.0000 | 0.8000±0.0000 | 19.74±0.16 |
| Fast-Sampling | | | 7.3839E+10±1.39E+08 | 0.8245±0.0128 | 0.7198±0.0242 | 19.49±0.22 |
| Fast-Filtering | | | 7.3691E+10±8.40E+07 | 0.8272±0.0101 | 0.7275±0.0185 | **4.45±0.09** |
| Fast-Estimation | | | 7.3983E+10±2.59E+06 | 0.8622±0.0009 | 0.7878±0.0012 | 279.69±0.81 |
| k-means++ | | | 1.1445E+11±4.80E+09 | 0.4276±0.0137 | 0.1235±0.0186 | **5.77±0.12** |
| Ergun | | | 7.2902E+10±2.31E+06 | **0.8693±0.0006** | **0.7979±0.0011** | 19.46±0.31 |
| Det | 7.1331E+10(39.24s) | 50 | 7.2428E+10±0.00E+00 | 0.8481±0.0000 | 0.7581±0.0000 | 19.98±0.18 |
| Fast-Sampling | | | 7.2349E+10±7.78E+07 | 0.8109±0.0193 | 0.6888±0.0365 | 19.67±0.14 |
| Fast-Filtering | | | **7.2331E+10±7.19E+07** | 0.8498±0.0064 | 0.7603±0.0107 | **4.37±0.12** |
| Fast-Estimation | | | 7.3004E+10±3.01E+06 | 0.8555±0.0006 | 0.7775±0.0013 | 334.36±1.25 |

Table 11: Comparisons on dataset PHY for varying $k$ and fixed $\alpha = 0.2$

| Dataset PHY (10,000 × 50) | | | | | | |
|---|---|---|---|---|---|---|
| Method | Ref | $k$ | Cost | NMI | ARI | Time(s) |
| k-means++ | | | 1.4500E+12±1.21E+11 | 0.8077±0.0399 | 0.6678±0.0769 | **0.08±0.01** |
| Ergun | | | 1.0332E+12±1.55E+09 | 0.9772±0.0043 | 0.9752±0.0051 | 2.75±0.66 |
| Det | 1.0148E+12(4.62s) | 10 | **1.0236E+12±1.22E-04** | **0.9801±0.0000** | **0.9806±0.0000** | 2.48±0.16 |
| Fast-Sampling | | | 1.0853E+12±1.12E+10 | 0.9370±0.0128 | 0.9207±0.0199 | 2.45±0.09 |
| Fast-Filtering | | | 1.0273E+12±1.69E+09 | 0.9763±0.0026 | 0.9757±0.0027 | **1.08±1.41** |
| Fast-Estimation | | | 1.0288E+12±3.19E+09 | 0.9709±0.0074 | 0.9679±0.0108 | 4.07±0.36 |
| k-means++ | | | 4.3425E+11±2.19E+10 | 0.8494±0.0176 | 0.6685±0.0442 | **0.13±0.08** |
| Ergun | | | 3.1167E+11±2.87E+08 | 0.9812±0.0041 | 0.9770±0.0060 | 2.69±0.36 |
| Det | 3.0028E+11(9.93s) | 20 | **3.0705E+11±6.10E-05** | **0.9882±0.0000** | **0.9866±0.0000** | 2.31±0.10 |
| Fast-Sampling | | | 3.2665E+11±7.21E+09 | 0.9455±0.0099 | 0.9118±0.0219 | 2.29±0.11 |
| Fast-Filtering | | | **3.0129E+11±5.01E+07** | 0.9789±0.0034 | 0.9740±0.0053 | **0.50±0.04** |
| Fast-Estimation | | | 3.0641E+11±2.58E+08 | 0.9787±0.0024 | 0.9731±0.0036 | 5.98±0.27 |
| k-means++ | | | 2.4134E+11±4.04E+09 | 0.8562±0.0085 | 0.6563±0.0284 | **0.15±0.06** |
| Ergun | | | 1.7710E+11±1.16E+08 | 0.9814±0.0020 | 0.9741±0.0034 | 2.53±0.09 |
| Det | 1.6596E+11(18.70s) | 30 | 1.7480E+11±3.05E-05 | **0.9855±0.0000** | **0.9825±0.0000** | 2.19±0.05 |
| Fast-Sampling | | | 1.8463E+11±3.71E+09 | 0.9393±0.0084 | 0.8845±0.0224 | 2.20±0.05 |
| Fast-Filtering | | | **1.6730E+11±8.38E+07** | 0.9563±0.0012 | 0.9214±0.0024 | **0.50±0.02** |
| Fast-Estimation | | | 1.7302E+11±1.08E+08 | 0.9783±0.0022 | 0.9709±0.0035 | 7.27±0.09 |
| k-means++ | | | 1.7705E+11±4.01E+09 | 0.8641±0.0070 | 0.6561±0.0249 | **0.16±0.04** |
| Ergun | | | 1.3471E+11±1.21E+08 | 0.9812±0.0024 | **0.9714±0.0048** | 2.73±0.15 |
| Det | 1.2380E+11(41.26s) | 40 | 1.3117E+11±1.53E-05 | **0.9816±0.0000** | 0.9694±0.0000 | 2.19±0.06 |
| Fast-Sampling | | | 1.4682E+11±4.54E+09 | 0.9346±0.0077 | 0.8693±0.0221 | 2.19±0.05 |
| Fast-Filtering | | | **1.2451E+11±4.92E+07** | 0.9763±0.0014 | 0.9601±0.0028 | **0.52±0.04** |
| Fast-Estimation | | | 1.3027E+11±9.43E+07 | 0.9798±0.0035 | 0.9684±0.0079 | 8.63±0.15 |
| k-means++ | | | 1.4425E+11±3.96E+09 | 0.8544±0.0065 | 0.6206±0.0223 | **0.19±0.05** |
| Ergun | | | 1.1089E+11±1.38E+08 | **0.9821±0.0017** | **0.9725±0.0031** | 2.62±0.08 |
| Det | 9.8416E+10(50.11s) | 50 | 1.0700E+11±0.00E+00 | 0.9785±0.0000 | 0.9653±0.0000 | 2.10±0.04 |
| Fast-Sampling | | | 1.1886E+11±3.56E+09 | 0.9321±0.0101 | 0.8490±0.0303 | 2.12±0.04 |
| Fast-Filtering | | | **1.0107E+11±6.73E+07** | 0.9705±0.0039 | 0.9456±0.0086 | **0.50±0.02** |
| Fast-Estimation | | | 1.0596E+11±1.27E+08 | 0.9750±0.0032 | 0.9572±0.0067 | 9.64±0.37 |

Table 12: Comparisons on dataset SUSY for varying $k$ and fixed $\alpha = 0.2$

| Dataset SUSY (5,000,00 × 18) | | | | | | |
|---|---|---|---|---|---|---|
| Method | Ref | $k$ | Cost | NMI | ARI | Time(s) |
| k-means++ | | | 4.2626E+07±1.5E+05 | 0.4212±0.0495 | 0.2851±0.0632 | **6.29±0.32** |
| Ergun | | | 3.1954E+07±3.4E+03 | 0.8037±0.0014 | 0.7953±0.0017 | 129.61±2.43 |
| Det | | | **3.1759E+07±0** | 0.8628±0 | 0.8696±0 | 159.18±2.90 |
| Fast-Sampling | 2.9914E+07(194.06s) | 10 | 3.2683E+07±1.4E+04 | 0.8599±0.0019 | **0.8760±0.0021** | 121.91±1.16 |
| Fast-Filtering | | | 3.1814E+07±1.1E+03 | **0.8662±0.0004** | 0.8747±0.0004 | **37.52±1.87** |
| Fast-Estimation | | | 3.1779E+07±7.7E+03 | 0.8479±0.0031 | 0.8521±0.0032 | 48.18±1.57 |
| k-means++ | | | 3.3848E+07±3.7E+05 | 0.4833±0.0107 | 0.2671±0.0099 | **9.27±0.24** |
| Ergun | | | 2.7086E+07±5.6E+03 | 0.8507±0 | 0.8201±0.0001 | 135.43±2.26 |
| Det | | | 2.6856E+07±0 | **0.8913±0** | **0.8805±0** | 155.56±1.50 |
| Fast-Sampling | 2.4533E+07(307.01s) | 20 | 2.6889E+07±3.1E+04 | 0.8899±0.0012 | 0.8778±0.0013 | 125.01±2.09 |
| Fast-Filtering | | | **2.6121E+07±3.1E+05** | 0.8247±0.0091 | 0.7898±0.0140 | **39.78±1.07** |
| Fast-Estimation | | | 2.6890E+07±6.9E+03 | 0.8807±0.0006 | 0.8664±0.0009 | 54.35±0.87 |
| k-means++ | | | 2.9916E+07±5.4E+05 | 0.5140±0.0048 | 0.2509±0.0112 | **12.30±0.15** |
| Ergun | | | 2.4828E+07±7.9E+03 | 0.8562±0 | 0.8183±0.0001 | 134.08±0.92 |
| Det | | | 2.4707E+07±0 | **0.8915±0** | **0.8759±0** | 152.09±1.89 |
| Fast-Sampling | 2.1903E+07(520.44s) | 30 | 2.4870E+07±8.9E+03 | 0.8870±0.0007 | 0.8753±0.0007 | 117.35±3.02 |
| Fast-Filtering | | | **2.3846E+07±1.6E+05** | 0.8564±0.0161 | 0.8278±0.0266 | **36.32±0.79** |
| Fast-Estimation | | | 2.4646E+07±1.7E+04 | 0.8705±0.0223 | 0.8435±0.0345 | 54.37±0.95 |
| k-means++ | | | 2.7709E+07±3.7E+05 | 0.5285±0.0152 | 0.2408±0.0185 | **15.56±0.83** |
| Ergun | | | 2.3405E+07±7.0E+03 | 0.8593±0.0001 | 0.8144±0.0002 | 144.44±2.17 |
| Det | | | 2.3197E+07±0 | **0.8813±0** | **0.8525±0** | 149.24±0.97 |
| Fast-Sampling | 2.0262E+07(913.86s) | 40 | 2.3107E+07±4.7E+04 | 0.8724±0.0112 | 0.8385±0.0151 | 110.81±2.21 |
| Fast-Filtering | | | **2.2082E+07±2.3E+05** | 0.8436±0.0065 | 0.7994±0.0120 | **35.08±1.72** |
| Fast-Estimation | | | 2.3153E+07±8.2E+04 | 0.8708±0.0198 | 0.8378±0.0306 | 55.65±1.08 |
| k-means++ | | | 2.5509E+07±2.4E+05 | 0.5447±0.0082 | 0.0237±0.0147 | **17.48±0.32** |
| Ergun | | | 2.1978E+07±3.7E+03 | 0.8586±0.0001 | 0.7988±0.0003 | 147.95±1.91 |
| Det | | | 2.1894E+07±0 | **0.8805±0** | **0.8371±0** | 144.89±1.95 |
| Fast-Sampling | 1.9136E+07(1345.42s) | 50 | 2.1799E+07±1.0E+05 | 0.8396±0.0289 | 0.7749±0.0468 | 101.44±3.42 |
| Fast-Filtering | | | **2.0645E+07±6.5E+04** | 0.8716±0.0065 | 0.8315±0.0112 | **29.77±1.34** |
| Fast-Estimation | | | 2.1435E+07±4.1E+04 | 0.8339±0.0026 | 0.7645±0.0046 | 54.19±1.51 |

Table 13: Comparisons on dataset HIGGS for varying $k$ and fixed $\alpha = 0.2$

| Dataset HIGGS (11,000,000 × 27) | | | | | | |
|---|---|---|---|---|---|---|
| Method | Ref | $k$ | Cost | NMI | ARI | Time(s) |
| k-means++ | | | 2.3351E+08±4.85E+06 | 0.3063±0.0073 | 0.1806±0.0030 | **14.92±0.12** |
| Ergun | | | 1.5783E+08±4.95E+03 | 0.9263±0.0034 | 0.9349±0.0039 | 427.50±6.17 |
| Det | | | 1.5626E+08±0.00E+00 | 0.9239±0.0000 | 0.9327±0.0000 | 481.12±4.88 |
| Fast-Sampling | 1.5371E+08(343.72s) | 10 | 1.5787E+08±3.68E+03 | **0.9445±0.0006** | **0.9595±0.0007** | 443.06±6.59 |
| Fast-Filtering | | | **1.5559E+08±1.99E+05** | 0.9028±0.0067 | 0.9079±0.0066 | **129.32±5.96** |
| Fast-Estimation | | | 1.5637E+08±7.66E+04 | 0.9176±0.0014 | 0.9260±0.0010 | 164.52±2.60 |
| k-means++ | | | 2.0632E+08±2.49E+06 | 0.3454±0.0042 | 0.1658±0.0029 | **20.56±0.29** |
| Ergun | | | 1.4375E+08±1.26E+03 | 0.9085±0.0011 | 0.9066±0.0012 | 445.15±2.97 |
| Det | | | 1.4227E+08±0.00E+00 | **0.9148±0.0000** | **0.9135±0.0000** | 484.35±0.50 |
| Fast-Sampling | 1.3908E+08(706.94s) | 20 | 1.4307E+08±6.93E+04 | 0.8827±0.0010 | 0.8768±0.0017 | 439.74±2.25 |
| Fast-Filtering | | | **1.4191E+08±1.38E+05** | 0.9017±0.0322 | 0.8937±0.0413 | **142.67±1.82** |
| Fast-Estimation | | | 1.4237E+08±2.91E+04 | 0.9020±0.0014 | 0.8971±0.0019 | 173.38±1.69 |
| k-means++ | | | 1.8190E+08±5.53E+05 | 0.4060±0.0092 | 0.1772±0.0061 | **28.81±0.08** |
| Ergun | | | 1.3663E+08±2.89E+03 | 0.9083±0.0000 | 0.8985±0.0000 | 450.15±9.71 |
| Det | | | 1.3520E+08±0.00E+00 | **0.9154±0.0000** | **0.9069±0.0000** | 478.83±6.94 |
| Fast-Sampling | 1.3241E+08(1437.67s) | 30 | 1.3554E+08±6.70E+04 | 0.8757±0.0001 | 0.8522±0.0005 | 425.87±9.23 |
| Fast-Filtering | | | **1.3425E+08±8.98E+04** | 0.8739±0.0087 | 0.8482±0.0128 | **147.26±7.24** |
| Fast-Estimation | | | 1.3525E+08±3.39E+03 | 0.8986±0.0002 | 0.8855±0.0003 | 175.96±3.08 |
| k-means++ | | | 1.7659E+08±1.75E+06 | 0.4153±0.0122 | 0.1610±0.0086 | **35.46±0.59** |
| Ergun | | | 1.3153E+08±2.79E+03 | 0.9046±0.0007 | 0.8921±0.0009 | 445.79±9.27 |
| Det | | | 1.3021E+08±0.00E+00 | **0.9146±0.0000** | **0.9032±0.0000** | 470.93±4.65 |
| Fast-Sampling | 1.2694E+08(2301.98s) | 40 | 1.3060E+08±6.99E+04 | 0.8633±0.0243 | 0.8276±0.0367 | 398.50±9.65 |
| Fast-Filtering | | | **1.2917E+08±1.05E+05** | 0.8936±0.0068 | 0.8727±0.0101 | **139.57±0.59** |
| Fast-Estimation | | | 1.3002E+08±4.50E+04 | 0.8958±0.0002 | 0.8775±0.0013 | 172.35±0.55 |
| k-means++ | | | 1.6815E+08±8.74E+05 | 0.4342±0.0032 | 0.1611±0.0057 | **38.57±0.02** |
| Ergun | | | 1.2730E+08±1.45E+04 | 0.8785±0.0229 | 0.8467±0.0338 | 436.17±0.26 |
| Det | | | 1.2589E+08±0.00E+00 | **0.9113±0.0000** | **0.8932±0.0000** | 443.46±0.38 |
| Fast-Sampling | 1.2336E+08(3574.13s) | 50 | 1.2645E+08±1.07E+05 | 0.8666±0.0218 | 0.8303±0.0324 | 364.05±3.10 |
| Fast-Filtering | | | **1.2490E+08±4.14E+04** | 0.8542±0.0015 | 0.8065±0.0018 | **133.17±0.40** |
| Fast-Estimation | | | 1.2587E+08±1.53E+04 | 0.8958±0.0017 | 0.8710±0.0019 | 160.75±0.99 |

Table 14: Comparisons on dataset SIFT for varying $k$ and fixed $\alpha = 0.2$

| | | | Dataset SIFT (100,000,000 × 128) | | | |
|---|---|---|---|---|---|---|
| Method | Ref | $k$ | Cost | NMI | ARI | Time(s) |
| k-means++ | | | 1.8947E+13±1.41E+11 | 0.2994±0.0180 | 0.2026±0.0087 | **801.75±114.99** |
| Ergun | | | 1.1067E+13±1.16E+08 | 0.9065±0.0000 | 0.9085±0.0000 | 23298.85±2009.69 |
| Det | | | 1.0835E+13±0.00E+00 | 0.9115±0.0000 | 0.9146±0.0000 | 14759.49±401.91 |
| Fast-Sampling | 1.0542E+13(844.18s) | 10 | 1.2093E+13±2.15E+11 | 0.5353±0.0306 | 0.4319±0.0255 | 14370.31±552.58 |
| Fast-Filtering | | | **1.0763E+13±5.73E+09** | 0.8879±0.0006 | 0.8959±0.0001 | **1013.03±77.22** |
| Fast-Estimation | | | 1.0856E+13±3.27E+08 | **0.9160±0.0004** | **0.9206±0.0005** | 14612.00±1312.35 |
| k-means++ | | | 1.6512E+13±1.26E+11 | 0.3269±0.0197 | 0.1353±0.0315 | **1329.66±173.52** |
| Ergun | | | 1.0200E+13±6.41E+07 | 0.9012±0.0000 | 0.8912±0.0001 | 20445.05±2033.90 |
| Det | | | 9.9855E+12±0.00E+00 | 0.9024±0.0000 | 0.8914±0.0000 | 14195.19±576.63 |
| Fast-Sampling | 9.7055E+12(1011.24s) | 20 | 1.1236E+13±2.55E+11 | 0.5600±0.0310 | 0.4002±0.0483 | 13877.33±778.01 |
| Fast-Filtering | | | **9.8954E+12±2.44E+09** | 0.8768±0.0055 | 0.8637±0.0062 | **1059.94±99.14** |
| Fast-Estimation | | | 9.9979E+12±4.08E+08 | **0.9077±0.0002** | **0.9002±0.0003** | 13987.80±1117.39 |
| k-means++ | | | 1.5338E+13±2.51E+11 | 0.3730±0.0112 | 0.1535±0.0094 | **2238.99±46.63** |
| Ergun | | | 9.7328E+12±3.49E+07 | 0.9013±0.0000 | 0.8737±0.0000 | 19099.74±1359.36 |
| Det | | | 9.5218E+12±0.00E+00 | 0.9027±0.0000 | 0.8745±0.0000 | 14119.17±1024.58 |
| Fast-Sampling | 9.2478E+12(1330.99s) | 30 | 1.1072E+13±4.91E+09 | 0.5429±0.0058 | 0.3297±0.0074 | 13738.71±763.40 |
| Fast-Filtering | | | **9.4282E+12±3.38E+09** | 0.8863±0.0061 | 0.8590±0.0082 | **1100.15±104.72** |
| Fast-Estimation | | | 9.5332E+12±6.38E+08 | **0.9072±0.0001** | **0.8832±0.0002** | 13633.17±1347.76 |
| k-means++ | | | 1.5006E+13±1.58E+11 | 0.3936±0.0072 | 0.1610±0.0027 | **2976.74±25.20** |
| Ergun | | | 9.4558E+12±6.92E+07 | 0.9019±0.0001 | 0.8699±0.0001 | 18493.35±995.04 |
| Det | | | 9.2487E+12±0.00E+00 | 0.9017±0.0000 | 0.8668±0.0000 | 13813.96±599.81 |
| Fast-Sampling | 8.9739E+12(1342.73s) | 40 | 1.0516E+13±9.63E+10 | 0.5826±0.0084 | 0.3736±0.0254 | 13533.55±460.98 |
| Fast-Filtering | | | **9.1579E+12±3.31E+09** | 0.8860±0.0065 | 0.8549±0.0076 | **1153.74±73.15** |
| Fast-Estimation | | | 9.2585E+12±4.64E+08 | **0.9070±0.0002** | **0.8780±0.0002** | 13920.46±847.87 |
| k-means++ | | | 1.4553E+13±1.43E+10 | 0.4146±0.0013 | 0.1681±0.0003 | **3410.89±157.58** |
| Ergun | | | 9.2450E+12±1.15E+07 | 0.9014±0.0000 | 0.8656±0.0001 | 16390.58±1034.44 |
| Det | | | 9.0398E+12±0.00E+00 | 0.8995±0.0000 | 0.8590±0.0000 | 13948.92±794.17 |
| Fast-Sampling | 8.7576E+12(1412.61s) | 50 | 1.0405E+13±9.76E+10 | 0.5582±0.0135 | 0.3143±0.0204 | 13484.68±621.02 |
| Fast-Filtering | | | **8.9381E+12±1.28E+09** | 0.8790±0.0003 | 0.8405±0.0003 | **1157.97±133.86** |
| Fast-Estimation | | | 9.0471E+12±4.11E+08 | **0.9059±0.0003** | **0.8733±0.0005** | 13367.68±1455.83 |

### A.6.2 Visualization for the Distribution of the predicted clusters

In this section, we provide examples of visualizations for the coordinates distribution on some of the datasets used in our experiments.

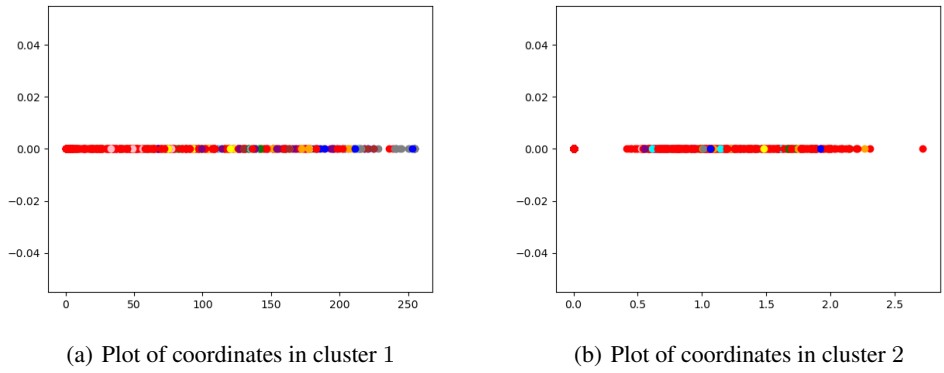

(a) Plot of coordinates in cluster 1       (b) Plot of coordinates in cluster 2

Figure 3: Examples of visualization of the coordinates distribution on dataset CIFAR10, where red points represent correctly predicted data points, and other colors indicate data points that are predicted incorrectly.

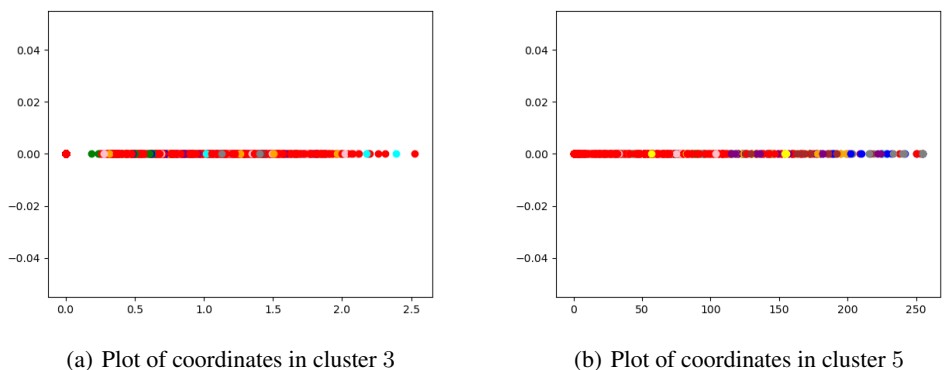

(a) Plot of coordinates in cluster 3       (b) Plot of coordinates in cluster 5

Figure 4: Examples of visualization of the coordinates distribution on dataset PHY, where red points represent correctly predicted data points, and other colors indicate data points that are predicted incorrectly.

