# OpenReview forum: "New Algorithms for the Learning-Augmented k-means Problem"
_ICLR.cc/2025/Conference — ICLR 2025 Poster_

### Official Review · Reviewer_Z7CR · 2024-11-01

**Soundness:** 3
**Presentation:** 2
**Contribution:** 2
**Rating:** 6
**Confidence:** 2

**Summary:**

The paper studies the learning-augmented k-means problem. It proposes three sampling-based algorithms that achieve faster running times compared to previous works. The paper provides theoretical proofs for the algorithms and experimentally tests their cost and time performance.

**Strengths:**

- The paper employs sampling-based techniques, which indeed improve the running time of existing algorithms.

- The paper includes sufficient theoretical analysis, and the experimental evaluation is also very comprehensive.

**Weaknesses:**

- The paper's explanation and intuitive introduction to the learning-augmented k-means problem are not sufficient. I think (Nguyen et al., 2022) has a nice example on this front.

- I'm not an expert in this area, but the technical innovations presented in the paper seem somewhat limited. Intuitively, the algorithm simply performs some sampling operations first and then moves on to subsequent steps, which appears to be a straightforward idea for improving the algorithm's running time.

**Questions:**

- Is this paper the first to use sampling techniques to improve the running time of the learning-augmented k-means problem? The paper does not mention other similar algorithms for comparison.

- In Figure 1, Nguyen clearly outperforms Fast-Estimation; however, in the results presented in the other tables, the two methods show similar performance. Could the authors provide an explanation for this discrepancy?

---

> ### Author Response · Authors · 2024-11-23
> **Response to Reviewer Z7CR**
>
> We thank the reviewer for the constructive comments and thoughtful feedbacks. Below, we address the concerns.
>
> **Weakness 2: I'm not an expert in this area, but the technical innovations presented in the paper seem somewhat limited. Intuitively, the algorithm simply performs some sampling operations first and then moves on to subsequent steps, which appears to be a straightforward idea for improving the algorithm's running time. Is this paper the first to use sampling techniques to improve the running time of the learning-augmented k-means problem? The paper does not mention other similar algorithms for comparison.**
>
> Response: Thanks for pointing these out. Previous methods for the learning-augmented $k$-means problem have also employed sampling-based strategies, but has different objectives compared with ours. For example, Ergun et al. (2021) used sampling to partition each predicted cluster, aiming to separate “true” points from “outliers” (“outliers” are the points that are predicted incorrectly), which takes $O(md)$ times. However, to achieve guarantees for center selection, their approach relies on a sorting process after sampling to evaluate and refine the separation, which takes time $O(md\log m)$. Although with sampling, the total runtime is dominated by the sorting process, leading to an overall runtime of $O(md \log m)$. To further improve the clustering quality, Nguyen et al. (2022) directly applied a sorting-based strategy to identify the densest intervals in each dimension of the predicted clusters, incurring the same $O(md \log m)$ runtime.
>
> The key challenges in enhancing the runtime efficiency of sampling-based methods are twofold: (1) minimizing approximation loss caused by sampling, and (2) bounding the size of the candidate center set based on the sample size to ensure linear runtime complexity.
>
> To address the first challenge, we propose the interval estimation technique. In the sampling process, interval estimation can identify potential locations for optimal clustering centers, which avoids missing high-quality candidates due to sampling errors. For the second challenge, we propose the candidate set construction technique, which further refines the search space by limiting the range of potential center locations. This ensures that the size of the candidate set is bounded relative to the sample size, maintaining linear runtime complexity.
>
> In the context of the learning-augmented $k$-means problem, to achieve runtime improvements, the key obstacle is how to improve the time caused by the sorting process (with runtime $O(md\log m)$). While sampling methods were also explored in Ergun et al. (2021), the objectives and techniques in our algorithms are totally different. Thus, our approach is the first to leverage sampling techniques to improve the runtime inefficiencies caused by the sorting process in the learning-augmented $k$-means problem.
>
> **Question 1: In Figure 1, Nguyen clearly outperforms Fast-Estimation; however, in the results presented in the other tables, the two methods show similar performance. Could the authors provide an explanation for this discrepancy?**
>
> Response: Thanks for raising this question. The plot in Figure 1 illustrates the worst-case clustering quality guarantees for the proposed algorithms, which represent performance under extreme conditions. For the learning-augmented $k$-means problem, the worst-case guarantee can occur in the following condition: if “outliers” (the points that are predicted incorrectly) within a predicted cluster form a group, and the clustering cost caused by these outliers is significantly lower than that of the true points. The algorithms in Figure 1 are compared based on worst-case theoretical guarantees.
>
> However, the worst-case theoretical guarantees may not always reflect practical performance. During our experiments, we observed that outlier aggregation rarely occurs in the datasets used
> (see Figure 3 and Figure 4 in Appendix A.6 for an example), which is the main reason that the two methods in Figure 1 can show similar performance.

---

> > ### Comment · Reviewer_Z7CR · 2024-11-26
> >
> > I appreciate the author's response that addressed my concerns. After reviewing the feedback and considering the other reviews, I raised my score.

---

### Official Review · Reviewer_yxDp · 2024-11-02

**Soundness:** 3
**Presentation:** 3
**Contribution:** 3
**Rating:** 8
**Confidence:** 5

**Summary:**

This paper considers the learning-augmented $k$-means problem. In this problem, the input consists of a data set and a predicted partition $\{P_1,...,P_k\}$. It is assumed that $P_i$ has a symmetric difference at most $\alpha$ fraction to the $k$-th optimal cluster. The goal is still to find a set of $k$ centers to minimize the $k$-means objective (sum of the squared distance to the closest center).

Existing results on this problem achieve a $1+O(\alpha)$ approximation with running time $O(md\log m)$ where $m$ is the size of the data set and $d$ is the dimension. The main contribution of this paper is to remove the $O(\log m)$ factor by introducing sampling-based algorithms. The authors show that one can use sampling to find a good candidate for every dimension of each optimal center by employing some nice properties of $k$-means objectives and the condition that $P_i$ overlaps a lot with the true cluster. Two results are presented, one runs $O(\epsilon^{-1/2} md\log (kd))$ and the other runs in $O(md+\alpha^{-1}\epsilon^{-5}kd)$. Experimental results also show about 6 times speedup upon existing algorithms.

I think this paper is a good addition to the study of learning-augmented $k$-means.

**Strengths:**

1. The running time is linear in the input size which improves previous results by avoiding using sorting.
2. Interesting techniques are used and the theoretical results look solid.
3. The algorithm is based on sampling and is easy to use in practice. The experiment also shows the improvement.

**Weaknesses:**

1. I am unsure if an $O(\log m)$ factor improvement on this problem is super attractive to the community since $O(md\log m)$ already sounds very fast in many cases. Can you explain why avoiding using sorting is so important in this problem?

**Questions:**

1. In Line 153, the range of $\alpha$ should be $(0,\frac{1}{3}-\frac{\epsilon}{3})$? Why $\alpha$ in the Sorting and Fast-Sampling algorithms can be 0 while in the Fast-Estimation case, it must be positive? Does that mean the formers can recover the ground truth (the optimal clusters) while the latter can not?

2. The techniques seem to be specific to $k$-means clustering. Can one obtain similar results for $k$-median?

---

> ### Author Response · Authors · 2024-11-23
> **Response to Reviewer yxDp**
>
> We thank the reviewer for the positive rating and the constructive comments. Below, we address the concerns.
>
> **Weakness 1. I am unsure if an O(logm) factor improvement on this problem is super attractive to the community since already sounds very fast in many cases. Can you explain why avoiding using sorting is so important in this problem?**
>
> Response: Thanks for raising this important question. From a theoretical perspective, achieving an $O(logm)$ factor improvement is both challenging and meaningful in the context of the learning-augmented clustering problem. Since $O(mlogm)$ is the lower bound for comparison-based sorting, achieving an $O(logm)$ runtime improvement requires developing new techniques to surpass the complexity barriers caused by sorting. Our sampling-based strategy eliminates the reliance on sorting, introducing new techniques that enhance the efficiency of the algorithms.
>
> From a practical perspective, even incremental improvements on the running time can lead to substantial benefits, particularly when handling large-scale datasets where computational efficiency is critical. Our experiments (details in Appendix A.6) reveal that algorithms with O(mdlogm) running time face scalability challenges as data sizes grow. For instance, on the SIFT dataset with 100 million data points, existing algorithms require over 4 hours to produce the clustering results. In contrast, our Fast-Filtering algorithm completes the task in just 20 minutes. This further demonstrates that even an O(logm) factor improvement on the running time can enable faster and more scalable clustering solutions for real-world applications.
>
> **Question 1. In Line 153, the range of alpha should be $(0, 1/3-\epsilon/3)$? Why $\alpha$ in the Sorting and Fast-Sampling algorithms can be 0 while in the Fast-Estimation case, it must be positive? Does that mean the formers can recover the ground truth (the optimal clusters) while the latter can not?**
>
> Response: Thanks for raising this interesting question. For the Fast-Estimation algorithm, the sample size used to estimate the clustering cost is proportional to $1/\alpha$. If $\alpha$ approaches zero, the required sample size would grow unbounded, resulting in an impractical runtime. To ensure a bounded runtime, $\alpha$ should remain positive. This requirement contrasts with the sorting-based algorithms, which do not rely on the sampling step for clustering cost estimation.
>
> However, the extreme case for $\alpha = 0$ can be avoided by directly selecting the geometric center for each predicted cluster as the final clustering centers, where ground truth can be recovered when $\alpha = 0$.
>
> **Question 2. The techniques seem to be specific to $k$-means clustering. Can one obtain similar results for $k$-median?**
>
> Response: Thanks for raising this important question. In our revised version, we have included a section (in Appendix A.5) to discuss the extension for the $k$-median objective. We show that our Fast-Sampling method can be extended to the learning-augmented $k$-median clustering problem, where similar results can be obtained. In Appendix A.5, we give an approximation plot and a table to compare our results with previous learning-augmented $k$-median algorithms. The results show that, our method can achieve much better approximation guarantees on clustering quality with slightly worse running time.

---

> > ### Comment · Reviewer_yxDp · 2024-11-25
> >
> > I have read the feedback and other reviews and I will keep my positive rating for this submission.

---

### Official Review · Reviewer_egdN · 2024-11-04

**Soundness:** 4
**Presentation:** 4
**Contribution:** 3
**Rating:** 8
**Confidence:** 3

**Summary:**

In the k-means problem, the input is a set P of m points in d-dimensional Euclidean space and the goal is to find a set C of k centers that minimizes the sum of the square of the distances from P to C. K-means is important problem in machine learning, and has been proven to NP-hard to find a solution with approximation ratio smaller than 1.07. Therefore, the popular practical algorithms for k-means have a large approximation ratio, while the algorithms that promise a small approximation ratio have exponential run-time. To overcome the barrier of inapproximability and still have a fast running speed, the learning-augmented k-means problem becomes popular these year. In the learning-augmented k-means problem, we are given an oracle that predict the label of points with an error rate alpha. The goal of the problem is to solve the k-means problem with the help of such oracle in short time and low approximation ratio.
In this paper, the authors propose three algorithms to improve the run-time compared to the previous research. The best result of previous research provides an algorithm that generates an (1+O(alpha))-approximation in O(md log m) time, while the algorithm in this paper achieves O(md) + ~O(kd/alpha) run-time. The novelty of their algorithms is that the author use a random sampling strategy to generate the solution, while the previous researches all use a sorting-based techniques, which inevitably induce a log m dependency for run-time.
In this paper, the authors first propose the Fast-Sampling algorithm that generates an (1+O(alpha))-approximation in O(md log (kd)) time. They decompose the problem in each dimension, find a good approximation of the coordinates of the centers and then compose the coordinates back to R^d. In Fast-Sampling algorithm, candidate coordinates closes to the coordinates of optimal solution is identified first, and then they use these candidate coordinates to define intervals that capture the optimal solution more precisely.
To further remove the log (kd) dependency in run-time, they propose their second algorithm Fast-Estimation, which generates an (1+O(alpha))-approximation in O(md) + ~O(kd/alpha) run-time. They use a sampling-based strategy to construct an estimator, which can give accurate estimation of cluster cost. With the help of such estimator, they succeed to achieve the O(md) + ~O(kd/alpha) run-time.
Although their algorithms have shorter run-time, the approximation ratio is worse than the algorithm in previous research. Hence, they give their third algorithm, Fast-Filtering, a heuristic that achieve low approximation ratio while still run fast. They also run experiments to test the performance of their algorithms.

**Strengths:**

They remove the log m dependency in the run-time compared to the previous algorithms, which is a huge improvement. Furthermore, they apply a new sampling-based technique to achieve such improvement, which is novel because the sorting-based technique is widely used for previous research. They provide both solid theoretical and practical support to verify their algorithms, which makes their algorithms very trustworthy.

**Weaknesses:**

Their Fast-Sampling and Fast-Estimation have worse approximation ratio compared to previous results. Although their Fast-Filtering performs well in approximation ratio, it is a heuristic and there lacks theoretical support for Fast-Filtering.

**Questions:**

The experiments are only run for Fast-Sampling and Fast-Filtering. What is the performance of Fast-Estimation?

---

> ### Author Response · Authors · 2024-11-23
> **Response to Reviewer egdN**
>
> We thank the reviewer for the positive rating and the constructive comments. Below, we address the concerns.
>
> **Weakness 1: Their Fast-Sampling and Fast-Estimation have worse approximation ratio compared to previous results. Although their Fast-Filtering performs well in approximation ratio, it is a heuristic and there lacks theoretical support for Fast-Filtering.**
>
> Response: Thanks for pointing this out. We agree that our proposed Fast-Sampling and Fast-Estimation algorithms achieve a trade-off between approximation ratio and running time. However, compromising the approximation ratio for faster runtime is natural and beneficial, especially for large-scale datasets where computational efficiency is often the top priority.
>
> For the Fast-Filtering algorithm, we provide a theoretical analysis in the revised version, as detailed in Appendix A.4. Specifically, by modifying the number of nearest neighbors selected (in step 4 and step 6 of the Fast-Filtering algorithm) and fine-tuning the sample sizes $R_1$ and $R_2$, the Fast-Filtering algorithm can achieve a $(1+O(\sqrt{\alpha}))$-approximation. This provides a theoretical guarantee for the heuristic without compromising its practical performances.
>
> Additionally, we have updated the experimental results to reflect these modifications. The revised Fast-Filtering algorithm consistently outperforms other methods across a variety of datasets, achieving much better runtime with improved clustering quality. On average, the Fast-Filtering algorithm is at least 3 times faster than previous learning-augmented algorithms with an average of  1.5% improvements in clustering quality.
>
> **Question 1:The experiments are only run for Fast-Sampling and Fast-Filtering. What is the performance of Fast-Estimation?**
>
> Response: Thanks for raising this question. In our revised version, we have included the experimental results for our proposed Fast-Estimation algorithm, which are presented in Appendix A.6. The results show that the Fast-Estimation algorithm can achieve much better running time on large-scale datasets. On average, the Fast-Estimation algorithm is at least 2 times faster than previous learning-augmented algorithms on large-scale datasets with sizes over 5 million.

---

> > ### Comment · Reviewer_egdN · 2024-11-25
> > **Response to the author**
> >
> > I thank the authors for addressing my concerns. It is a great paper.

---

### Official Review · Reviewer_BxBK · 2024-11-04

**Soundness:** 3
**Presentation:** 3
**Contribution:** 2
**Rating:** 6
**Confidence:** 3

**Summary:**

The authors provide asymptotically faster algorithms for the Learning-Augmented k-means problem by going beyond the initial sorting strategies and instead using sampling-based strategies. Their run time is $\mathcal{O}(m \cdot d) + \mathcal{O}_{\epsilon}(k\cdot d)$ from
$\mathcal{O}(m \cdot d \log m)$. They further propose a heuristic algorithm using their sampling-based ideas that perform faster in real-world datasets compared to the existing learning-augmented methods.

**Strengths:**

The learning augmented K-Means problem seems like an interesting setting (even though considerably less studied/without clear applications), and in practical scenarios, a speed-up of  $\log m$ is a noticeable improvement.

The authors clearly explain their ideas and results and provide several experimental results.

**Weaknesses:**

1. The run-time of the theoretically guaranteed algorithms provided by the authors *does not improve* for the experimental datasets used compared to the existing algorithms.

2. The authors provide a heuristic algorithm to improve their algorithms with guarantees that it has a faster runtime in practice; I think comparing its runtime with algorithms with theoretical guarantees seems unfair. There is a large (formal as well as informal) literature on many different variants of K-Means. One of the primary strengths of the papers by Ergun et al. and Nguyen et al. is that they have theoretical guarantees.

**Questions:**

1) What is the quality of the clustering output for the K-Means algorithms proposed in the paper and those used for reference with respect to the ground truth labeling of the datasets (such as NMI and/or ARI values)?

2) When compared to K-Means++ (without any prediction), what is the improvement in performance on the Learning augmented K-Means algorithms on the datasets the authors have considered? (both in terms of K-Means cost as defined in the paper and ARI, NMI values w.r.t underlying ground truth labels)

---

> ### Author Response · Authors · 2024-11-23
> **Response to Reviewer BxBK**
>
> We thank the reviewer for the constructive comments and thoughtful feedbacks. Below, we address the concerns.
>
> **Weakness 1: The run-time of the theoretically guaranteed algorithms provided by the authors does not improve for the experimental datasets used compared to the existing algorithms.**
>
> Response: Thanks for pointing this out. We acknowledge that our Fast-Sampling algorithm may not demonstrate much faster experimental performance compared to the existing methods. This discrepancy arises because the runtime guarantees of the algorithms are derived from worst-case analysis, which may not always reflect the practical performance. To achieve better practical performance, based on the strategies of the algorithms with theoretical guarantees, we proposed a heuristic algorithm (Fast-Filtering), where the running time is at least 3 times faster than the existing methods with an average improvement of 1.5% on clustering quality. For completeness, in the revised version, we provide a theoretical guarantee for the Fast-Filtering algorithm, demonstrating that it provides a $(1 + O(\sqrt{\alpha}))$-approximation on clustering quality.
>
> **Weakness 2: The authors provide a heuristic algorithm to improve their algorithms with guarantees that it has a faster runtime in practice; I think comparing its runtime with algorithms with theoretical guarantees seems unfair. There is a large (formal as well as informal) literature on many different variants of K-Means. One of the primary strengths of the papers by Ergun et al. and Nguyen et al. is that they have theoretical guarantees.**
>
> Response: Thanks for the valuable feedback. The original purpose of the Fast-Filtering algorithm was to further accelerate learning-augmented algorithms in practice while maintaining or improving clustering quality. From the experimental results, the running time of Fast-Filtering algorithm is at least 3 times faster than the existing methods with an average improvement of 1.5% on clustering quality.
>
> We agree that comparing a heuristic algorithm without theoretical guarantees to algorithms with guarantees may seem unfair. For fair comparison, we give theoretical guarantee analysis for the Fast-Filtering algorithm in the revised version, as detailed in Appendix A.4. To achieve an approximation guarantee, the Fast-Filtering algorithm is slightly modified by adjusting the number of nearest neighbors and the sample sizes $R_1$ and $R_2$ (the modifications are marked as blue color in the paper). With these changes, the Fast-Filtering algorithm achieves a $(1 + O(\sqrt{\alpha}))$-approximation guarantee.
>
> We have also updated the experimental results to reflect these modifications. As shown in Appendix A.6, the revised Fast-Filtering algorithm consistently outperforms other methods across various datasets, delivering improvements in runtime and clustering quality. On average, it is at least 3 times faster than previous learning-augmented algorithms, with an average 1.5% improvement in clustering quality.
>
> **Question 1: What is the quality of the clustering output for the K-Means algorithms proposed in the paper and those used for reference with respect to the ground truth labeling of the datasets (such as NMI and/or ARI values)?**
>
> Response: Thanks for raising this question. In the revised version, we have included the NMI and ARI values for the clustering solutions produced by each algorithm in Appendix A.6. Our proposed algorithms consistently achieve NMI and ARI values above 0.80 across most datasets. Specifically, they exhibit better performance on datasets MNIST and SIFT, which is probably due to the spatial coherence of these datasets. Meanwhile, the Det algorithm outperforms on SUSY, HIGGS, and PHY, which involve high-dimensional data, and Ergun’s algorithm delivers the best results on CIFAR10, which has complex image-based features.
>
> The results show that there is no strict correlation between clustering cost (minimized by the algorithms) and metrics like NMI and ARI. Clustering cost focuses on spatial coherence, while NMI and ARI assess how well the clusters align with the label distributions, which may vary depending on dataset characteristics.

---

> ### Author Response · Authors · 2024-11-23
> **Response to Reviewer BxBK**
>
> **Question 2: When compared to K-Means++ (without any prediction), what is the improvement in performance on the Learning augmented K-Means algorithms on the datasets the authors have considered? (both in terms of K-Means cost as defined in the paper and ARI, NMI values w.r.t underlying ground truth labels)**
>
> Response: Thanks for raising this interesting question. In the revised version, we have included a detailed comparison between our learning-augmented algorithms and the K-Means++ algorithm in Appendix A.6. Compared with K-Means++ algorithm, our proposed algorithms demonstrate better performance in terms of clustering cost, ARI, and NMI values.
>
> On average, our Fast-Filtering algorithm achieves at least a 20% reduction in clustering cost compared to K-Means++. It also provides over 50% improvements in ARI and NMI values across most datasets. These results show the advantage of incorporating learning-augmented and sampling strategies for improving the clustering quality.

---

> > ### Comment · Reviewer_BxBK · 2024-11-26
> >
> > I thank the authors for their detailed review and for taking the time to run the experiments along the lines of my inquiries. The results are definitely promising, and I adjust my score to reflect my position after reading the response.

---

### Author Response · Authors · 2024-11-23
**To all the reviewers: summary of changes**

We appreciate the reviewers for their insightful comments and constructive feedback. Below we will address the concerns separately.

To reflect the reviewers' feedback, we have uploaded an updated version of the manuscript, with most of the changes highlighted in blue. Due to space limit, most of the revisions have been delivered into the Appendix. The key updates include the following:

- Added a motivation discussion in the introduction.
- Provided a theoretical analysis for the proposed Fast-Filtering algorithm in Appendix A.4.
- Included a new subsection discussing the extension to the $k$-median objective in Appendix A.5.
- Conducted additional experiments to evaluate the NMI and ARI values of each algorithm, with most results presented in Appendix A.6.
- Added experimental results for the $k$-means++ and Fast-Estimation algorithms, with most results detailed in Appendix A.6.

---

### Meta-Review · Area_Chair_ARHs · 2024-12-16

**Metareview:**

An interesting paper on an interesting topic for which the reviewers early agreed upon in average. The rebuttal phase bumped up further the scores for the lowest ones. The authors have made substantial updates to the paper, and it is important to polish the camera ready to seamlessly include those updates, eventually by rebalancing the main paper vs the appendices on results (e.g. weakness #2 and Question 2 BxBK).

Most importantly, reviewers had general concerns on what is formal vs what is heuristic, the way to present it vs the state of the art, which brought important updates to the submission ; I am confident nobody would agree on a one size-fits-all way to make a careful, honest and clear accounting of both kinds of contributions to the reader, so I am not going to impose any such blueprint. Rather, the reviewers have made a very good job on pointing key parts and I can only *highly recommend* that the authors take some time to eventually reorganise their narrative by using this important feedback, try to think from the reader's standpoint. It is not an easy task, but the paper being accepted and tackling a relatively recent problem that is likely to bring further work forward, the authors have all incentives to improve its organisation so it takes its rightful place in the state of the art.

**Additional Comments On Reviewer Discussion:**

The most important part was the effort the authors made on the theory vs heuristic part of their work in response to BxBK, egdN. An important update for the final decision.

---

### Decision · Program_Chairs · 2025-01-22

Accept (Poster)